# Functional interdependence of the actin nucleator Cobl and Cobl-like in dendritic arbor development

**Maryam Izadi, Eric Seemann, Dirk Schlobinski, Lukas Schwintzer, Britta Qualmann[†]\*, Michael M Kessels[†]\***

Institute of Biochemistry I, Jena University Hospital/Friedrich-Schiller-University Jena, Jena, Germany

**Abstract** Local actin filament formation is indispensable for development of the dendritic arbor of neurons. We show that, surprisingly, the action of single actin filament-promoting factors was insufficient for powering dendritogenesis. Instead, this required the actin nucleator Cobl and its only evolutionary distant ancestor Cobl-like acting interdependently. This coordination between Cobl-like and Cobl was achieved by physical linkage by syndapins. Syndapin I formed nanodomains at convex plasma membrane areas at the base of protrusive structures and interacted with three motifs in Cobl-like, one of which was $Ca^{2+}$/calmodulin-regulated. Consistently, syndapin I, Cobl-like's newly identified N terminal calmodulin-binding site and the single $Ca^{2+}$/calmodulin-responsive syndapin-binding motif all were critical for Cobl-like's functions. In dendritic arbor development, local $Ca^{2+}$/CaM-controlled actin dynamics thus relies on regulated and physically coordinated interactions of different F-actin formation-promoting factors and only together they have the power to bring about the sophisticated neuronal morphologies required for neuronal network formation in mammals.

**\*For correspondence:**
Britta.Qualmann@med.uni-jena.de (BQ);
Michael.Kessels@med.uni-jena.de (MMK)

[†]These authors contributed equally to this work

**Competing interests:** The authors declare that no competing interests exist.

## Introduction

The actin cytoskeleton is crucial for a huge variety of key processes in cell biology. Yet, only few factors were found that can promote the de novo formation of actin filaments (*Chesarone and Goode, 2009*; *Qualmann and Kessels, 2009*). Thus, the initial idea that each of the discovered actin nucleators may be responsible for the formation of specific, perhaps tissue- or cell-type-specific F-actin structures obviously had to be dismissed as too simple. Theoretically, the required functional diversity in actin filament formation despite a limited set of powerful effectors could be achieved by combinatory mechanisms specific for a given cell biological process. However, experimental evidence for such combinatory actions of actin nucleators is still very sparse. On top of that, which mechanisms may orchestrate these powerful effectors to bring about a certain cellular processes also remains a fundamental question in cell biology.

Neurons need to extend elaborate cellular protrusions – the single signal-sending axon and multiple signal-receiving, highly branched dendrites – to form neuronal networks. These very demanding and specialized cellular morphogenesis processes are driven by the actin cytoskeleton (*Kessels et al., 2011*). The formation of the dendritic arbor involves local $Ca^{2+}$ and calmodulin (CaM) signals coinciding with transient F-actin formation by the evolutionary young actin nucleator Cobl (Cordon-bleu) (*Ahuja et al., 2007*) at dendritic branch induction sites (*Hou et al., 2015*). $Ca^{2+}$/CaM regulates both Cobl's loading with monomeric actin and its different modes of plasma membrane association (*Hou et al., 2015*). Cobl is furthermore regulated via arginine methylation by PRMT2 (*Hou et al., 2018*). All of these aspects were required for Cobl's crucial role in dendritic arbor formation (*Ahuja et al., 2007*; *Haag et al., 2012*; *Hou et al., 2015*, *Hou et al., 2018*).

Interestingly, also Cobl's evolutionary distant ancestor Cobl-like (COBLL1; Coblr1) was recently discovered to be important for Ca²⁺/CaM-controlled neuromorphogenesis (*Izadi et al., 2018*). While Cobl uses three Wiskott-Aldrich syndrome protein Homology 2 (WH2) domains to nucleate actin (*Ahuja et al., 2007*), Cobl-like employs a unique combinatory mechanism of G-actin binding by its single, C terminal WH2 domain and Ca²⁺/CaM-promoted association with the actin-binding protein Abp1 (*Kessels et al., 2000*) to promote F-actin formation (*Izadi et al., 2018*). Cobl-like was also found to interact with cyclin-dependent kinase 1 and to shape prostate cancer cells by not yet fully clear mechanisms (*Takayama et al., 2018*).

Here, we show that Cobl and Cobl-like work at the same nascent dendritic branch sites in a strictly interdependent manner choreographed by physically bridging by the membrane-binding F-BAR protein syndapin I (*Qualmann et al., 1999*; *Itoh et al., 2005*; *Dharmalingam et al., 2009*; *Schwintzer et al., 2011*), which we identified to interact with Cobl-like and to specifically occur in nanoclusters at the convex membrane surfaces at the base of nascent membrane protrusions of developing neurons. The finding that one of the three syndapin binding sites of Cobl-like was regulated by Ca²⁺/CaM unveiled a further important mechanism of local control and coordination of actin dynamics in neuromorphogenesis.

Our work thereby provides insights into how two actin filament formation-promoting components – each critical for dendritic arbor formation – power actin-mediated dendritic branch initiation in a strictly coordinated manner and how this process can be directly linked to local membrane shaping to give rise to the complex morphologies required for proper neuronal network formation.

## Results

### Cobl-like and the actin nucleator Cobl largely phenocopy each other in their critical role in dendritic arborization and seem to work at the same dendritic branching sites

The actin nucleator Cobl and its evolutionary ancestor protein Cobl-like are molecularly quite distinct (*Figure 1—figure supplement 1*), however, both critical for dendritic arbor formation (*Ahuja et al., 2007*; *Izadi et al., 2018*). Side-by-side loss-of-function analysis of both factors in developing primary hippocampal neurons using IMARIS software-based evaluations for detailed analyses of the elaborate morphology of such cells (*Izadi et al., 2018*) revealed surprisingly similar phenotypes (*Figure 1—figure supplement 2A–D*). Dendritic branch and terminal point numbers as well as total dendritic length all were severely affected by lack of Cobl-like (*Figure 1—figure supplement 2E–G*). Corresponding Cobl loss-of-function showed that, while dendritic growth processes seemed largely unaffected by Cobl deficiency, also Cobl deficiency led to a significant reduction of terminal points and in particular to severe loss of dendritic branch points. With −35%, these defects were about as strong as those caused by Cobl-like RNAi (*Figure 1—figure supplement 2E–G*). Cobl RNAi mostly affected dendritic arborization in proximal areas, as demonstrated by evaluations of morphological intersections with concentric circles of increasing size (Sholl analysis; *Sholl, 1953*). Cobl-like RNAi led to reduced Sholl intersections throughout the dendritic arbor (*Figure 1—figure supplement 2H*).

The phenotypical comparison of Cobl and Cobl-like unveiled that both cytoskeletal effectors have somewhat similar functions in dendritic arborization. Colocalization studies showed that Flag-mCherry-Cobl and GFP-Cobl-like did not show any obvious spatial segregation (neither in proximal nor in peripheral dendritic arbor of developing neurons) but largely colocalized. Dendritic accumulations of Cobl usually showed corresponding albeit less pronounced accumulations of Cobl-like (*Figure 1A,B*; arrows). This suggested that Cobl and Cobl-like are not responsible for distinct branching sites but work at the same sites.

### Cobl-like functions in dendritic arborization strictly depend on Cobl and likewise Cobl functions depend on Cobl-like

Observations of two powerful molecular components for actin filament formation at the same place may either reflect functional redundancy and/or parallel action to drive cellular processes effectively in response to (putatively different) signaling cues or may even reflect interlinked functions. Functional redundancy and/or parallel action seemed unlikely, because both individual loss-of-function DIV4-to-DIV6 phenotypes were so severe that about a third of the entire arborization normally

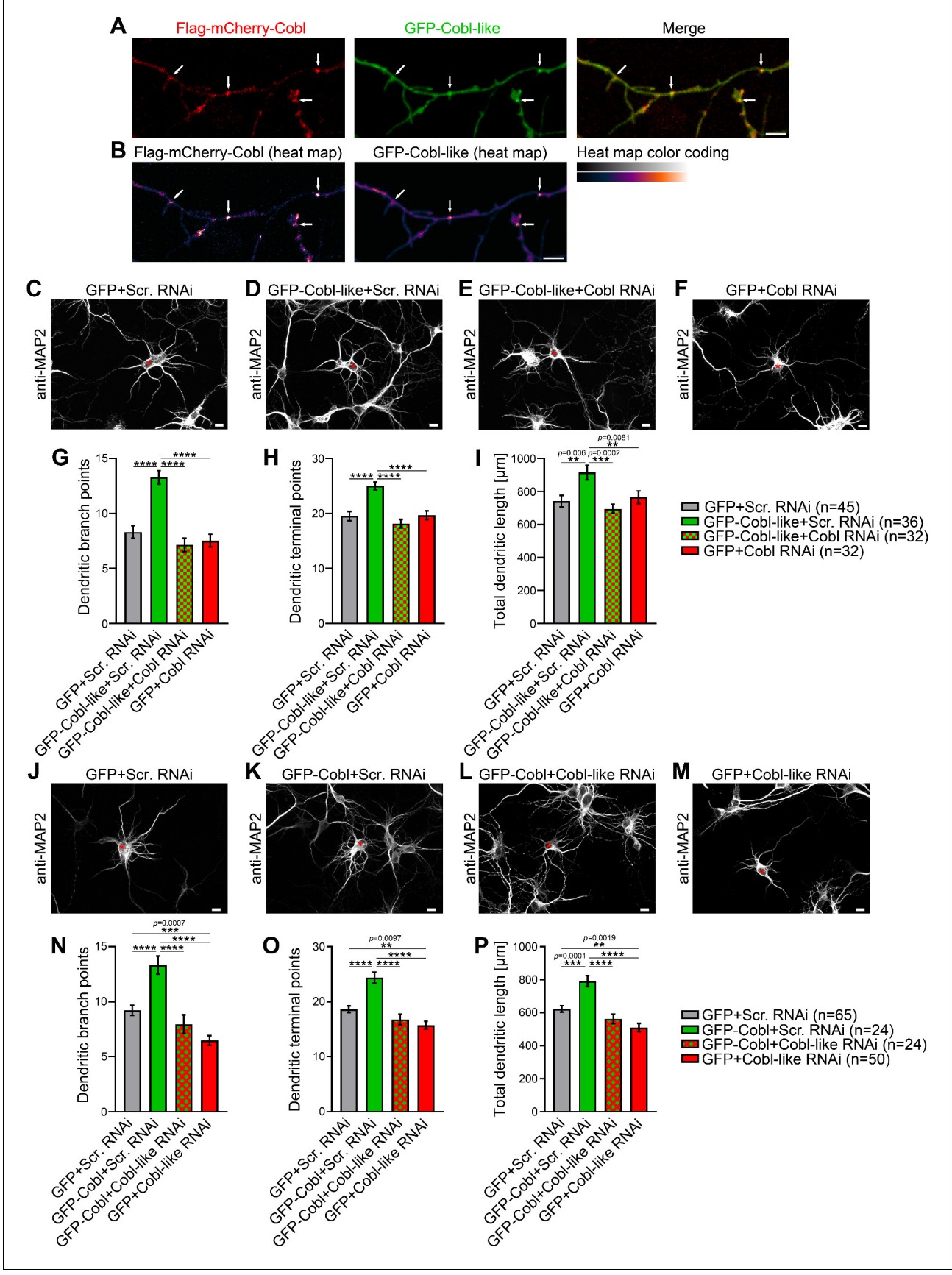

**Figure 1.** Cobl-like and the actin nucleator Cobl work at the same dendritic branching sites and show functional interdependence in dendritic arbor formation. (A,B) Maximum intensity projections (MIPs) of GFP-Cobl-like and Flag-mCherry-Cobl in dendrites of developing hippocampal neurons (DIV6) in standard colors (A) and as heat map representation (B), respectively. Arrows, examples of putative, nascent dendritic branch induction sites with accumulations of both Cobl and Cobl-like. Bar, 5 µm. (C–F) MIPs of neurons showing the suppression of the Cobl-like gain-of-function phenotype (cotransfection at DIV4; fixation 34 hr thereafter) (D; GFP-Cobl-like+Scr. RNAi) by mCherryF-reported RNAi plasmids directed against Cobl (E; GFP-Cobl-like+Cobl RNAi) in comparison to control neurons (C; GFP+Scr. RNAi) and Cobl RNAi neurons (F; GFP+Cobl RNAi). (G–I) Quantitative determinations of indicated dendritic arborization parameters unveiling a full suppression of all Cobl-like functions in dendritic arbor formation by a lack of Cobl. (J–P) Related images (J–M) and quantitative data (N–P) of experiments revealing a functional dependence of Cobl on Cobl-like. Asterisks, transfected neurons. Bars, 10 µm. Data, mean ± SEM. One-way ANOVA+Tukey. Also see *Figure 1—source data 1* and *2*.

The online version of this article includes the following source data and figure supplement(s) for figure 1:

**Source data 1.** Raw data and numerical data graphically presented in *Figure 1G–I*.
**Source data 2.** Raw data and numerical data graphically presented in *Figure 1N–P*.
**Figure supplement 1.** Comparison of domain structures and alignment of Cobl-like with the actin nucleator Cobl.
**Figure supplement 2.** Cobl-like and the actin nucleator Cobl work at the same dendritic branching sites and largely phenocopy each other in their critical role in dendritic arborization.
**Figure supplement 2—source data 1.** Raw data and numerical data graphically presented in *Figure 1—figure supplement 2 A-D*.
**Figure supplement 2—source data 2.** Raw data and numerical data graphically presented in *Figure 1—figure supplement 2E-G*.

reached at DIV6 was lacking (Cobl, −33%; Cobl-like, −39%) (*Figure 1—figure supplement 2E*). Thus, we focused on the remaining hypotheses.

Interestingly, Cobl-like-driven dendritic arborization was completely impaired by the lack of Cobl (*Figure 1C–I*). Dendritic branch point numbers, terminal point numbers, and the total dendritic arbor length of neurons cotransfected with GFP-Cobl-like and Cobl RNAi all were significantly below the values of neurons cotransfected with Cobl-like and scrambled RNAi. The suppression of the Cobl-like gain-of-function effects by Cobl RNAi was so strong that under the chosen condition (only 34 hr expression; GFP coexpression), all three parameters were indistinguishable from control levels. As under the chosen conditions neurons subjected to only Cobl RNAi (+GFP expression) for comparison showed a dendritic arbor development with all three parameters evaluated at about control levels, these experiments permitted the clear conclusion that the effects of Cobl RNAi in Cobl-like-overexpressing neurons did not reflect putative independent and thus merely additive effects on dendritic arborization but clearly represented a full suppression of Cobl-like's functions in dendritic branching in the absence of Cobl (*Figure 1G–I*).

To our surprise, likewise, Cobl-promoted dendritic arbor formation turned out to be massively affected by absence of Cobl-like (*Figure 1J–M*). The dendritic parameters of developing neurons expressing Cobl and Cobl-like RNAi did not show any Cobl gain-of-function phenotype but were statistically not significantly different from those of control or Cobl-like RNAi (*Figure 1N–P*). The fact that the suppressive effects of Cobl-like RNAi on Cobl gain-of-function clearly exceeded the effects of Cobl-like RNAi alone (GFP+Cobl-like RNAi) allowed us to firmly conclude that Cobl functions in dendritic arbor formation depended on Cobl-like. This conclusion was further underscored by the fact that GFP-Cobl+Cobl-like RNAi and GFP+Cobl-like RNAi were statistically not different from each other in any of the three parameters. The lack of any Cobl effects upon coexpression of Cobl-like RNAi thus represented a full and direct suppression of all Cobl functions in the absence of Cobl-like.

Taken together, Cobl and Cobl-like both are cellular factors promoting actin filament formation and significantly differ in their properties, yet, in dendritic branch formation, they do not work independently but surprisingly strongly depend on each other.

## Cobl-like associates with syndapins

The surprising functional interdependence of Cobl and Cobl-like in dendritic arbor formation raised the question how this may be organized mechanistically with two proteins that seem to employ quite different molecular mechanisms (*Ahuja et al., 2007*; *Izadi et al., 2018*).

Using a variety of different methods, we failed to observe any obvious interactions of Cobl and Cobl-like (also see below). Thus, the crosstalk of Cobl-like and Cobl had to be less direct and more sophisticated.

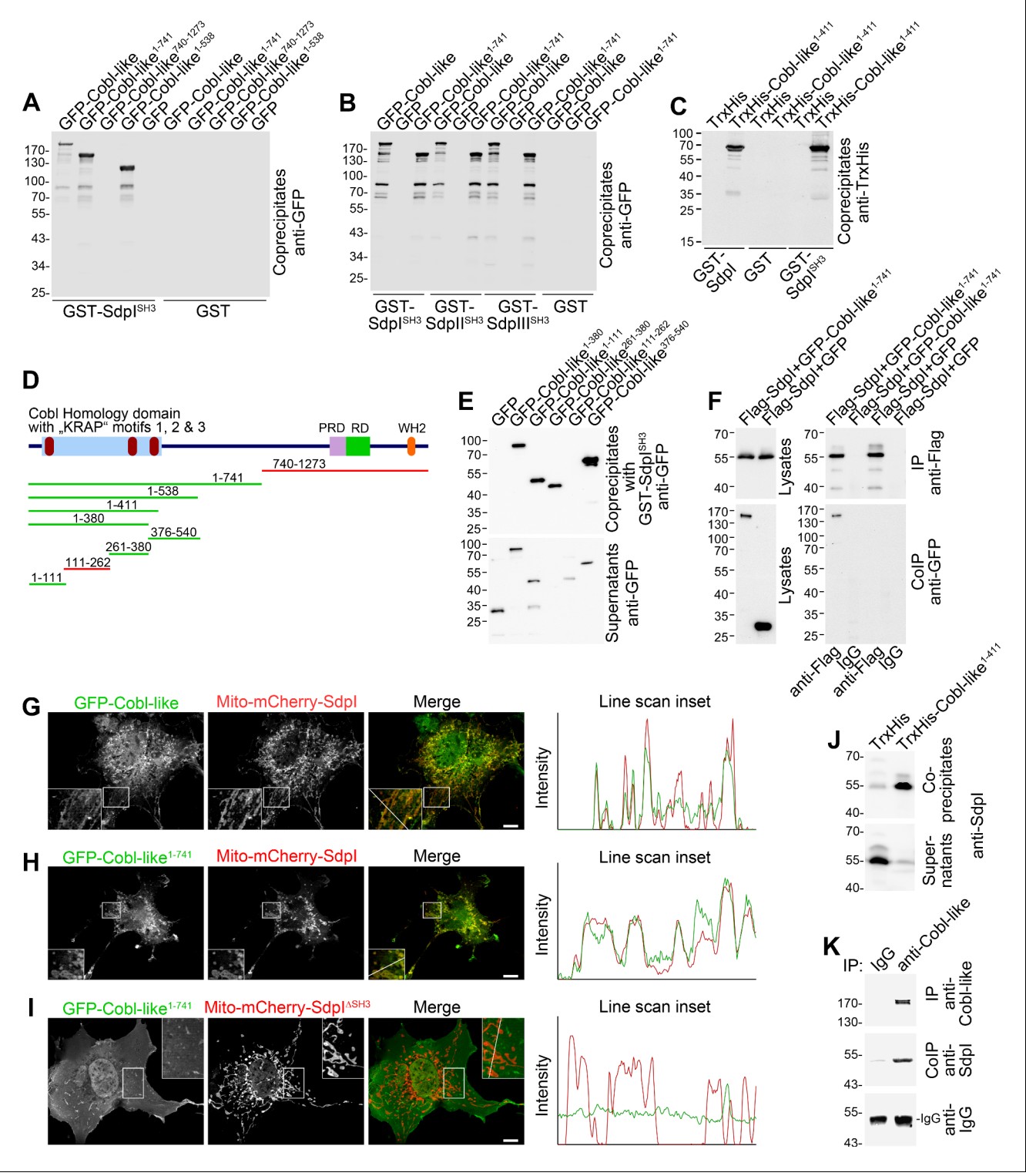

**Figure 2.** Cobl-like associates with syndapins. (**A**) Coprecipitation analyses of GFP-tagged Cobl-like and deletion mutants thereof with immobilized syndapin I SH3 domain (GST-SdpI[SH3]). (**B**) Related coprecipitation studies with the SH3 domains of syndapin I, syndapin II (SdpII[SH3]), and syndapin III (SdpIII[SH3]), respectively. (**C**) Reconstitution of the association of TrxHis-Cobl-like[1-411] with purified GST-syndapin I and GST-syndapin I SH3 domain but not with GST. (**D**) Scheme of Cobl-like with its domains (PRD, proline-rich domain; RD, Abp1-binding repeat domain; WH2, WH2 domain) and deletion mutants used (red, not binding syndapins; green, binding). Red in the Cobl Homology domain, 'KRAP' motifs (not drawn to scale). (**E**) Coprecipitation assays with Cobl-like deletion mutants mapping Cobl-like's syndapin binding sites. (**F**) Coimmunoprecipitations unveiling a specific association of GFP-Cobl-like[1-741] with Flag-syndapin I. (**G–I**) Reconstitution and visualization of Cobl-like/syndapin I complexes in COS-7 cells using mitochondrially

*Figure 2 continued on next page*

*Figure 2 continued*

targeted syndapin I (G,H) as well as a mutant lacking the SH3 domain (Mito-mCherry-SdpI$^{\Delta SH3}$) (I) with GFP-Cobl-like (G) and GFP-Cobl-like$^{1-741}$ (H,I). Boxed areas are shown at higher magnification as insets. Line scans of fluorescence intensities of both channels are along the respective line indicated in the merged insets in **G–I**. Bars, 10 µm. (**J**) Coprecipitation of endogenous syndapin I from mouse brain lysates by TrxHis-Cobl-like$^{1-411}$. (**K**) Endogenous coimmunoprecipitation of Cobl-like and syndapin I from mouse brain lysates.

The online version of this article includes the following figure supplement(s) for figure 2:

**Figure supplement 1.** Cobl-like associates with syndapins.

Cobl was demonstrated to use complexly choreographed membrane-binding mechanisms involving its direct binding partner syndapin I (*Schwintzer et al., 2011*; *Hou et al., 2015*). Syndapins can self-associate (*Kessels and Qualmann, 2006*) and could therefore theoretically link Cobl and Cobl-like physically. As a prerequisite, syndapin I would have to associate with Cobl-like. Indeed, GFP-Cobl-like specifically coprecipitated with immobilized syndapin I SH3 domain. The interaction was mediated by N terminal proline-rich regions of Cobl-like (*Figure 2A*; *Figure 2—figure supplement 1A*) and was conserved among syndapin I, syndapin II, and syndapin III (*Figure 2B*; *Figure 2—figure supplement 1B*).

In vitro reconstitutions with purified components proved that syndapin I/Cobl-like interactions were direct (*Figure 2C*; *Figure 2—figure supplement 1C*) and were furthermore based on classical SH3 domain/PxxP motif interactions, as proven by using a P434L-mutated SH3 domain (*Figure 2—figure supplement 1D*).

Cobl-like deletion mutants (*Figure 2D,E*) showed that specifically three regions in Cobl-like's Cobl Homology domain (Cobl-like$^{1-111}$, Cobl-like$^{261-380}$, and Cobl-like$^{376-540}$) contained syndapin I interfaces (*Figure 2E*, *Figure 2—figure supplement 1E*). Each of them has a single 'KRAP' motif (*Figure 1—figure supplement 1*).

Specific coimmunoprecipitation of GFP-Cobl-like$^{1-741}$ with Flag-tagged syndapin I demonstrated that the identified interaction can also occur in vivo (*Figure 2F*). GFP-Cobl-like$^{1-741}$ also specifically coimmunoprecipitated with Flag-syndapin II-s and Flag-syndapin III (*Figure 2—figure supplement 1F,G*), that is, with syndapin family members showing a wider expression than syndapin I (*Kessels and Qualmann, 2004*).

It was furthermore possible to directly visualize Cobl-like/syndapin I complex formation in intact cells by demonstrating specific recruitments of GFP-Cobl-like and GFP-Cobl-like$^{1-741}$ to mitochondria decorated with Mito-mCherry-syndapin I (*Figure 2G,H*; *Figure 2—figure supplement 1H*). This firmly excluded postsolubilization artifacts, which theoretically could compromise biochemical studies. Deletion of the syndapin I SH3 domain (Mito-mCherry-SdpI$^{\Delta SH3}$) disrupted complex formations with both GFP-Cobl-like$^{1-741}$ (*Figure 2I*) and GFP-Cobl-like full-length (*Figure 2—figure supplement 1I*).

Cobl-like/syndapin interactions also are of relevance in the brain, as immobilized, recombinant TrxHis-tagged Cobl-like$^{1-411}$ specifically precipitated endogenous syndapin I from mouse brain lysates (*Figure 2J*). Furthermore, endogenous Cobl-like/syndapin I complexes in vivo were demonstrated by coimmunoprecipitation analyses from mouse brain lysates (*Figure 2K*).

## Syndapin I is crucial for Cobl-like's ability to promote dendritic arbor extension and branching

We next addressed whether the identified Cobl-like interaction partner syndapin I would indeed be critical for Cobl-like's functions. GFP-Cobl-like massively promoted dendritic arborization already after very short times (*Izadi et al., 2018*). Strikingly, all Cobl-like gain-of-function phenotypes in developing primary hippocampal neurons were completely suppressed upon syndapin I RNAi (*Figure 3A–C*). Cobl-like-overexpressing neurons cotransfected with syndapin I RNAi showed dendritic branch points, dendritic terminal points, and an overall length of the dendritic arbor that were statistically significantly different from Cobl-like-overexpressing neurons and indistinguishable from those of control cells. The syndapin I RNAi-mediated suppression of Cobl-like functions occurred in all dendritic arbor parts affected by Cobl-like gain-of-function (*Figure 3D–G*).

Cobl-like's functions in dendritic arbor formation thus are fully dependent on the availability of its direct interaction partner syndapin I.

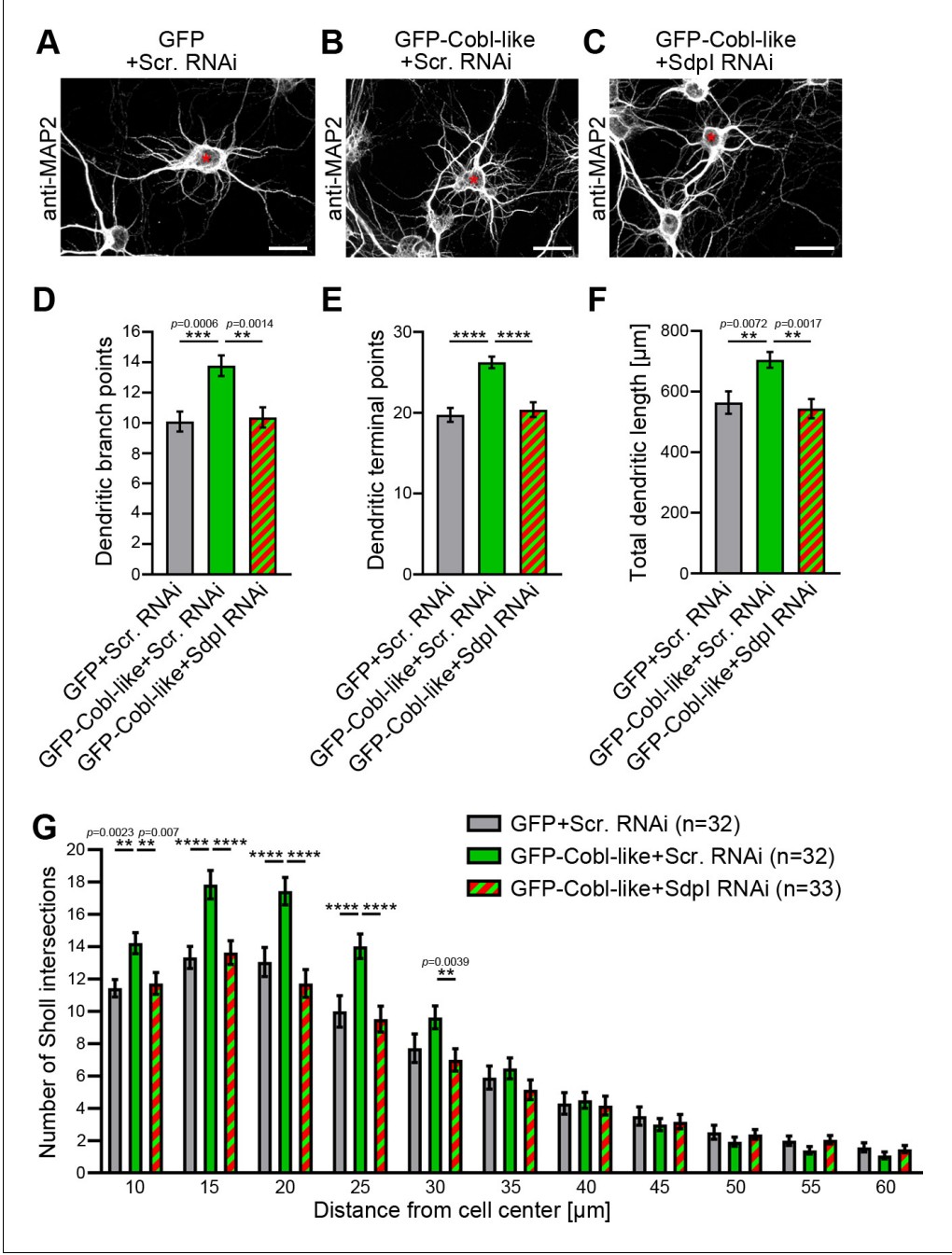

**Figure 3.** Cobl-like functions in dendritic arbor formation rely on syndapin I. (**A–C**) Maximum intensity projections (MIPs) of DIV5.5 neurons transfected as indicated. Asterisks, transfected neurons. Bars, 20 µm. (**D–G**) Quantitative determinations of key dendritic arborization aspects promoted by Cobl-like for their dependence on syndapin I (cotransfection at DIV4; fixation 34 hr thereafter). Data, mean ± SEM. One-way ANOVA+Tukey (**D–F**) and two-way ANOVA+Bonferroni (**G**). Also see *Figure 3—source data 1* and *2*.

The online version of this article includes the following source data for figure 3:

**Source data 1.** Raw data and numerical data graphically presented in *Figure 3D–F*.
**Source data 2.** Raw data and numerical data graphically presented in *Figure 3G*.

## Syndapins physically interconnect Cobl-like with Cobl

In order to unravel molecular mechanisms underlying the strict functional interdependence of Cobl and Cobl-like in dendritic arborization, we asked whether syndapin I may indeed be able to directly bridge the two actin cytoskeletal effectors. To exclude putative indirect interactions via actin, we used immobilized GST-Cobl-like[1-411], which comprises the syndapin binding sites (*Figure 2*). Cobl-like[1-411] indeed formed specific protein complexes with GFP-Cobl[1-713] when syndapin I was present (*Figure 4A*). No GFP-Cobl[1-713] was precipitated when syndapin I was omitted. Thus, direct interactions between Cobl-like and Cobl did not occur but complex formation required syndapin I acting as a bridge (*Figure 4A*). Likewise, also syndapin III mediated complex formation of Cobl-like with Cobl (*Figure 4B*).

Complexes of all three components are also formed at membranes and in intact cells. Mito-GFP-Cobl[1-713] was successfully targeted to the cytosolic membrane of mitochondria and did not only successfully recruit syndapin I to mitochondrial membranes (*Figure 4—figure supplement 1*) but was also able to recruit Cobl-like[1-741] (*Figure 4C–H*).

The visualized complex formations (*Figure 4C,F*) were specific and mediated by syndapin I acting as bridging component between Cobl and Cobl-like, as omitting syndapin I did not lead to any Cobl-like[1-741] mitochondrial presence and also Mito-GFP did not lead to any syndapin/Cobl-like colocalization at mitochondria (*Figure 4D,E,G,H*).

## Syndapin I and Cobl-like colocalize at sites of dendritic branch induction

In line with the BAR domain hypothesis (*Peter et al., 2004*; *Qualmann et al., 2011*; *Daumke et al., 2014*; *Kessels and Qualmann, 2015*; *Carman and Dominguez, 2018*), syndapin I may sense/induce certain membrane topologies and thereby provide spatial cues for Cobl and Cobl-like functions. Thus, three syndapin I-related aspects needed to be experimentally addressed: (i) Where and when do Cobl-like and syndapin I occur together in developing neurons? (ii) Would a given syndapin I localization indeed reflect specifically membrane-associated syndapin I? (iii) Would putative accumulations of membrane-associated syndapin I then really correspond to convex membrane topologies?

Dual time-lapse imaging of GFP-Cobl-like and syndapin I-mRubyRFP in developing primary hippocampal neurons showed that syndapin I accumulated in defined spots along dendrites coinciding with subsequent branch induction events. Such accumulations occurred as early as 1 min prior to detectable dendritic branch protrusion, were spatially restricted to very small areas (diameters, ~250–1200 nm), and were spatially and temporally colocalized with Cobl-like at branch initiation sites (*Figure 5A,B*).

Afterward, the accumulations of both proteins at the base of newly formed protrusions faded. This suggested a highly mobile subpool of syndapin I and Cobl-like in the dendritic arbor.

Sites with repetitive dendritic protrusion attempts showed accumulations of both syndapin I and Cobl-like prior to the first as well as prior to the second protrusion initiation (*Figure 5A,B*).

Determinations of maximal fluorescence intensities prior to dendritic branch induction showed that Cobl-like and syndapin I but also syndapin's binding partner Cobl as well as the calcium sensor protein CaM showed accumulations that were about 75–100% above neighbored control regions in the same dendrite. In contrast, control fluorescent protein (mCherry) significantly differed and did not show such accumulations (*Figure 5C*). Maximal accumulations of all four proteins hereby occurred in a time window of in average −40 to −25 s prior to protrusion induction (*Figure 5D*). Interestingly, despite the observed high variances of especially Cobl-like and syndapin I, these two players in dendritic branch induction seemed to be accumulating slightly earlier than Cobl.

Taken together our observations show that Cobl-like, syndapin I, Cobl, and CaM all accumulate a branch initiation sites prior to dendritic branch induction and show some temporal overlap at these particular sites.

## Membrane-bound syndapin I occurs preferentially at protrusive membrane topologies in developing neurons and forms nanoclusters at such sites

3D time-lapse studies do not resolve whether the observed syndapin I accumulations at nascent branch sites represent membrane-associated syndapin I or a cytosolic subpool, for example, associated with putative cytoskeletal components at such sites. Immunogold labeling of freeze-fractured

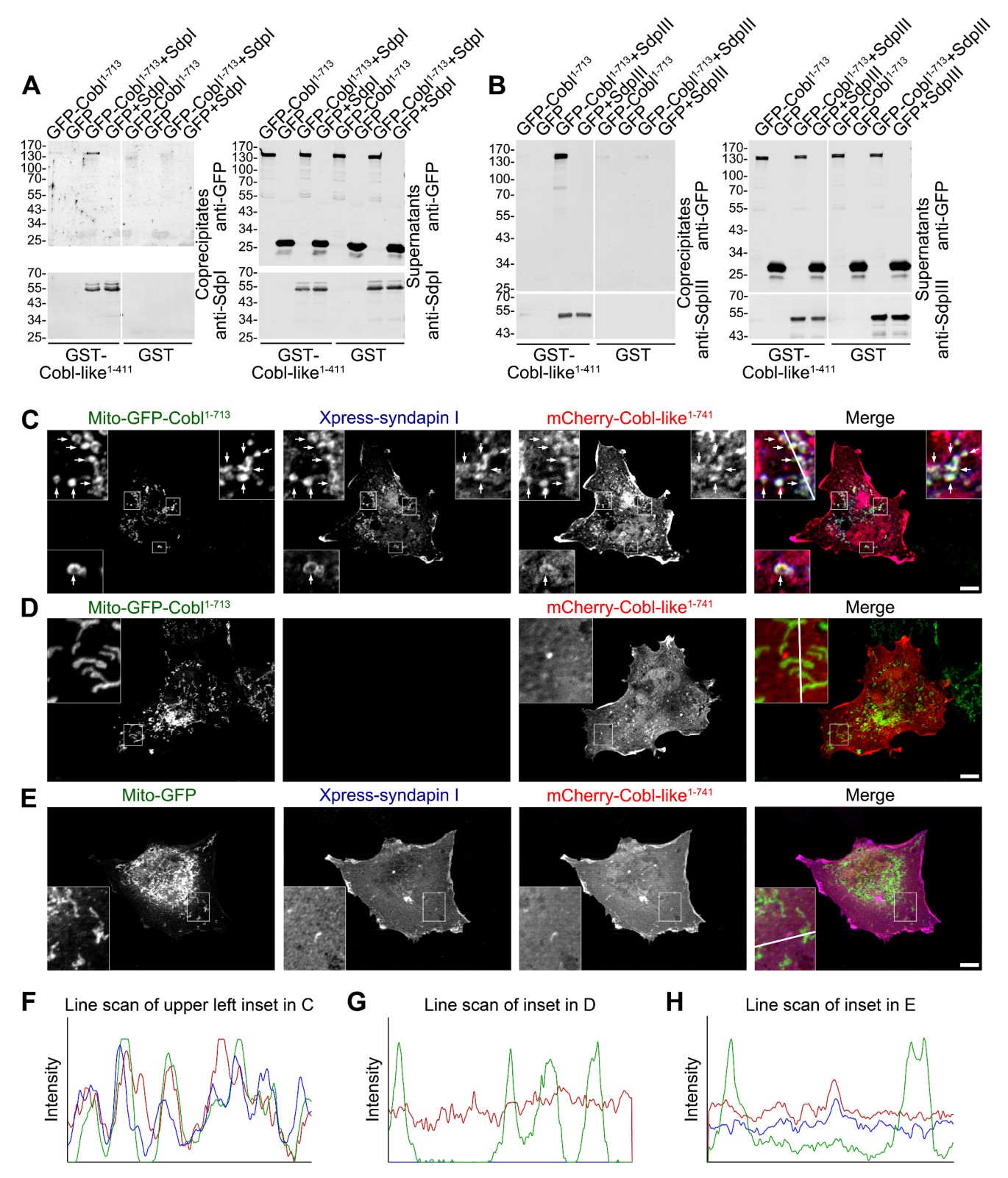

**Figure 4.** Cobl-like is physically linked to Cobl via syndapin I acting as a bridging component. (**A,B**) Coprecipitation analyses unveiling specific and syndapin-dependent formation of complexes composed of immobilized GST-Cobl-like, syndapin I (**A**) and syndapin III (**B**), respectively, as well as GFP-Cobl[1-713]. White lines indicate omitted blot lanes. (**C–E**) Reconstitution and visualization of Cobl-like/syndapin I/Cobl protein complexes in COS-7 cells. Mito-GFP-Cobl[1-713] (**C,D**) but not Mito-GFP (**E**) recruited mCherry-Cobl-like[1-741] in the presence of Xpress-syndapin I (**C**) but not in its absence (**D**).

*Figure 4 continued on next page*

*Figure 4 continued*

Boxes in **C–E** areas presented as magnified insets (**C,D**, fourfold; **E**, threefold). Arrows, examples of colocalization of all three channels. Boxed areas are shown at higher magnification as insets. (**F–H**) Line scans of fluorescence intensities of all three channels are along the respective line indicated in the insets of **C–E**. Bars, 10 μm.

The online version of this article includes the following figure supplement(s) for figure 4:

**Figure supplement 1.** Recruitment of syndapin I to mitochondrial surfaces in intact cells by Mito-GFP-Cobl[1-713].

plasma membranes is a technique that per se exclusively focuses on membrane-integrated proteins, provides membrane topology information, and can be applied to neuronal networks (e.g. see *Tanaka et al., 2005*; *Holderith et al., 2012*; *Schneider et al., 2014*; *Nakamura et al., 2015*). We have recently shown that plasma membranes of still developing neurons can in principle be freeze-fractured and immunolabeled, too (*Wolf et al., 2019*).

Whereas neither Cobl nor Cobl-like seemed to be preservable by the procedure, immunogold labeling of syndapin I, which can insert hydrophobic wedges into one membrane leaflet (*Wang et al., 2009*), was successfully obtained (*Figure 6A–B*). In principle, anti-syndapin I immuno-gold labeling was seen at both cylindrical and protrusive membrane topologies. However, even at the conditions of saturated labeling applied, cylindrical membrane surfaces merely showed sparse anti-syndapin I immunogold labeling and were mostly decorated with single gold particles or by pairs of labels. In contrast, at protrusive sites, the labeling density was about three times as high as at cylindrical surfaces (*Figure 6A,A',B*).

Protrusive sites also showed a statistically highly significant enrichment of syndapin I nanoclusters (≥3 anti-syndapin I labels in circular ROIs of 35 nm radius) (*Figure 6B–D*). Interestingly, syndapin I was usually not localized to the tip of the protrusion but preferentially occurred at membrane topol-ogy transition zones at the protrusion base (*Figure 6A''*).

The accumulation of syndapin I clusters at such sites was in line with a promotion of membrane curvature induction and/or with a stabilization of the complex membrane topologies found at such sites by syndapin I.

## Cobl-like's N terminus is a target for the Ca²⁺ sensor CaM and Ca²⁺ signals increase Cobl-like's associations with syndapin I

The formation of neuronal networks involves local Ca²⁺ and CaM signals, which coincide with tran-sient F-actin formation at sites of dendritic branch induction (*Hou et al., 2015*). Cobl-like was identi-fied as Ca²⁺/CaM target. Yet, this CaM association occurred in the C terminal part of Cobl-like and regulated Cobl-like's association with the F-actin-binding protein Abp1 (*Izadi et al., 2018*). Interest-ingly, also GFP-Cobl-like[1-411] showed Ca²⁺-dependent CaM binding, whereas middle parts, such as Cobl-like[376-540] and Cobl-like[537-740], did not (*Figure 7A,B*).

Surprisingly, further analyses demonstrated that the central parts of the Cobl Homology domain of Cobl-like, that is, Cobl-like[111-262] and Cobl-like[182-272], both also did not show any Ca²⁺-dependent CaM binding (*Figure 7A,B*), although the central Cobl Homology domain corresponds to the CaM-binding area in Cobl (*Hou et al., 2015*) and represents an area of at least moderately higher sequence conservation between Cobl and Cobl-like (33% identity; *Figure 1—figure supplement 1*). Instead, it was the most N terminal part of Cobl-like represented by Cobl-like[1-111] and Cobl-like[1-58] that was targeted by CaM (*Figure 7A,B*).

Coprecipitation experiments with purified recombinant proteins confirmed that Ca²⁺/CaM and syndapin I can bind Cobl-like[1-411] simultaneously (*Figure 7C*). We hypothesized that the discovered Ca²⁺/CaM binding to the Cobl-like N terminus may play a role in regulating the syndapin binding of the neighbored Cobl Homology domain. Quantitative syndapin I coimmunoprecipitation experi-ments with Cobl-like[1-741] demonstrated an improved complex formation of Cobl-like with syndapin I when Ca²⁺ was added. With an increase of ~70%, syndapin I binding to Cobl-like[1-741] turned out to be massively promoted by Ca²⁺ (*Figure 7D,E*).

Thus, the identified N terminal complex formation with syndapin I is Ca²⁺/CaM-regulated.

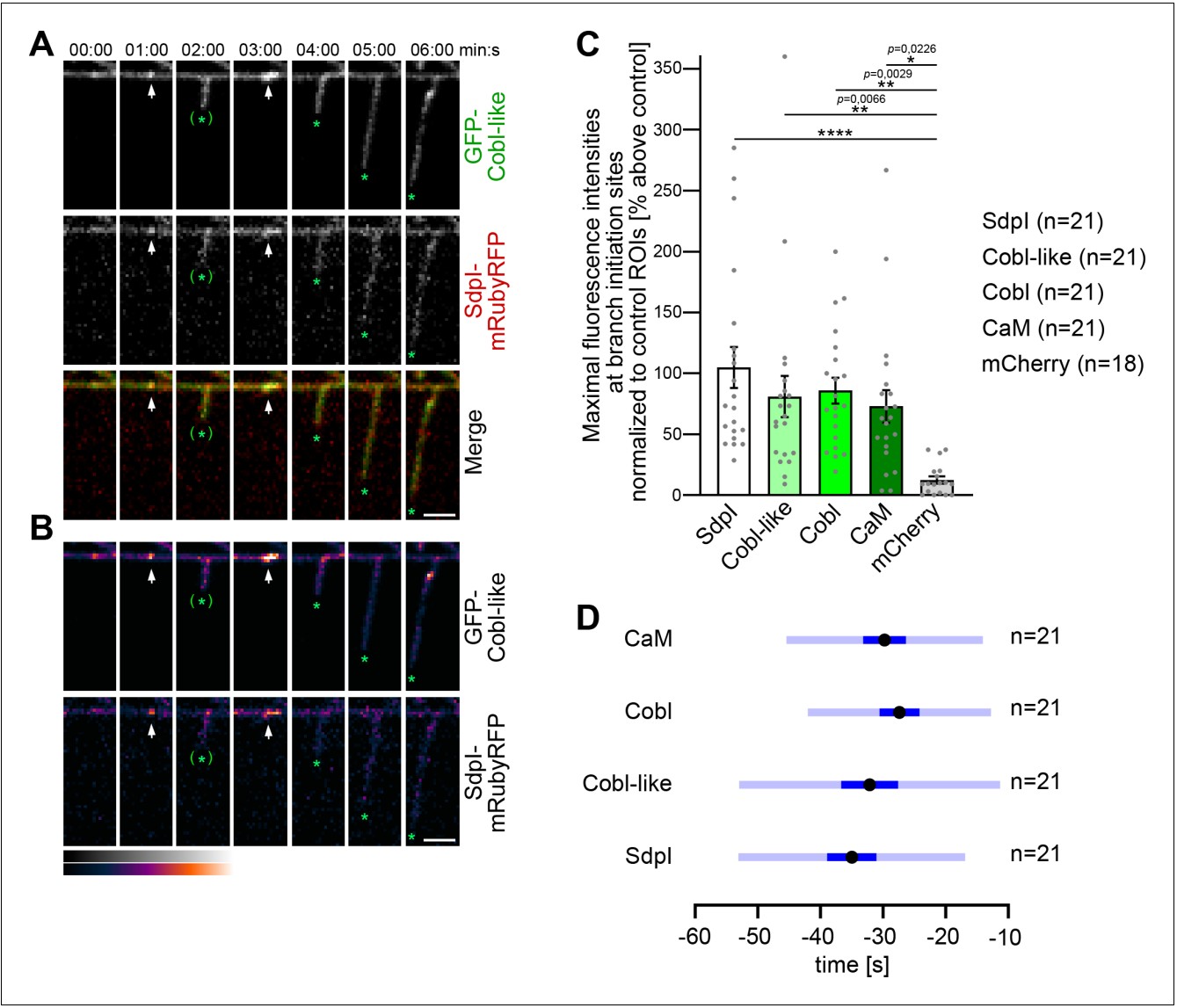

**Figure 5.** Cobl-like and syndapin I coincide at nascent dendritic branch sites. (A) Maximum intensity projections (MIPs) of individual frames of a 3D time-lapse recording of a dendrite segment of a DIV7 rat hippocampal neuron coexpressing GFP-Cobl-like and syndapin I (SdpI)-mRubyRFP. Arrows, GFP-Cobl-like and syndapin I enrichments prior to protrusion initiation from these dendritic sites; *, tips of growing dendritic protrusions; (*), abandoned protrusions. (B) Heat map representations. Bars, 2.5 µm. (C) Quantitation of maximal intensities of fluorescent syndapin I, Cobl-like, Cobl and CaM fusion proteins as well as of mCherry as control at dendritic branch induction sites prior to protrusion formation (time frame: the six 10 s frames prior to protrusion start) in relation to a control ROI at the same dendrite at a position neighboring the branch induction site (shown as % above this control ROI). Data, mean ± SEM. Bar/dot plot overlay of individual data points averaged. (D) Temporal analyses of the maximal fluorescence intensities of syndapin I, Cobl-like, Cobl, and CaM occurring at dendritic branch induction sites prior to protrusion formation. Data, mean (black dot) ± SD (light blue) and ± SEM (dark blue). One-way-ANOVA (C). Also see *Figure 5—source data 1*.

The online version of this article includes the following source data for figure 5:

**Source data 1.** Raw data and numerical data graphically presented in *Figure 5C,D*.

## Cobl-like's N terminal CaM-binding site regulating syndapin I association levels is crucial for dendritic arbor formation

The N terminal region of Cobl-like (*Figure 1—figure supplement 1*) indeed contains putative CaM-binding motifs. Coprecipitation analyses clearly showed that, in contrast to GFP-Cobl-like[1-741], a corresponding deletion mutant (GFP-Cobl-like[1-741ΔCaM NT]; GFP-Cobl-like[1-741Δ11-45]) did not show any $Ca^{2+}$-dependent CaM binding (*Figure 8A*).

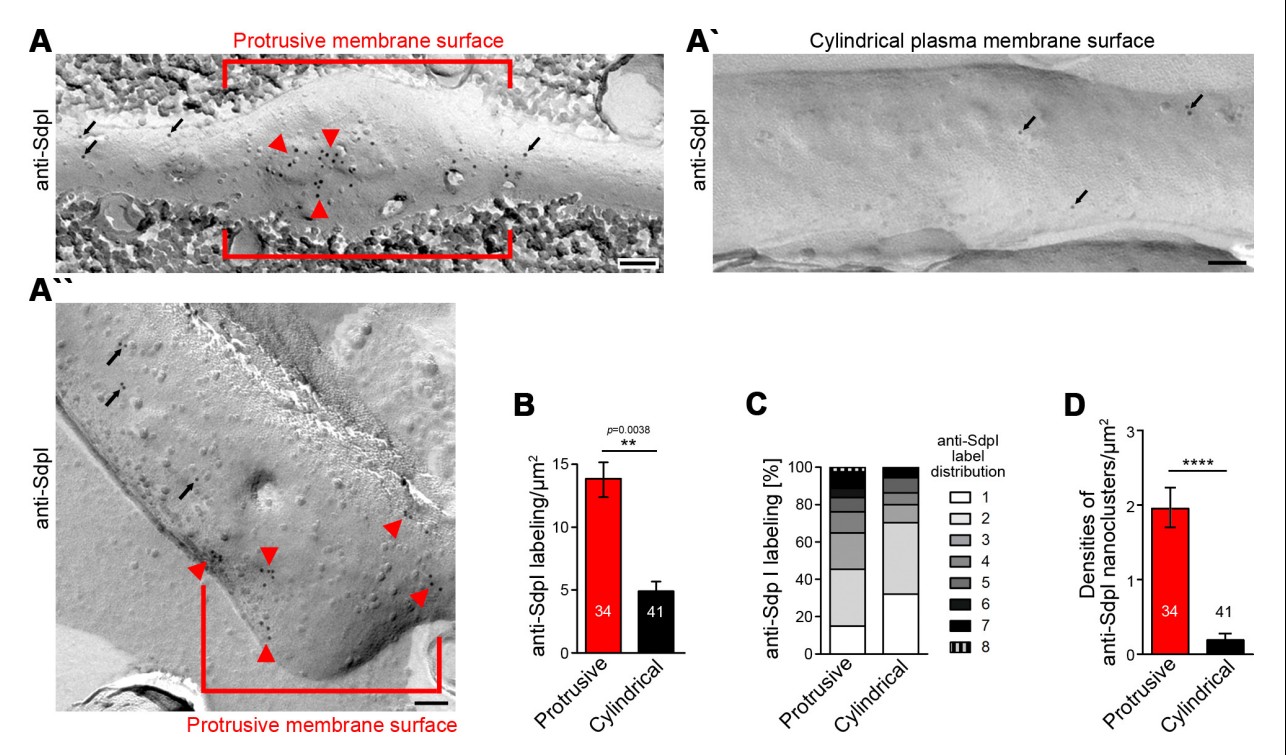

**Figure 6.** Syndapin I nanoclusters are enriched at sites of dendritic protrusion. (**A,A',A''**) Transmission electron microscopy (TEM) images of anti-syndapin I immunogold-labeled freeze-fracture replica of developing neurons (DIV7). Red lines highlight membrane topologies protruding from regular cylindrical topology. Arrowheads, abundant and clustered anti-syndapin I immunogold labeling (10 nm) at protrusive sites. Arrows, sparse and rarely clustered anti-syndapin immunogold labeling at regular, cylindrical membrane structures. Bars, 200 nm. (**B**) Quantitative evaluations of anti-syndapin I labeling densities at protrusive and cylindrical membrane topologies. (**C**) Quantitative analysis of the relative abundance of differently clustered syndapin I labels (ROIs, 35 nm radius). In total, 335 (protrusive) and 130 (cylindrical) labels were evaluated. (**D**) Quantitative analysis of the density of anti-syndapin I nanoclusters (≥3 anti-syndapin I immunogold labels/ROI) at regular cylindrical membrane surfaces and at those with protrusive topology. Data (**B,D**), mean ± SEM. One-way ANOVA (**B**); two-tailed Student's t-test (**D**). Also see *Figure 6—source data 1*.

The online version of this article includes the following source data for figure 6:

**Source data 1.** Raw data and numerical data graphically presented in *Figure 6B–D*.

Strikingly, a RNAi-resistant (*) Cobl-like mutant solely lacking the N terminal CaM-binding site (GFP-Cobl-like*$^{\Delta CaM\ NT}$) failed to rescue the Cobl-like loss-of-function phenotypes in dendritic arborization (*Figure 8B,C*). Quantitative analyses unveiled that reexpression of GFP-Cobl-like*$^{\Delta CaM\ NT}$ instead of resupplying the neurons with RNAi-insensitive wild-type (WT) Cobl-like*, which rescued all Cobl-like deficiency phenotypes, was unable to rescue the RNAi-mediated defects in dendritic branch point numbers, terminal point numbers, and total dendritic length. These defects were as severe as those caused by Cobl-like RNAi without rescue attempt (*Figure 8D–F*).

Also Sholl analyses confirmed that GFP-Cobl-like*$^{\Delta CaM\ NT}$ showed a significant lack of rescue performance in all proximal and central parts of the dendritic arbor (see all Sholl intersections up to 30 μm) when compared to Cobl-like RNAi/GFP-Cobl-like* (*Figure 8G*).

The identified N terminal CaM-binding site of Cobl-like regulating the syndapin I interactions thus was absolutely indispensable for Cobl-like's functions in dendritic arbor formation.

## Ca²⁺/CaM signaling exclusively promotes the syndapin I association with the first of the three 'KRAP' motifs

The critical N terminal CaM-binding site was adjacent to the most N terminal of the three syndapin binding areas. As a prerequisite for further analyses uncovering the regulatory mechanism, we next confirmed that the interactions with syndapins were indeed solely mediated by the 'KRAP' motif-containing regions. Both Cobl-like$^{1-741\Delta KRAP}$ and Cobl-like$^{\Delta KRAP}$ indeed were not able to interact with

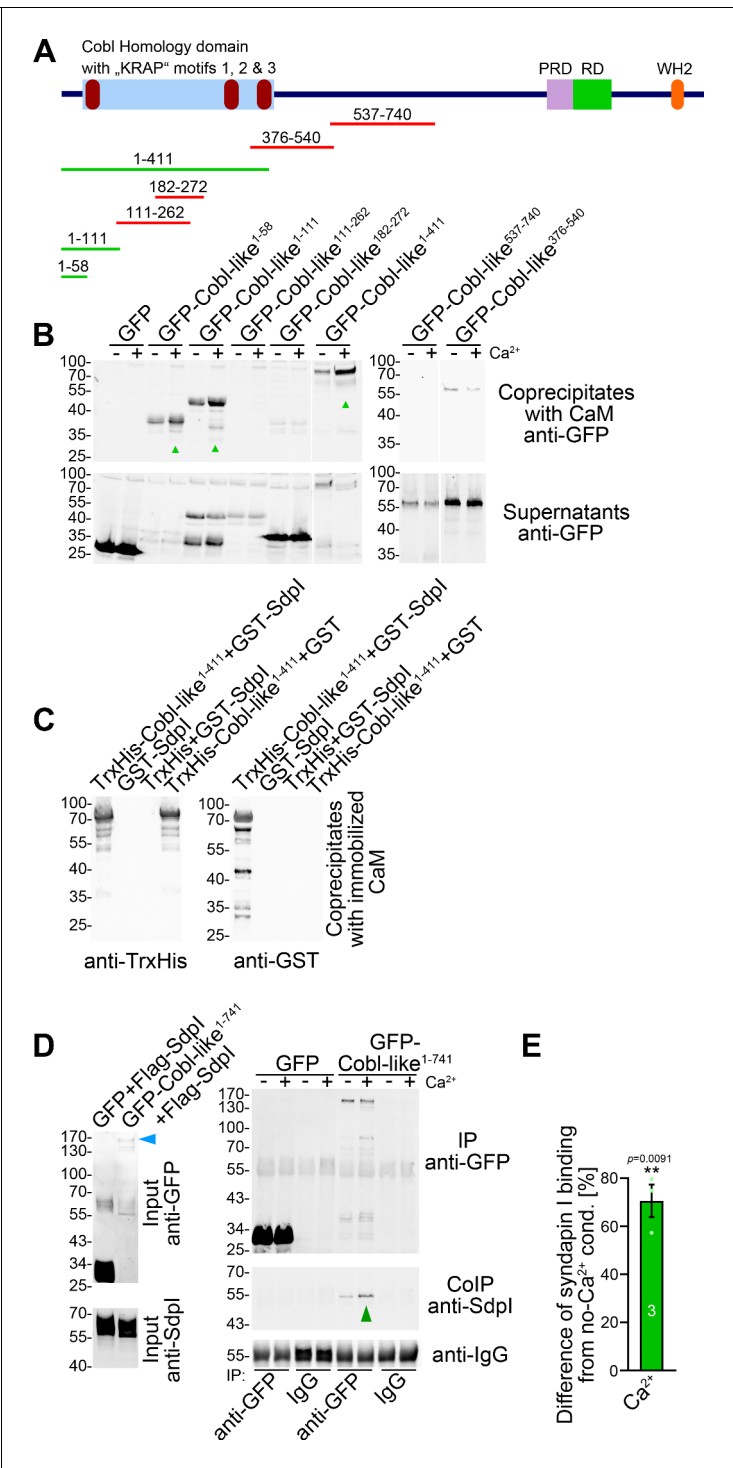

**Figure 7.** Ca$^{2+}$/CaM associates with the N terminus of Cobl-like and positively regulates Cobl-like's syndapin I association. (**A**) Scheme of Cobl-like and deletion mutants used for CaM-binding studies (**B**) (red, no Ca$^{2+}$-dependent binding; green, Ca$^{2+}$-dependent binding). (**B**) Coprecipitations with immobilized CaM in presence (500 µM) and absence of Ca$^{2+}$ and different Cobl-like deletion mutants. Green arrowheads, increased CaM interactions in the presence of Ca$^{2+}$. White lines, lanes omitted from blots. (**C**) Coprecipitation analyses with immobilized CaM and purified TrxHis-Cobl-like$^{1-411}$ and GST-syndapin I (GST-SdpI) showing direct and simultaneous interactions of Cobl-like$^{1-411}$ with both CaM and syndapin I. (**D,E**) Quantitative coimmunoprecipitation analyses demonstrating that Ca$^{2+}$/CaM signaling leads to increased syndapin I coimmunoprecipitation with Cobl-like$^{1-741}$. Blue arrowhead, position of the only faintly detected GFP-Cobl-like$^{1-741}$ in the lysates (**D**). Green arrowhead, increase of coimmunoprecipitated syndapin I (**D**). (**E**) Anti-syndapin I signal per immunoprecipitated Cobl-like (expressed as change from conditions without Ca$^{2+}$). Data, bar/dot plot overlays with mean ± SEM. Unpaired Student's t-test. Also see *Figure 7—source data 1*.

*Figure 7 continued on next page*

*Figure 7 continued*

The online version of this article includes the following source data for figure 7:

**Source data 1.** Raw data and numerical data graphically presented in *Figure 7E*.

the syndapin I SH3 domain, as shown by coprecipitation studies and by reconstitutions of complex formations with syndapin I in vivo (*Figure 9A*; *Figure 9—figure supplement 1*).

Strikingly, quantitative coimmunoprecipitation analyses unveiled a full abolishment of the about 50% increase of syndapin I interaction with the Cobl Homology domain of Cobl-like (Cobl-like$^{1-457}$) upon Ca$^{2+}$ addition when the first 'KRAP' motif (KRAP1) was deleted (Cobl-like$^{1-457\Delta KRAP1}$; Cobl-like$^{1-457\Delta 59-69}$) (*Figure 9B–D*). This insensitivity of Cobl-like$^{1-457\Delta KRAP1}$ to Ca$^{2+}$/CaM signaling revealed that it was exclusively the first 'KRAP' motif (aa59-69) that was regulated by Ca$^{2+}$/CaM signaling.

Side-by-side analyses of GFP-Cobl-like$^{1-457}$ and Cobl-like$^{1-457\Delta KRAP1}$ under Ca$^{2+}$-free control conditions revealed that without Ca$^{2+}$ Cobl-like$^{1-457}$ and the corresponding $\Delta$KRAP1 mutant thereof coimmunoprecipitated the same amount of syndapin I (*Figure 9E,F*). Thus, without Ca$^{2+}$ and under the stringency of in vivo conditions, as reflected by coimmunoprecipitations, 'KRAP' motif 1 seemed not to contribute to syndapin I complex formation but awaited activation by Ca$^{2+}$/CaM signaling.

## The single Ca$^{2+}$/CaM-regulated syndapin I binding site of Cobl-like is crucial for Cobl-like's function in dendritic arbor formation

A lack of proper syndapin I interaction by deletion of 'KRAP' motif 1 resulted in a reduced localization of Cobl-like to the cortex of developing neurons when endogenous Cobl-like was replaced by GFP-Cobl-like*$^{\Delta KRAP1}$ (*Figure 9G–I*).

In line with the importance of the identified N terminal CaM-binding site, also deletion of only the first, that is, the Ca$^{2+}$/CaM-regulated, syndapin I binding interface was as detrimental for Cobl-like's critical functions in dendritic arborization as lacking the entire N terminal part of Cobl-like all together (GFP-Cobl-like*$^{\Delta 1-412}$). Both GFP-Cobl-like*$^{\Delta 1-412}$ and GFP-Cobl-like*$^{\Delta KRAP1}$ completely failed to rescue the Cobl-like loss-of-function phenotypes in dendritic arborization (*Figure 9J–L*).

Instead, cotransfections with Cobl-like RNAi in both cases merely led to dendritic morphologies identical to those of neurons deficient for Cobl-like. The dendritic branch points, terminal points, and total dendritic length all remained significantly reduced in comparison to control neurons (scrambled RNAi/GFP) and did not differ from those of Cobl-like RNAi neurons (*Figure 9J–L*).

This complete failure to rescue any of the Cobl-like loss-of-function phenotypes in dendritic arborization demonstrated that the Ca$^{2+}$/CaM-regulated 'KRAP' motif 1 of Cobl-like is absolutely critical for Cobl-like's functions in dendritic arbor formation.

Together, our analyses unveiled that the actin nucleator Cobl and its distant relative Cobl-like – each of them critical for dendritic arbor formation – in fact need to cooperate with each other in a syndapin-coordinated and Ca$^{2+}$/CaM-regulated manner to bring about the complex morphology of hippocampal neurons (*Figure 10*).

## Discussion

Development of proper dendritic arbors of neuronal cells is key for the complex brains of vertebrates, as neuronal morphologies have direct consequences for brain organization patterns, cell-cell connectivity, and information processing within neuronal networks. Here, we show that this fundamental process is powered by the coordinated, strictly interdependent action of two components, which both promote the formation of actin filaments at the cell cortex, the actin nucleator Cobl (*Ahuja et al., 2007*) and its only distant relative Cobl-like (*Izadi et al., 2018*).

Cobl-like is already present in bilateria and considered as an evolutionary ancestor of the actin nucleator Cobl. Yet, we did not observe any redundant or additive functions in dendritic arborization of developing neurons. Instead, Cobl and Cobl-like both enriched at the same nascent dendritic branching sites and their functions were cooperative and each crucial for dendritic branch induction. Our findings that Cobl-like interacts with the F-BAR domain protein syndapin I providing links to Cobl and that the syndapin I-binding N terminal part of Cobl-like is regulated by Ca$^{2+}$/CaM signaling

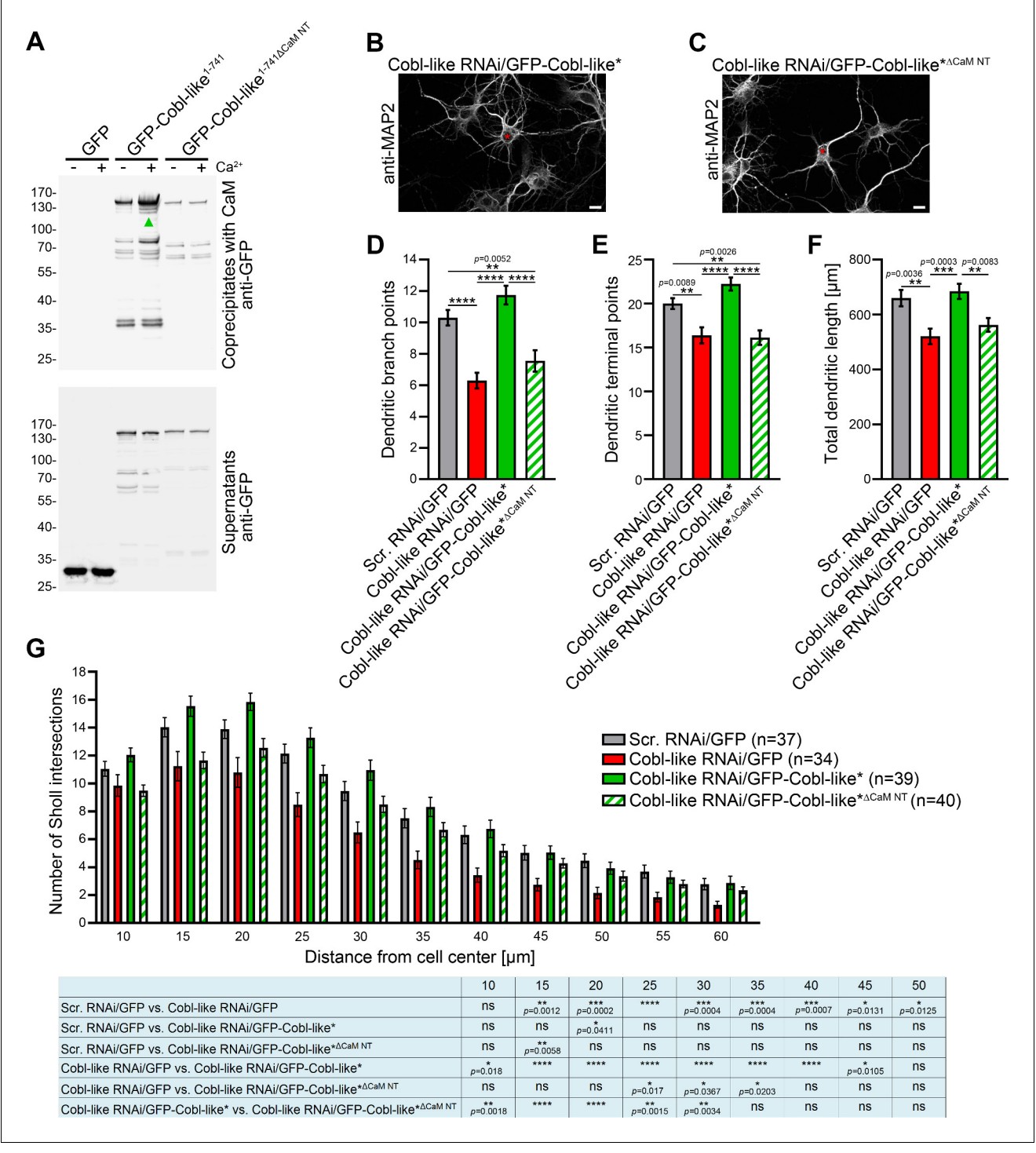

**Figure 8.** The N terminal CaM-binding site of Cobl-like is indispensable for all critical functions of Cobl-like in dendritic arbor formation. (A) Coprecipitation analyses of Cobl-like$^{1-741}$, Cobl-like$^{1-741\Delta CaM\ NT}$ ($\Delta 11$–$45$), and GFP with immobilized CaM in Ca$^{2+}$ presence and absence. Arrowhead, increased CaM interaction of Cobl-like$^{1-741}$ upon Ca$^{2+}$ (disrupted in Cobl-like$^{1-741\Delta CaM\ NT}$). (B,C) Functional analyses in primary hippocampal neurons (transfection at DIV4; fixation 37 hr thereafter) unveiling that an RNAi-insensitive (*) Cobl-like mutant lacking the N terminal CaM-binding site (GFP-Cobl-like*$^{\Delta CaM\ NT}$) failed to rescue the Cobl-like loss-of-function phenotypes. Red asterisks, transfected neurons (transfection, DIV4; analyses, DIV5.5). Bars, 10 µm. (D–G) Quantitative evaluations of indicated dendritic parameters. Data, mean ± SEM. One-way ANOVA+Tukey (D–F); two-way ANOVA +Bonferroni (G). Also see *Figure 8—source data 1* and *2*.

The online version of this article includes the following source data for figure 8:

**Source data 1.** Raw data and numerical data graphically presented in *Figure 8D–F*.
**Source data 2.** Raw data and numerical data graphically presented in *Figure 8G*.

in a positive manner unveil the mechanisms of the striking functional interdependence of Cobl and Cobl-like (*Figure 10*).

The identified syndapin/Cobl-like interactions were mediated by three 'KRAP' motif-containing regions in Cobl-like and the SH3 domain of syndapin I. Cobl-like's 'KRAP' motifs are highly conserved among each other and among different species (consensus, Kr+APxpP). They furthermore show similarity to those of Cobl (consensus, KrRAPpPP) (*Schwintzer et al., 2011*), as well as to other mapped syndapin I binding sites, such as RRQAPPPP in dynamin I (*Anggono and Robinson, 2007*), RKKAPPPPKR in ProSAP1/Shank2 (*Schneider et al., 2014*), and KKPPPAKPVIP in the glycine receptor beta subunit (*del Pino et al., 2014*).

Coprecipitation of endogenous syndapin I with Cobl-like from brain extracts, coimmunoprecipitations of endogenous Cobl-like and syndapin I from mouse brain lysates, as well as visual proof of complex formation in intact cells underscore the in vivo relevance of the Cobl-like/syndapin I interactions we identified.

Syndapin I/Cobl-like interactions clearly were of functional importance, as syndapin I deficiency completely suppressed Cobl-like-mediated dendritic arbor formation. In line, syndapin I accumulated together with Cobl-like at nascent dendritic branch sites and membrane-bound syndapin I clusters were found at convex membrane curvatures at the base of protrusions in developing neurons. With their topology changes in different directions, these membrane areas fit the structure of the membrane-binding F-BAR domain of syndapin I, which seems unique among the BAR protein superfamily (*Qualmann et al., 2011*) and shows overall curvature but also a strongly kinked tilde shape (*Wang et al., 2009*).

We furthermore demonstrated biochemically and in intact cells that Cobl-like and Cobl can physically be interconnected by syndapin I acting as a bridge. Physical interconnection of Cobl and Cobl-like by syndapin I provides a plausible molecular mechanism for the striking functional interdependence of Cobl and Cobl-like and is in line with syndapin I/Cobl interactions (*Schwintzer et al., 2011*) as well as with syndapin I's F-BAR domain-mediated self-association ability (*Kessels and Qualmann, 2006*; *Shimada et al., 2007*; *Wang et al., 2009*; *Figure 10*). This would leave the SH3 domain of each syndapin I free for recruiting effector proteins, for spatially organizing them at specific, curved membrane areas at nascent dendritic branch sites and for thereby coordinating their functions in dendritic branch induction. Consistently, all three components of complexes composed of Cobl-like, syndapin I, and Cobl showed spatial coordination at branch induction sites. These data were in line with previous reports of accumulations of Cobl-like and of Cobl at dendritic branch induction sites (*Izadi et al., 2018*; *Hou et al., 2015*). Our quantitative determinations of the time points of maximal accumulation prior to dendritic branch induction also demonstrated that, besides being spatially coordinated, both cytoskeletal components and syndapin I interlinking them also are temporally coordinated during dendritic arborization and accumulate together −40 to −25 s prior to dendritic branch induction.

Importantly, we found that the interlinkage of Cobl-like and Cobl was not static. The Cobl-like/syndapin I interaction was regulated by Ca$^{2+}$ signals. This is well in line with the involvement of transient, local Ca$^{2+}$ signals in dendritic arborization of developing neurons (*Rajan and Cline, 1998*; *Fink et al., 2003*; *Gaudillière et al., 2004*), with the spatial and temporal coordination of CaM with all of these players at nascent dendritic branch points and with membrane targeting and cytoskeletal functions of Cobl also being controlled by Ca$^{2+}$/CaM (*Hou et al., 2015*). The additional Ca$^{2+}$/CaM regulation of the Cobl-like/syndapin I interface would now provide another key regulatory mechanism right at the interlinking bridge between Cobl and Cobl-like.

The regulatory mechanism is based on an N terminal stretch of amino acids of Cobl-like proteins in front of the so-called Cobl Homology domain, which is absent in Cobl proteins. With 'KRAP1', Ca$^{2+}$ signaling allowed for the modulation of specifically one out of three syndapin I binding sites of Cobl-like (*Figure 10*). This particular, Ca$^{2+}$/CaM-regulated syndapin I binding interface played an important role in the recruitment of Cobl-like to the cell cortex of developing neurons. The decline of the cortical Cobl-like localization we observed upon 'KRAP' motif 1 deletion suggested that, at least in the cell body, Ca$^{2+}$ levels are usually high enough to ensure that first 'KRAP motif' binds to

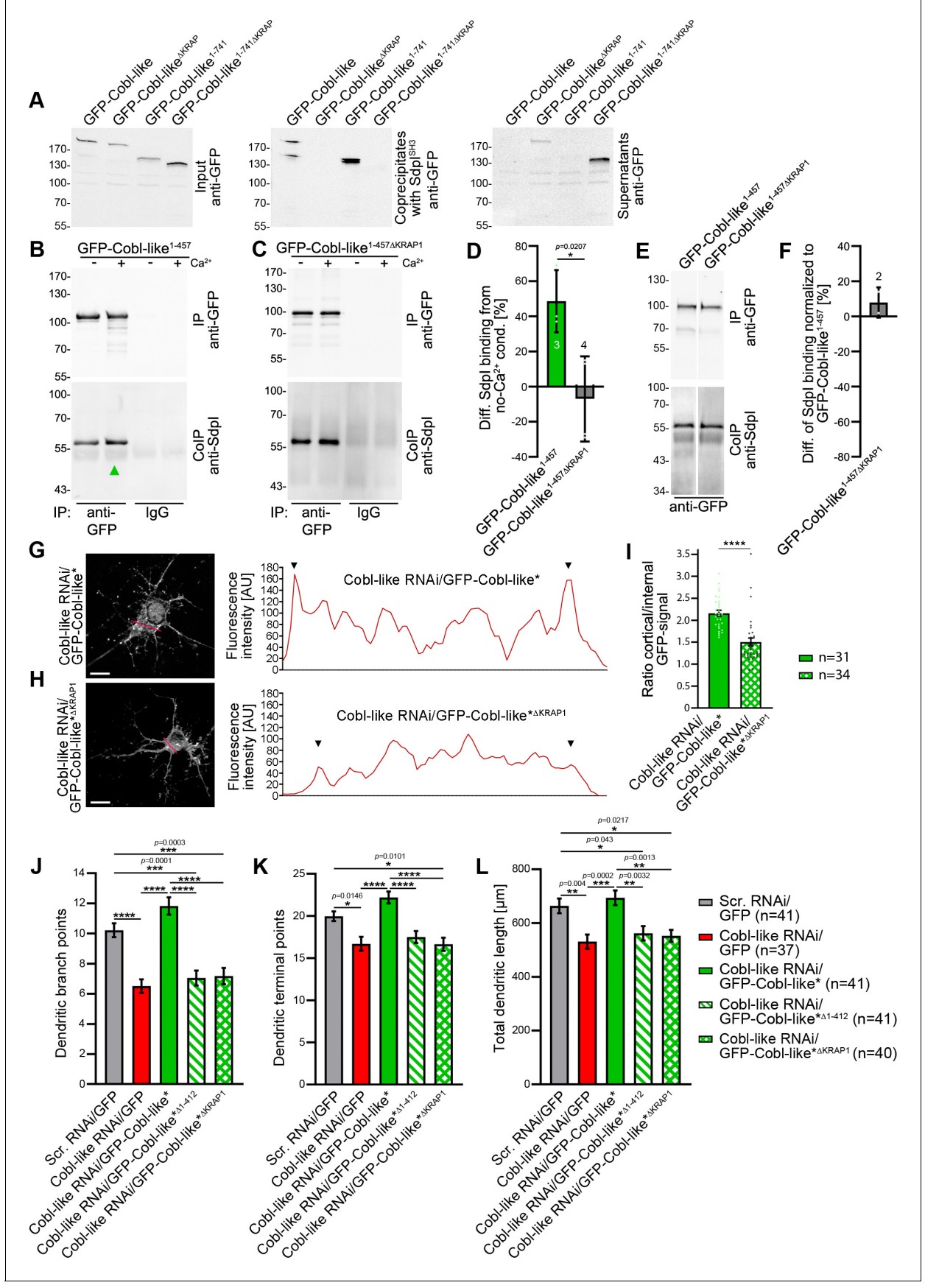

**Figure 9.** Ca$^{2+}$/CaM signaling exclusively promotes the syndapin I association with the first of the three 'KRAP' motifs and this single Ca$^{2+}$/CaM-regulated motif is crucial for Cobl-like's functions. (A) Coprecipitations with immobilized syndapin I SH3 domain (SdpI$^{SH3}$) and Cobl-like and ΔKRAP mutants thereof. (B,C) Quantitative coimmunoprecipitation analyses with GFP-Cobl-like$^{1-457}$ (B) in comparison to a corresponding mutant solely lacking the first 'KRAP' motif (GFP-Cobl-like$^{1-457ΔKRAP1}$) in the presence and absence of Ca$^{2+}$, respectively (C). Arrowhead, increase of coimmunoprecipitated Flag-syndapin I with GFP-Cobl-like$^{1-457}$ upon Ca$^{2+}$. (D) Quantitation of anti-syndapin I coimmunoprecipitation upon Ca$^{2+}$ presence normalized to immunoprecipitated GFP-Cobl-like$^{1-457}$ and GFP-Cobl-like$^{1-457ΔKRAP1}$, respectively (as deviation from conditions without Ca$^{2+}$). (E,F) Side-by-side comparison of syndapin I coimmunoprecipitations with GFP-Cobl-like$^{1-457}$ and GFP-Cobl-like$^{1-457ΔKRAP1}$ (E) and quantitative analysis thereof (F). White line, lanes omitted from blot. (G,H) Images of Apotome sections showing the cortical localization of GFP-Cobl-like expressed in Cobl-like-deficient background (Cobl-like RNAi/GFP-Cobl-like*) in primary hippocampal neurons transfected at DIV4 and imaged 37 hr later (G) compared to the subcellular distribution of GFP-Cobl-like*$^{ΔKRAP1}$ in the same background (H). Lines indicate positions of line scans shown. Bars, 10 μm. (I) Quantitative assessment of cortical GFP intensities (marked with arrowheads in the line scans) normalized to the GFP intensity of an internal ROI in the same cell. (J–K) Functional analyses of the importance of Cobl-like's CaM-regulated syndapin I binding site (KRAP1) by loss-of-function rescue experiments evaluating the indicated dendritic arbor parameters of developing neurons (transfection, DIV4; analysis, DIV5.5). Note that neither a Cobl-like mutant lacking the entire N terminal part (GFP-Cobl-like*$^{Δ1-412}$) nor GFP-Cobl-like*$^{ΔKRAP1}$ was able to rescue Cobl-like's loss-of-function phenotypes. Data, bar/dot plot overlays with mean ± SEM (D,I) and mean ± absolute error (F) as well as bar plots with mean ± SEM (J–L). Unpaired Student's t-test (D,F,I); one-way ANOVA+Tukey (J–L). Also see *Figure 9—source data 1–4*.

The online version of this article includes the following source data and figure supplement(s) for figure 9:

**Source data 1.** Raw data and numerical data graphically presented in *Figure 9D*.
**Source data 2.** Raw data and numerical data graphically presented in *Figure 9F*.
**Source data 3.** Raw data and numerical data graphically presented in *Figure 9I*.
**Source data 4.** Raw data and numerical data graphically presented in *Figure 9J–L*.
**Figure supplement 1.** The three 'KRAP' motifs are critical for Cobl-like's association with syndapin I.

some syndapin I and that the 'KRAP' motif 1 is important for interactions with the plasma membrane-binding and shaping protein syndapin I in developing neurons (*Figure 9G–I*). Importantly, Cobl-like mutants, which either solely lacked the N terminal CaM-binding site or the single Ca$^{2+}$/CaM-regulated syndapin binding 'KRAP1', both completely failed to rescue Cobl-like loss-of-function phenotypes in dendritic branching. Thus, explicitly the N terminal CaM association regulating the 'KRAP1'/syndapin I interaction and consistently also the 'KRAP1' were absolutely critical for Cobl-like's functions in dendritogenesis.

Our data clearly show that dendritic arborization of developing neurons requires the Ca$^{2+}$/CaM- and syndapin I-coordinated, joined action of Cobl and Cobl-like. In other cells and/or cellular processes, Cobl and Cobl-like also seem to have their independent, individual functions, such as the critical role of Cobl in F-actin formation right beneath the sensory apparatus of outer hair cells in the inner ear, the loss of which correlated with defects in pericentriolar material organization, in postnatal planar cell polarity refinement and in hearing (*Haag et al., 2018*). Further studies of *Cobl* knockout (KO) mice unveiled an importance of Cobl for a specialized set of filaments interconnecting structural elements in the F-actin-rich terminal web of microvilli-decorated epithelial cells in the small intestine. However, there were no indications of an additional Cobl-like involvement (*Beer et al., 2020*). Also in proteomic analyses of myoblasts, which upon IGFN1 deficiency show altered G-to-F-actin ratios, only Cobl was identified but not Cobl-like (*Cracknell et al., 2020*).

Likewise, apart from dendritic branching of neurons studied here, there are no hints on any Cobl roles in functions that the *Cobl-like* (Cobll-1) gene has been linked to, such as diabetes and obesity (*Mancina et al., 2013*; *Sharma et al., 2017*). Cobl-like was also suggested as biomarker for different cancer types (*Gordon et al., 2003*; *Gordon et al., 2009*; *Wang et al., 2013*; *Han et al., 2017*; *Plešingerová et al., 2018*; *Takayama et al., 2018*), to be suppressed by Epstein-Barr virus infection (*Gillman et al., 2018*) and to be involved in B-cell development (*Plešingerová et al., 2018*) but there are no hints on Cobl roles in any of these processes.

The extension of very fine and elaborately branched cellular structures over hundreds of micrometers, as in dendritogenesis of neurons, certainly represents an extreme and rather special case of cellular morphogenesis. It is therefore well conceivable that a joined action of both Cobl and Cobl-like is required to promote actin filament formation at locally restricted sites to drive further branching. It currently seems plausible that Cobl and Cobl-like's different actin filament formation mechanisms – spatial rearrangement of three actin monomers by the three WH2 domains of Cobl generating actin nuclei (*Ahuja et al., 2007*) *versus* use of the single WH2 domain of Cobl-like and the actin-binding cofactor Abp1 in a structurally not fully understood trans-mechanism (*Izadi et al.,*

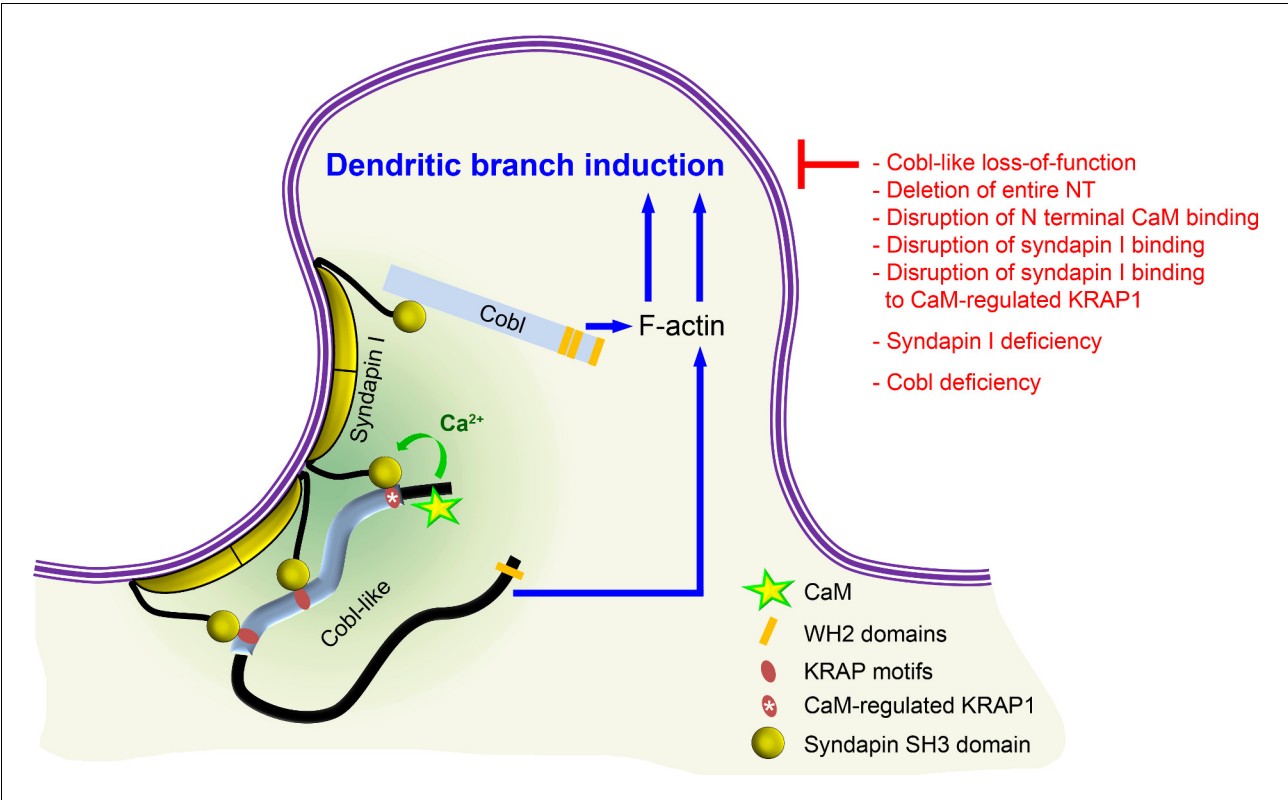

**Figure 10.** Model depicting how Cobl and Cobl-like functions in dendritic branch initiation are joined, coordinated, and controlled. Cobl and Cobl-like functions are not only both critical for dendritic branch formation but both factors promoting the formation of actin filaments were found to act in an interdependent manner. The underlying mechanisms of coordination and control are depicted and include physical linkage of Cobl and Cobl-like by syndapin I forming dimers and multimeric clusters at the convexly bent membrane areas at the base of nascent branch sites. The newly identified interaction of syndapin I with Cobl-like is mediated by three independent KRAP motifs (red), the most N terminal of which (marked by white asterisk) is regulated by a newly discovered CaM association to Cobl-like's N terminus. All mechanistic aspects unveiled in this study are depicted in detail and the corresponding functional evaluations conducted are listed in brief. The WH2 domains of Cobl and Cobl-like are shown to indicate the C terminal domains of both proteins and their cytoskeletal functions.

*2018*) – and their different modes of regulation by Ca$^{2+}$ (*Hou et al., 2015*; *Izadi et al., 2018*) represent the distinct functions of Cobl and Cobl-like that have to be combined to power dendritic arborization. The recently identified Cobl regulation by PRMT2-mediated arginine methylation (*Hou et al., 2018*) may potentially also provide a unique aspect that needs to be integrated into the joined, interdependent function of Cobl and Cobl-like in dendritic branch induction.

Although to our knowledge, besides the here reported functions of Cobl and Cobl-like and some not yet fully explored crosstalk of the actin nucleator Cobl with the Arp2/3 complex (*Schwintzer et al., 2011*), no information on functional cooperations of any actin nucleators is available for any actin cytoskeletal process in the function or development of hippocampal neurons, some initial studies on other actin nucleators also hint toward more interactive roles than initially thought. The actin nucleator Spire associates with and works with the actin nucleator formin 2 in *Drosophila* oocytes (*Quinlan et al., 2007*; *Pfender et al., 2011*; *Montaville et al., 2014*). Also in actin filament formation linked to DNA damage, Spire and formin 2 seem to be involved together (*Belin et al., 2015*). The formin mDia1 was reported to synergize with the APC protein (*Okada et al., 2010*; *Breitsprecher et al., 2012*). mDia1 was furthermore very recently found to indirectly interact with the Arp2/3 complex functionally cooperating in cortical F-actin stiffening of mitotic HeLa cells (*Cao et al., 2020*). Furthermore, the actin nucleator JMY was found to interact with the Arp2/3 complex in in vitro reconstitutions (*Zuchero et al., 2009*; *Firat-Karalar et al., 2011*). However, as JMY RNAi did not cause any statistically significant decline in cell migration (*Zuchero et al., 2009*; *Firat-Karalar et al., 2011*) – a process firmly established to involve the Arp2/

3 complex – the functional importance of a putative cooperation of JMY with the Arp2/3 complex in the formation of actin filaments remains unclear.

While these initial observations and the in part apparently conflicting data show that we are only at the very beginning of identifying and understanding any cooperative functions of actin nucleators, the here studied cell biological process of dendritic branching highlights that clearly there are actin filament-driven cellular processes, which require the coordinated action of not only one but at least two effectors promoting the formation of F-actin. Our mechanistic and functional studies clearly demonstrate that with Cobl and Cobl-like shaping neurons into their complex morphologies involves regulated and physically coordinated interactions of different actin filament formation-promoting factors at the base of nascent dendritic protrusion sites.

# Materials and methods

## DNA constructs

Plasmids encoding for GFP-Cobl-like and parts thereof were described previously (*Izadi et al., 2018*) and generated by PCR using the EST clone UniProtID Q3UMF0 as template, respectively. GFP-Cobl-like$^{111-262}$ and Cobl-like$^{1-111}$ were generated by subcloning with the help of internal restriction sites. Additional Cobl-like deletion mutants were generated by combining the following forward primers, aa1 fw: 5'-AATTAGATCTATGGACCGCAGCGTCCCCGATCC-3'; aa261 fw: 5'-AAAGATCTGATATCAGCAGAGAG-3'; aa537 fw: 5'-AAAGATCTAAGGATCCTGATTCAGC-3'; aa740 fw: 5'-GCCTCAAGAGAATTCAGG-3'; aa376 fw: 5'-TTGAATTCTTAAACCATGATCGCTTC-3'; aa182 fw: 5'- TTAGATCTCCTACACCTATAATC-3' with the following reverse primers, aa457 rv: 5'-AACTCGAGCCCGGGACCAAGGGAGC-3'; aa741 rv: 5'-TCCTGAATTCTCTTGAGG-3'; aa540 rv: 5'-TTCTCGAGTTAATCAGGATCCTTCTC-3'; aa411 rv: 5'-GCAAGCTTGGTTTTCGAAGGTGG-3'; aa272 rv: 5'-AAGAATTCTCAGTTGTGTGATATTTG-3'; aa380 rv: 5'-TTGAATTCGAAGCGATCATGGTG-3' using either the introduced or an internal restriction site (1-538; BamH1). Cobl-like$^{1-741}$ fused to mCherry was generated by subcloning from GFP-Cobl-like$^{1-741}$ into Cherry-pCMV.

Cobl-like mutants lacking the N terminal CaM-binding site Cobl-like$^{\Delta CaM\ NT}$ ($\Delta$aa11-45) were generated by fusing a PCR product (primers, aa1-10+46–51 fw: 5'-AAAGATCTATGGACCGCAGCG TCCCGGATCCCGTACCCAAGAATCACAAATTCCTG-3' and aa741 rv) with Cobl-like$^{740-1273}$ using the internal EcoRI site (corresponding to aa740/741) to obtain the respective mutated full-length protein.

Cobl-like mutants lacking only the first 'KRAP' motif ($\Delta$aa59-69; $\Delta$KRAP1) were generated by fusing a DNA fragment obtained by PCR using an RNAi-resistant Cobl-like construct (*Izadi et al., 2018*) as template and primers aa1 fw and aa58 rv (5'-TTAAGCTTGCTCTGACAAATATG-3') with a second PCR product (primer aa70 fw and aa457 rv) using a HindIII site. The resulting Cobl-like$^{1-457\Delta KRAP1}$ was either used as such or fused with a Cobl-like$^{458-1273}$ fragment generated by Sma I digestion of Cobl-like RNAi/GFP-Cobl-like* to generate the respective full-length Cobl-like mutant Cobl-like*-$^{\Delta KRAP1}$ in pRNAT H1.1.

Cobl-like mutants lacking all three 'KRAP' motifs were generated by PCR using primers aa70 fw and aa333 5'-TTAAGCTTTGCATCCGAGGGC-3' rv and fusing the resulting PCR product with a second PCR product obtained by using primers aa413 fw 5'-TTAAGCTTCTGGCTCAGACTGATG-3' and aa457 rv as well as with a PCR product resulting from the above described aa1 fw and aa58 rv primers to give rise to aa59-69+334–412 deletion construct (Cobl-like$^{1-457\Delta KRAP}$). Using an internal SmaI restriction site, Cobl-like$^{1-457\Delta KRAP}$ was then fused to the more C terminal parts of Cobl-like to give rise to either Cobl-like$^{1-741\Delta KRAP}$ or Cobl-like$^{\Delta KRAP}$ mutants.

A Cobl-like deletion mutant lacking the N terminal Cobl Homology domain Cobl-like$^{\Delta 1-412}$ was generated by fusing a PCR product (primers, aa413 fw and aa741 rv) with Cobl-like$^{740-1273}$ to obtain Cobl-like$^{\Delta 1-412}$.

Plasmids encoding for GST fusion proteins of Cobl-like$^{1-411}$ were generated by subcloning into pGEX-4T-2 (GE Healthcare). A plasmid encoding for TrxHis-Cobl-like$^{1-411}$ was generated by PCR and subcloning into pET-32 (Novagen) (primers, aa1 fw; aa411 rv Sal I: 5'-GGGTCGACGGTTTTCGAAGG TGG-3') using EcoRI and SalI sites.

The RNAi construct directed against mouse and rat Cobl-like coexpressing GFP (Cobl-like RNAi#1) and scrambled RNAi control were described before (*Izadi et al., 2018*; *Pinyol et al., 2007*).

Additionally, Cobl-like RNAi and scrambled RNAi were inserted into a pRNAT vector coexpressing farnesylated mCherry (mCherryF) (pRNAT-mCherryF; *Schneider et al., 2014*). Plasmids for rescue attempts were built by replacing the GFP reporter by either RNAi-insensitive, GFP-Cobl-like (Cobl-like RNAi/Cobl-like* and scrambled RNAi/Cobl-like*; *Izadi et al., 2018*), or by mutant GFP-Cobl-like sequences generated based on Cobl-like*.

Plasmids encoding for GST-tagged syndapin I full-length and SH3 domain (aa376-441), respectively, as well as for a P434L-mutated SH3 (SdpI^SH3mut^) were described previously (*Qualmann et al., 1999*; *Qualmann and Kelly, 2000*). An alternative syndapin I SH3 domain (aa378-441) was as described (*Braun et al., 2005*).

Plasmids encoding for Xpress-tagged syndapin I full-length, GFP-syndapin I, Flag-syndapin I and mitochondrially targeted syndapin I (Mito-mCherry-SdpI), syndapin I ΔSH3 (Mito-mCherry-SdpI^ΔSH3^), and mCherry (Mito-mCherry) were described by *Qualmann et al., 1999*, *Kessels and Qualmann, 2006 Qualmann and Kelly, 2000*, *Kessels and Qualmann, 2002*, *Braun et al., 2005* and *Dharmalingam et al., 2009*, respectively. Syndapin I-mRubyRFP was generated by subcloning syndapin I into a derivative of pEGFP-N1 containing mRubyRFP instead of GFP.

The mCherryF-coexpressing SdpI RNAi (bp297-317; for validations, see *Dharmalingam et al., 2009*) vector and the corresponding control pRNAT vector were described previously (*Schneider et al., 2014*).

GST-SdpII^SH3^ (SdpII-l, aa383-488) and GST-SdpIII^SH3^ (aa366-425) were described before (*Qualmann and Kelly, 2000*; *Seemann et al., 2017*). Flag-syndapin II-s was as described (*Dharmalingam et al., 2009*). Flag-syndapin III was described before (*Braun et al., 2005*).

GFP-Cobl, Flag-mCherry-Cobl, and GFP-Cobl^1-713^ were described previously (*Hou et al., 2015*). Mito-GFP-Cobl^1-713^ was generated by subcloning the respective Cobl-encoding sequence into the mitochondrial targeting vector.

RNAi constructs against rat and mouse Cobl coexpressing farnesylated mCherry were generated by subcloning into pRNAT-mCherryF (*Schneider et al., 2014*). The control expressing scrambled RNAi and mCherryF were as described (*Schneider et al., 2014*).

A plasmid encoding for GFP-CaM was generated by subcloning from TrxHis-CaM described previously (*Hou et al., 2015*).

Correct cloning by PCR was verified by sequencing in all cases.

## Antibodies, reagents, and proteins

Rabbit anti-Cobl-like antibodies were raised against a combination of two GST-Cobl-like fusion proteins (GST-Cobl-like^537-741^ and GST-Cobl-like^740-1015^) as described previously (*Izadi et al., 2018*). The antibodies were affinity-purified according to procedures described previously (*Qualmann et al., 1999*; *Kessels et al., 2000*). Anti-syndapin I and anti-syndapin III antibodies were described previously (*Qualmann et al., 1999*; *Koch et al., 2011*). Anti-GST and anti-TrxHis antibodies from guinea pig and rabbit were described before, too (*Qualmann and Kelly, 2000*; *Braun et al., 2005*; *Schwintzer et al., 2011*).

Polyclonal rabbit anti-GFP was from Abcam (RRID:AB_303395). Monoclonal mouse anti-GFP antibodies (JL-8) were from Clontech(RRID:AB_10013427). Monoclonal mouse anti-Flag (M2) (RRID:AB_259529) and anti-MAP2 (HM-2) (RRID:AB_477193) antibodies as well as polyclonal rabbit anti-Flag antibodies were from Sigma-Aldrich (RRID:AB_439687). Anti-Xpress antibodies were from Invitrogen (RRID:AB_2556552).

Secondary antibodies used included, Alexa Fluor488- and 568-labeled goat anti-guinea pig antibodies (RRID:AB_142018 and RRID:AB_141954), Alexa Fluor488- and 568-labeled donkey anti-mouse antibodies (RRID:AB_141607 and RRID:AB_2534013), Alexa Fluor647- and 680-labeled goat anti-mouse antibodies (RRID:AB_141725 and RRID:AB_1965956), Alexa Fluor488-labeled donkey anti-rabbit (RRID:AB_141708), Alexa Fluor568-, 647-, and 680-labeled goat anti-rabbit antibodies (RRID:AB_143011; RRID:AB_141775; RRID:AB_2535758) (Molecular Probes) as well as DyLight800-conjugated goat anti-rabbit (RRID:AB_2556775) and anti-mouse antibodies (RRID:AB_2556774) (Thermo Fisher Scientific). Donkey anti-guinea pig antibodies coupled to IRDye680 and IRDye800, respectively, were from LI-COR Bioscience (RRID:AB_10956079 and RRID:AB_1850024). Donkey anti-rabbit and goat anti-guinea pig as well as anti-mouse-peroxidase antibodies were from Jackson ImmunoResearch and Dianova, respectively (RRID:AB_2617176; RRID:AB_10015282; RRID:AB_2337405); 10 nm gold-conjugated goat anti-guinea pig antibodies for electron microscopical

examinations of freeze-fractured samples were from BBI Solutions (RRID:AB_2892072). MitoTracker Deep Red 633 was from Molecular Probes.

Sepharose 4B-coupled CaM was from GE Healthcare. GST- and TrxHis-tagged fusion proteins were purified from *Escherichia coli* lysates using glutathione-agarose or -sepharose (Sigma-Aldrich; GenScript) and Talon metal affinity resin (Clontech), respectively, as described previously (*Schwintzer et al., 2011*; *Qualmann and Kelly, 2000*). After purification, fusion proteins were dialyzed against PBS, characterized by SDS-PAGE, and snap-frozen and stored at −80°C.

Tag-free syndapin I and III were generated by expressing both proteins in the pGEX-6P vector (GE Healthcare) and cutting of the GST tag from purified proteins using PreScission protease (GE Healthcare) in 150 mM NaCl, 2 mM DTT, and 20 mM HEPES pH 7.4 buffer overnight at 4°C (during dialysis after elution). Cleaved off GST and non-cleaved GST fusion proteins were removed with glutathione-sepharose.

## In vitro reconstitutions of direct protein-protein interactions

Direct protein/protein interactions were demonstrated by coprecipitation assays with combinations of recombinant TrxHis- and GST-tagged fusion proteins purified from *E. coli* and/or immobilized CaM, respectively, in 10 mM HEPES pH 7.4, 300 mM NaCl, 0.1 mM $MgCl_2$, 1% (v/v) Triton X-100 supplemented with EDTA-free protease inhibitor cocktail as well as in some cases with 500 µM $Ca^{2+}$ added.

Eluted proteins were analyzed by SDS-PAGE, transferred to PVDF membranes by either semi-dry or tank blotting and then subjected to immunodetection with anti-TrxHis and anti-GST antibodies. Primary antibodies were detected with fluorescent secondary antibodies using a Licor Odyssey System.

## Culture and transfection, and immunostaining of cells

Culturing of HEK293 (RRID:CVCL_0045) and COS-7 cells (RRID:CVCL_0224) and their transfection using TurboFect (Thermo Fisher Scientific) as well as their immunolabeling was essentially done as described (*Kessels et al., 2001*; *Haag et al., 2012*). The cell lines are regularly tested for mycoplasma and were mycoplasma-negative.

In reconstitutions and visualizations of protein complex formations at the surfaces of mitochondria in intact cells, mitochondria of COS-7 cells were stained with 0.2 µM MitoTracker Deep Red 633 in medium at 37°C for 30 min and cells were subsequently fixed with 4% (w/v) paraformaldehyde (PFA) for 7 min.

## Preparation of HEK293 cell lysates

HEK293 cells were washed with PBS 24–48 hr after transfection, harvested and subjected to sonication for 10 s and/or lyzed by incubation in lysis buffer (10 mM HEPES pH 7.4, 0.1 mM $MgCl_2$, 1 mM EGTA, 1% (v/v) Triton X-100) containing 150 mM NaCl and EDTA-free protease inhibitor Complete (Roche) for 20–30 min at 4°C (*Kessels and Qualmann, 2006*). Cell lysates were obtained as supernatants from centrifugations at 20,000×*g* (20 min at 4°C).

## Coprecipitation of proteins from HEK293 cell lysates

Coprecipitation experiments with extracts from HEK293 cells expressing different GFP fusion proteins were essentially performed as described before (*Qualmann et al., 1999*; *Schwintzer et al., 2011*). In brief, HEK293 cell lysates were incubated with purified, recombinant GST fusion proteins immobilized on glutathione-sepharose beads for 3 hr at 4°C. The reactions were then washed several times with lysis buffer containing 150 mM NaCl and EDTA-free protease inhibitor Complete. Bound protein complexes were subsequently eluted with 20 mM reduced glutathione, 120 mM NaCl, 50 mM Tris/HCl pH 8.0 (30 min RT) or obtained by boiling the beads in 4×SDS sample buffer.

For coprecipitations with CaM, HEK293 cell lysates were prepared in an EGTA-free lysis buffer containing 150 mM NaCl, EDTA-free protease inhibitor cocktail, and 200 µM calpain I inhibitor. Cell lysates were supplemented with either 1 mM EGTA or 500 µM $Ca^{2+}$ to obtain conditions without and with $Ca^{2+}$, respectively. After incubation with 25 µl CaM-sepharose 4B for 3 hr at 4°C and washing, bound proteins were isolated by boiling in 4×SDS sample buffer. Lysates, supernatants, and eluates were analyzed by immunoblotting.

Triple coprecipitations, that is, the examinations of GST-Cobl-like$^{1-411}$/syndapin/GFP-Cobl$^{1-713}$ complexes with either syndapin I or syndapin III as bridging component, were essentially performed as described above (lysis buffer containing 150 mM NaCl) except that the extracts from HEK293 cells expressing GFP-Cobl$^{1-713}$ were not only incubated with immobilized GST-Cobl-like$^{1-411}$ but also with tag-free syndapin I or syndapin III for 3 hr. Bound proteins were eluted with 20 mM reduced glutathione, 120 mM NaCl, 50 mM Tris/HCl pH 8.0. Eluates and supernatants were separated by SDS-PAGE and analyzed by anti-syndapin I/III, anti-GST, and anti-GFP immunoblotting.

## Coprecipitation of endogenous syndapin I from mouse brain lysates

For coprecipitation of endogenous syndapin I, brain lysates were prepared from mice sacrificed by cervical dislocation. Extracts were prepared using an Ultra Turrax homogenizer (Ika Ultra Turrax T5Fu; 20,000 rpm, 10 s) in lysis buffer containing EDTA-free protease inhibitor Complete and supplemented with 100 mM NaCl and 200 µM calpain inhibitor I. After clearing the lysates from cell debris by centrifugation at $1000 \times g$ for 20 min, the supernatants were used to precipitate endogenous syndapin I by TrxHis-Cobl-like$^{1-411}$ fusion proteins immobilized on Talon metal affinity resin. Bound proteins were eluted by boiling in sample buffer, separated by SDS-PAGE, and analyzed by anti-syndapin I immunoblotting.

## Heterologous and quantitative coimmunoprecipitation analyses

Heterologous coimmunoprecipitations addressing Cobl-like/syndapin I, syndapin II, and syndapin III interactions were done with lysates of HEK293 cells transfected with GFP-Cobl-like fusion proteins and GFP, respectively, in combination with Flag-tagged syndapins. The cell lysates were incubated with anti-Flag antibodies or non-immune IgGs in lysis buffer containing 100 mM NaCl and EDTA-free protease inhibitor Complete for 3 hr at 4°C. Antibody-associated protein complexes were isolated by 2 hr incubation with protein A agarose (Santa Cruz Biotechnology) at 4°C. The immunoprecipitates were washed with lysis buffer containing 100 mM NaCl, eluted from the matrix by boiling in a mix of 2 M (final) urea and SDS sample buffer and analyzed by immunoblotting.

Comparisons of GFP-Cobl-like$^{1-457}$ and Cobl-like$^{1-457\Delta KRAP1}$ for their ability to associate with Flag-syndapin I were also done by anti-GFP immunoprecipitations from lysates of transfected HEK293 cells generated according to the procedure described above.

For quantitative evaluations of the regulation of Cobl-like/syndapin I complexes, anti-GFP immunoprecipitations of GFP-Cobl-like fusion proteins were done in the presence (2 µM $CaCl_2$ added) and in the absence of $Ca^{2+}$ (1 mM EGTA added), respectively.

The amounts of coimmunoprecipitated Flag-syndapin I were quantified based on the detection of fluorescent antibody signals using a Licor Odyssey System providing a linear, quantitative read-out over several orders of magnitude. Anti-syndapin I coimmunoprecipitation signals were normalized to the amounts of anti-GFP signal representing the immunoprecipitated material. This ensured that similar amounts of GFP-Cobl-like proteins were examined for their extent of Flag-syndapin I coimmunoprecipitation. Both fluorescence signals were detected on the same blot using the two different fluorescence channels of the Licor Odyssey System. Data were expressed as percent difference from $Ca^{2+}$-free conditions.

## Endogenous coimmunoprecipitations from mouse brain extracts

Mice were sacrificed and the brain was cut into small pieces and homogenized in 10 mM HEPES pH 7.5, 30 mM NaCl, 0.1 mM $MgCl_2$, and 1 mM EGTA with protease inhibitors. Afterward, Triton X-100 was added (0.2% v/v final) and the homogenates were extracted for 1 hr at 4–6°C. The samples were then centrifuged at $100,000 \times g$ for 30 min at 4°C and the resulting supernatants (mouse brain lysates) were incubated with affinity-purified rabbit anti-Cobl-like antibodies and non-immune rabbit IgGs, respectively, bound to protein A agarose (preincubation at 4°C and washing with above buffer and 0.2% (v/v) Triton X-100 [CoIP buffer]). After 4 hr of incubation at 6°C, the proteins bound to the protein A agarose were washed with ice-cold CoIP buffer, eluted with SDS sample buffer (100°C, 5 min), and analyzed by immunoblotting using anti-Cobl-like and anti-syndapin I antibodies.

## Microscopy

Images were recorded as z-stacks using a Zeiss AxioObserver.Z1 microscope (Zeiss) equipped with an ApoTome, Plan-Apochromat 100×/1.4, 63×/1.4, 40×/1.3, and 20×/0.5 objectives and an Axio-Cam MRm CCD camera (Zeiss).

Digital images were recorded by ZEN2012 (PRID:SCR_013672). Image processing was done by Adobe Photoshop (RRID:SCR_014199).

## Spinning disk live microscopy of developing neurons

Primary rat hippocampal neurons were transiently transfected using Lipofectamine 2000 at DIV6. For imaging, the culture medium was replaced by 20 mM HEPES pH 7.4, 140 mM NaCl, 0.8 mM $MgCl_2$, 1.8 mM $CaCl_2$, 5 mM KCl, 5 mM D-glucose (live imaging buffer) adjusted to isoosmolarity using a freezing point osmometer (Osmomat 3000; Gonotec).

Live imaging was conducted at 37°C 16–24 hr after transfection employing an open coverslip holder, which was placed into a temperature- and $CO_2$-controlled incubator built around a spinning disk microscope based on a motorized Axio Observer (Zeiss). The microscope was equipped with a spinning disk unit CSU-X1A 5000, 488 nm/100 mW OPSL laser, and 561 nm/40 mW diode lasers as well as with a QuantEM 512SC EMCCD camera (Zeiss).

Images were taken as stacks of 7–17 images at Z-intervals of 0.31 μm depending on cellular morphology using a C-Apochromat objective (63×/1.20 W Korr M27; Zeiss). The time intervals were set to 10 s. Exposure times of 50–200 ms and 3–12% laser power were used.

Image processing was done using ZEN2012 and Adobe Photoshop software.

For quantitative determination of the degree of accumulation of Cobl-like, syndapin I, Cobl, and CaM fused to fluorescent proteins as well as for mCherry as control, the maximal fluorescence intensity was determined at an identified dendritic branch induction site in a time window prior to dendritic branch initiation (3D imaging frame rate, 10 s; six frames prior to protrusion start defined as t=0) and normalized to a neighboring ROI at the same dendrite.

For spatiotemporal analyses, the time points of the frames with the highest accumulation of Cobl-like, syndapin I, Cobl and CaM were averaged. In the rare case that two maxima of equal intensity occurred prior to branch initiation, both time values were considered in the averaging. As above, 3D imaging stacks were recorded every 10 s and six frames prior to protrusion start (defined as t=0) were evaluated.

## Culturing, transfection, and immunostaining of primary rat hippocampal neurons

Primary rat hippocampal neuronal cultures were prepared, maintained, and transfected as described previously (*Qualmann et al., 2004*; *Pinyol et al., 2007*; *Schwintzer et al., 2011*). In brief, neurons prepared from hippocampi of E18 rats were seeded at densities of about 60,000/well (24-well plate) and 200,000/well (12-well plate), respectively. Cells were cultured in Neurobasal medium containing 2 mM L-glutamine, 1× B27, and 1 μM/ml penicillin/streptomycin. The neurons were maintained at 37°C with 90% humidity and 5% $CO_2$.

Transfections were done in antibiotic-free medium using 2 μl Lipofectamine 2000 and 1 μg DNA per well in 24-well plates. After 4 hr, the transfection medium was replaced by conditioned medium and neurons were cultured further. All analyses were done with several independent neuronal preparations.

Fixation was done in 4% (w/v) PFA in PBS pH 7.4 at RT for 5 min. Permeabilization and blocking were done with 10% (v/v) horse serum, 5% (w/v) BSA in PBS with 0.2% (v/v) Triton X-100 (blocking solution). Antibody incubations were done in the same buffer without Triton X-100 according to *Kessels et al., 2001* and *Pinyol et al., 2007*. In brief, neurons were incubated with primary antibodies for 1 hr at RT and washed three times with blocking solution. Afterward, they were incubated with secondary antibodies (1 hr, RT). Finally, the coverslips were washed with blocking solution, PBS and water and mounted onto coverslips using Moviol.

## Quantitative analyses of dendrites of primary hippocampal neurons

Comparative Cobl and Cobl-like loss-of-function analyses were done 46 hr subsequent to transfection at DIV4 to allow for clear development of loss-of-function phenotypes. For Cobl-like loss-of-

function analyses and corresponding rescue experiments, 37 hr post-transfection (transfection at DIV4) was sufficient for loss-of-function phenotypes to clearly develop.

For suppressions of Cobl-like overexpression phenotypes, DIV4 hippocampal neurons were transfected with RNAi against Cobl, RNAi against syndapin I and control vectors, respectively, and fixed and immunostained about 34 hr later (DIV5.5). Suppression of Cobl overexpression phenotypes by Cobl-like RNAi was analyzed similarly. Due to the shorter time frame and/or due to the lower expression caused by cotransfection of two plasmids, the suppression effects exceed the effects of the RNAi effects alone evaluated in comparison. This experimental design helps to exclude putative merely additive, unrelated effects of the two manipulations working in opposite directions and simplifies the evaluation of whether a given knock-down can indeed suppress the function of GFP-Cobl and Cobl-like, respectively.

Two to six independent coverslips per condition per assay and neurons of at least two independent neuronal preparations were analyzed based on the anti-MAP2 immunostaining of transfected neurons.

Transfected neurons were sampled systematically on each coverslip. Morphometric measurements were based on anti-MAP2 immunolabeling of transfected neurons to identify dendrites.

Using IMARIS 7.6 software (RRID:SCR_007370), the number of dendritic branching points, dendritic terminal points, and dendritic filament length was determined and Sholl analyses (*Sholl, 1953*) were conducted according to procedures established previously (*Izadi et al., 2018*). For each neuron, a 'filament' (morphological trace) was drawn by IMARIS 7.6 software using the following settings: largest diameter, cell body diameter; thinnest diameter, 0.2 µm; start seed point, 1.5× of cell body diameter; disconnected points, 2 µm; minimum segment size, 10 µm. Immunopositive areas that were erroneously spliced by IMARIS or protrusions belonging to different cells as well as filament branch points that the software erroneously placed inside of the cell body were manually removed from the filament. Parameters determined were saved as Excel files and subjected to statistical significance calculations using GraphPad Prism5 and Prism6 software (RRID:SCR_002798).

## Quantitative, visual assessments of subcellular distributions

Quantitative determinations of the subcellular distribution of Cobl-like and the ΔKRAP1 mutant thereof were done using line scans across Apotome sections of the cell bodies of developing hippocampal neurons similar to methods described before (*Schwintzer et al., 2011*). The neurons were transfected at DIV4, immunostained for MAP2, and imaged 37 hr later. Cobl-like and the ΔKRAP1 mutant thereof were analyzed in a Cobl-like-deficient background by expressing Cobl-like RNAi vectors that coexpressed RNAi-resistant WT GFP-Cobl-like* and GFP-Cobl-like*$^{\Delta KRAP1}$, respectively. As described before (*Schwintzer et al., 2011*), the fluorescence intensities reached in cortical areas were extracted from the line scans and expressed in relation to the average intensity of an ROI covering a large part of the cytoplasmic area of the neuron.

## Freeze-fracturing and immunogold labeling

Hippocampal neurons were grown for 7 days on poly-D-lysine-coated sapphire disks (diameter 4 mm; Rudolf Brügger, Swiss Micro Technology) in 24-well plates, washed with PBS and subjected to ultrarapid freezing (4000 K/s) as well as to freeze-fracturing as described for mature neurons (*Schneider et al., 2014*).

Freeze-fracturing of developing neurons led to low yields of rather fragile replica of inner and outer membrane leaflets, which, however, were preserved during subsequent washing, blocking, and incubation (*Wolf et al., 2019*). Replica were incubated with guinea pig anti-syndapin I antibodies (1:50; overnight, 4°C) and 10 nm gold-conjugated secondary antibodies as described for mature neurons (*Schneider et al., 2014*).

Controls addressing the specificity of anti-syndapin I labeling of freeze-fracture replica included evaluations of labeling at the E-face (almost no labeling) and quantitative analyses of syndapin I labeling densities at control surfaces not representing cellular membranes (low, unspecific immunogold labels at a density of only 0.4/µm$^2$). Further controls including secondary antibody controls and labeling of syndapin I KO material were described previously (*Schneider et al., 2014*).

Replica were collected and analyzed using transmission electron microscopy and systematic grid explorations as described (*Schneider et al., 2014*; *Seemann et al., 2017*). Images were recorded

digitally and processed by using Adobe Photoshop software. All analyses were done with two independent neuronal preparations.

Membrane areas with parallel membrane orientations (cylindrical) were distinguished from protrusive topologies, as established previously (*Wolf et al., 2019*). Anti-syndapin I immunogold labeling densities were determined using the complete area of the respective membrane topology on each image (measured using ImageJ [RRID:SCR_003070]).

Anti-syndapin I cluster analyses were conducted using circular ROIs of 70 nm diameter. Density of clusters at cylindrical and protrusive membranes were calculated considering ≥3 anti-syndapin I labels per ROI as one cluster. Additionally, anti-syndapin I labeling being single, paired, and clustered in three to eight labels, respectively, was analyzed as percent of total anti-syndapin labeling.

## Statistical analyses and sample size estimation

No explicit power analyses were used to compute and predefine required sample sizes. Instead, all neuronal analyses were conducted by systematic sampling of transfected cells across coverslips to avoid any bias. Morphometric analyses were then conducted using IMARIS software.

All data were obtained from two to five independent neuronal preparations seeded onto several independent coverslips for each condition for transfection and immunostaining. For each condition, n numbers of individual neurons ranging from about 30 to 40 were aimed for to fully cover the biological variances of the cells. Higher n numbers yielded from the systematic sampling were accepted, too (e.g. see the control in *Figure 1G–I* [n=45] and in *Figure 1N–P* [65]). Lower n numbers were only accepted for the established Cobl overexpression phenotype (n=24) and the Cobl-like-mediated suppression of it (also n=24), as results were clear and sacrificing further rats for further primary neuron preparations could thus be avoided (*Figure 1N–P*).

Outliers or strongly scattering data reflect biological variance and were thus not excluded from the analyses.

All n numbers are reported directly in the figures of the manuscript and all are numbers of independent biological samples (i.e. neurons) or biochemical assays, as additional replicates to minimize measurement errors were not required because the technical errors were small in relation to the biological/biochemical variances. In live imaging analyses (*Figure 5C*), the n numbers given are dendritic branch induction events (18-21 from 4 to 12 neurons). In the EM experiments (*Figure 6B,D*), the n numbers are dendritic segments, i.e. EM images (34 and 41, respectively).

All quantitative biochemical data (*Figure 7E*, *Figure 9D*, and *Figure 9F*) as well as the determinations of accumulations of proteins at dendritic branch sites (*Figure 5C*) and the quantitative analyses of subcellular distributions (*Figure 9I*) are provided as bar and dot plot overlays to report the individual raw data and the deviations of individual data points around the mean.

Quantitative data represent mean ± SEM throughout the manuscript. Exceptions are *Figure 5D*, in which the mean is reported with both ± SD and ± SEM, *Figure 9F*, in which data represents mean±absolute error (n=2; no difference), and *Figure 6C* (the percent of total labeling shown in *Figure 6C* per definition has no error).

Normal data distribution and statistical significance were tested using GraphPad Prism 5 and Prism 6 software (SCR_002798). The statistical tests employed are reported in the respective figure legends.

Dendritic arbor parameters (number of dendritic branch points, number of terminal points, and total dendritic length) were analyzed for statistical significance employing one-way ANOVA and Tukey post-test throughout.

All Sholl analyses were tested by two-way ANOVA and Bonferroni post-test.

Quantitative evaluation of syndapin I coimmunoprecipitations with Cobl-like[1-741], Cobl-like[1-457], and Cobl-like[1-457ΔKRAP1] were analyzed by unpaired Student's t-test.

Anti-syndapin I immunogold labeling densities at different surfaces of freeze-fractured replica of membranes of developing neurons were analyzed by one-way ANOVA and the densities of anti-syndapin I clusters were analyzed by two-tailed Student's t-test.

Statistical significances were marked by *p < 0.05, **p < 0.01, ***p < 0.001, and ****p < 0.0001 throughout. In addition, the numbers of p-values are reported directly in the figures. Note that for p < 0.0001 (****), no values were provided by the software Prism 6, as the p-values are too small.

### Ethics statement

As exclusively cells and tissue samples isolated from postmortem WT animals were used in this study, neither a permission of animal experiments nor a breeding permission for genetically modified animals (*Zuchtrahmenantrag*) was required.

Mice and rats used to obtain biological material were bred by the animal facility of the Jena University Hospital in strict compliance with the European Union guidelines for animal experiments and approved by the *Thüringer Landesamt für Verbraucherschutz*.

All biological samples from mice and rat were obtained from animals that were sacrificed by cervical dislocation by trained personal. The training status of personal involved was approved by the *Thüringer Landesamt für Verbraucherschutz* in the context of the breeding permission *ZRA* (*UKJ-17– 021*). Embryos were removed from uteri and additionally decapitated.

## Acknowledgements

We thank A Kreusch, B Schade, and K Gluth for excellent technical support. This work was supported by *DFG* grants KE685/4-2 to MMK as well as QU116/6-2 and QU116/9-1 to BQ.

## Additional information

### Funding

| Funder | Grant reference number | Author |
|---|---|---|
| Deutsche Forschungsgemeinschaft | KE685/4-2 | Michael M Kessels |
| Deutsche Forschungsgemeinschaft | QU116/6-2 | Britta Qualmann |
| Deutsche Forschungsgemeinschaft | QU116/9-1 | Britta Qualmann |

The funders had no role in study design, data collection and interpretation, or the decision to submit the work for publication.

### Author contributions

Maryam Izadi, Conceptualization, Data curation, Formal analysis, Validation, Investigation, Visualization, Methodology, Writing - original draft; Eric Seemann, Data curation, Investigation, Visualization; Dirk Schlobinski, Lukas Schwintzer, Investigation; Britta Qualmann, Conceptualization, Supervision, Funding acquisition, Writing - Original draft; Writing - review and editing; Michael M Kessels, Conceptualization, Data curation, Supervision, Funding acquisition, Visualization, Writing - original draft, Writing - review and editing

### Author ORCIDs

Britta Qualmann ![ORCID] https://orcid.org/0000-0002-5743-5764
Michael M Kessels ![ORCID] https://orcid.org/0000-0001-5967-0744

### Ethics

Animal experimentation: As exclusively cells and tissue samples isolated from postmortem WT animals were used in this study, neither a permission of animal experiments nor a breeding permission for genetically modified animals (Zuchtrahmenantrag) was required. Mice and rats used to obtain biological material were bred by the animal facility of the Jena University Hospital in strict compliance with the European Union guidelines for animal experiments and approved by the Thüringer Landesamt für Verbraucherschutz.

### Decision letter and Author response

Decision letter https://doi.org/10.7554/eLife.67718.sa1
Author response https://doi.org/10.7554/eLife.67718.sa2

## Additional files

### Supplementary files
• Transparent reporting form

### Data availability
All data generated or analysed during this study are included in the manuscript and supporting files. Source data files have been provided and cover all quantitative data shown in the figures and their supplements.

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

# Appendix 1

**Appendix 1—key resources table**

| Reagent type (*species*) or resource | Designation | Source or reference | Identifiers | Additional information |
|---|---|---|---|---|
| Gene (*Mus musculus*) | *Cobl-like (Cobll1)* | *Izadi et al., 2018* | AK144943.1, GI: 74201418 | |
| Gene (*Mus musculus*) | *Cordon-Bleu (Cobl)* | *Ahuja et al., 2007* | NM_172496.3, GI: 162135965 | The common abbreviation of Cordon-Bleu is Cobl |
| Gene (*Rattus norvegicus*) | *Syndapin I (Pacsin1)* | *Braun et al., 2005* | AF104402.1, GI: 4324451 | *Syndapin* is used as gene name in most model organisms, such as rat, worms, flies |
| Gene (*Rattus norvegicus*) | *Calmodulin (Calm1)* | *SenGupta et al., 1987* | M19312.1, GI: 203255 | The common abbreviation of calmodulin is CaM |
| Strain, background (*Escherichia coli*) | *E. coli* commercial strain BL21-CodonPlus(DE3)-RIPL | Agilent | Cat#230280 | |
| Strain, background (*Escherichia coli*) | *E. coli* commercial strain XL10-Gold | Agilent | Cat#200314 | |
| Cell line (African green monkey) | COS-7 | Cell Lines Services GmbH | RRID:CVCL_0224 | |
| Cell line (human) | HEK293 | Cell Lines Services GmbH | RRID:CVCL_0045 | |
| Biological sample (*Rattus norvegicus*) | Primary hippocampal neurons (Wistar rat; Crl:WI; mixed sex) | Charles River | RRID:RGD_68115 | Primary neurons isolated from E18 rat embryos (sex undetermined) |
| Biological sample (*Mus musculus*) | Isolated brains (Mouse strain C57BL/6J, female) | Jackson Labs | RRID:IMSR_JAX:000664 | Brain material processed for protein biochemical examinations |
| Antibody | Anti-Cobl-like (Rabbit polyclonal) | *Izadi et al., 2018* | N/A | WB (1:1000) |
| Antibody | Anti-syndapin I (Guinea pig polyclonal) | *Braun et al., 2005* | N/A | WB (1:500) EM (1:50) |
| Antibody | Anti-syndapin III (Guinea pig polyclonal) | *Koch et al., 2011* | N/A | WB (1:500) |
| Antibody | Anti-TrxHis (Rabbit polyclonal) | This paper | N/A | WB (1:1000) |
| Antibody | Anti-GST (Rabbit polyclonal) | *Qualmann and Kelly, 2000* | N/A | WB (1:1000) |

*Continued on next page*

*Appendix 1—key resources table continued*

| Reagent type (*species*) or resource | Designation | Source or reference | Identifiers | Additional information |
|---|---|---|---|---|
| Antibody | Anti-TrxHis (Guinea pig polyclonal) | *Schwintzer et al., 2011* | N/A | WB (1:2000) |
| Antibody | Anti-GST (Guinea pig polyclonal) | *Braun et al., 2005* | N/A | WB (1:1000) |
| Antibody | Anti-GFP (ab290) (Rabbit polyclonal) | Abcam | Cat#ab290 RRID:AB_303395 | WB (1:2000) |
| Antibody | Anti-GFP (JL-8) (Mouse monoclonal) | Clontech | Cat#632380 RRID:AB_10013427 | WB (1:4000) |
| Antibody | Anti-Flag antibody (M2) (Mouse monoclonal) | Sigma-Aldrich | Cat#F3165 RRID:AB_259529 | WB (1:500) |
| Antibody | Anti-MAP2 (HM-2) (Mouse monoclonal) | Sigma-Aldrich | Cat#M4403 RRID:AB_477193 | IF (1:500) |
| Antibody | Anti-Flag antibody (Rabbit polyclonal) | Sigma-Aldrich | Cat#F7425 RRID:AB_439687 | WB (1:1000) |
| Antibody | Anti-Xpress antibody (Mouse monoclonal) | Invitrogen | Cat#R910-25; RRID:AB_2556552 | IF (1:500) |
| Antibody | Alexa Fluor488-labeled goat anti-guinea pig (Goat polyclonal) | Molecular Probes | Cat#A-11073 RRID:AB_142018 | IF (1:1000) |
| Antibody | Alexa Fluor568-labeled goat anti-guinea pig (Goat polyclonal) | Molecular Probes | Cat#A-11075 RRID:AB_141954 | IF (1:1000) |
| Antibody | Alexa Fluor488-labeled donkey anti-mouse (Donkey polyclonal) | Molecular Probes | Cat#A-21202 RRID:AB_141607 | IF (1:1000) |
| Antibody | Alexa Fluor568-labeled donkey anti-mouse (Donkey polyclonal) | Molecular Probes | Cat#A10037 RRID:AB_2534013 | IF (1:1000) |
| Antibody | Alexa Fluor647-labeled goat anti-mouse (Goat polyclonal) | Molecular Probes | Cat#A-21236 RRID:AB_141725 | IF (1:1000) |
| Antibody | Alexa Fluor488-labeled donkey anti-rabbit (Donkey polyclonal) | Molecular Probes | Cat#A-21206 RRID:AB_141708 | IF (1:1000) |

*Continued on next page*

*Appendix 1—key resources table continued*

| Reagent type (*species*) or resource | Designation | Source or reference | Identifiers | Additional information |
|---|---|---|---|---|
| Antibody | Alexa Fluor568-labeled goat anti-rabbit (Goat polyclonal) | Molecular Probes | Cat#A-11036 RRID:AB_143011 | IF (1:1000) |
| Antibody | Alexa Fluor647-labeled goat anti-rabbit (Goat polyclonal) | Molecular Probes | Cat#A-21245 RRID:AB_141775 | IF (1:1000) |
| Antibody | Alexa Fluor680-labeled goat-anti-rabbit (Goat polyclonal) | Molecular Probes | Cat#A-21109 RRID:AB_2535758 | WB (1:10000) |
| Antibody | Alexa Fluor680-labeled goat-anti-mouse (Goat polyclonal) | Molecular Probes | Cat#35519 RRID:AB_1965956 | WB (1:10000) |
| Antibody | DyLight800-conjugated goat anti-rabbit (Goat polyclonal) | Thermo Fisher Scientific | Cat#SA5-35571 RRID:AB_2556775 | WB (1:10000) |
| Antibody | DyLight800-conjugated goat anti-mouse (Goat polyclonal) | Thermo Fisher Scientific | Cat#SA5-35521 RRID:AB_2556774 | WB (1:10000) |
| Antibody | IRDye680-conjugated donkey anti-guinea pig (Donkey polyclonal) | LI-COR Bioscience | Cat#926–68077 RRID:AB_10956079 | WB (1:10000) |
| Antibody | IRDye800-conjugated donkey anti-guinea pig (Donkey polyclonal) | LI-COR Bioscience | Cat#926–32411 RRID:AB_1850024 | WB (1:10000) |
| Antibody | Peroxidase-AffiniPure donkey anti-rabbit antibody (Donkey polyclonal) | Jackson ImmunoResearch Labs | Cat#711-035-152 RRID:AB_10015282 | WB (1:5000) |
| Antibody | Peroxidase-AffiniPure goat anti-guinea pig antibody (Goat polyclonal) | Jackson ImmunoResearch Labs | Cat#106-036-003 RRID:AB_2337405 | WB (1:5000) |
| Antibody | Peroxidase-goat F (ab')2 anti-mouse (Goat polyclonal) | Dianova | Cat#115-036-003 RRID:AB_2617176 | WB (1:5000) |
| Antibody | Goat anti-guinea pig IgG 10 nm gold (Goat polyclonal) | BBI Solutions | Cat#EM.GAG10 RRID:AB_2892072 | EM (1:50) |
| Recombinant DNA reagent | Flag-mCherry Cobl (Plasmid) | This paper | N/A | See Materials and methods |

*Continued on next page*

*Appendix 1—key resources table continued*

| Reagent type (*species*) or resource | Designation | Source or reference | Identifiers | Additional information |
|---|---|---|---|---|
| Recombinant DNA reagent | GFP-Cobl-like (Plasmid) | *Izadi et al., 2018* | N/A | |
| Recombinant DNA reagent | Scr. RNAi in pRNAT-H1.1 (Plasmid) | *Pinyol et al., 2007* | N/A | |
| Recombinant DNA reagent | Scr. RNAi in pRNAT-mCherryF (Plasmid) | *Schneider et al., 2014* | N/A | |
| Recombinant DNA reagent | Cobl-RNAi in pRNAT-mCherryF (Plasmid) | This paper | N/A | See Materials and methods |
| Recombinant DNA reagent | Cobl-like RNAi (#1) in pRNAT-H1.1 (Plasmid) | *Izadi et al., 2018* | N/A | |
| Recombinant DNA reagent | Cobl-like RNAi (#1) in pRNAT-mCherryF (Plasmid) | This paper | N/A | See Materials and methods |
| Recombinant DNA reagent | GFP-Cobl (Plasmid) | *Hou et al., 2015* | N/A | |
| Recombinant DNA reagent | GFP-Cobl$^{1-713}$ (Plasmid) | *Hou et al., 2015* | N/A | |
| Recombinant DNA reagent | Mito-GFP-Cobl$^{1-713}$ (Plasmid) | This paper | N/A | |
| Recombinant DNA reagent | GFP-Cobl-like$^{1-741}$ (Plasmid) | This paper | N/A | See Materials and methods |
| Recombinant DNA reagent | GFP-Cobl-like$^{740-1273}$ (Plasmid) | *Izadi et al., 2018* | N/A | |
| Recombinant DNA reagent | GFP-Cobl-like$^{1-538}$ (Plasmid) | This paper | N/A | See Materials and methods |
| Recombinant DNA reagent | GFP-Cobl-like$^{1-411}$ (Plasmid) | This paper | N/A | See Materials and methods |
| Recombinant DNA reagent | GFP-Cobl-like$^{1-380}$ (Plasmid) | This paper | N/A | See Materials and methods |
| Recombinant DNA reagent | GFP-Cobl-like$^{376-540}$ (Plasmid) | This paper | N/A | See Materials and methods |
| Recombinant DNA reagent | GFP-Cobl-like$^{261-380}$ (Plasmid) | This paper | N/A | See Materials and methods |
| Recombinant DNA reagent | GFP-Cobl-like$^{111-262}$ (Plasmid) | This paper | N/A | See Materials and methods |
| Recombinant DNA reagent | GFP-Cobl-like$^{1-111}$ (Plasmid) | This paper | N/A | See Materials and methods |
| Recombinant DNA reagent | GFP-Cobl-like$^{537-740}$ (Plasmid) | This paper | N/A | See Materials and methods |
| Recombinant DNA reagent | GFP-Cobl-like$^{182-272}$ (Plasmid) | This paper | N/A | See Materials and methods |
| Recombinant DNA reagent | GFP-Cobl-like$^{1-58}$ (Plasmid) | This paper | N/A | See Materials and methods |
| Recombinant DNA reagent | Cobl-like RNAi/ GFP-Cobl-like* in pRNAT H1.1 (Plasmid) | *Izadi et al., 2018* | N/A | |

*Continued on next page*

*Appendix 1—key resources table continued*

| Reagent type (*species*) or resource | Designation | Source or reference | Identifiers | Additional information |
|---|---|---|---|---|
| Recombinant DNA reagent | Cobl-like RNAi/ GFP-Cobl-like*$^{\Delta CaM\ NT}$ in pRNAT H1.1 (Plasmid) | This paper | N/A | See Materials and methods |
| Recombinant DNA reagent | mCherry-Cobl-like1-711 | This paper | N/A | See Materials and methods |
| Recombinant DNA reagent | GFP-Cobl-like$^{1-741\Delta CaM\ NT}$ (Plasmid) | This paper | N/A | See Materials and methods |
| Recombinant DNA reagent | GFP-Cobl-like$^{\Delta KRAP}$ (Plasmid) | This paper | N/A | See Materials and methods |
| Recombinant DNA reagent | GFP-Cobl-like$^{1-741\Delta KRAP}$ (Plasmid) | This paper | N/A | See Materials and methods |
| Recombinant DNA reagent | GFP-Cobl-like$^{1-457\Delta KRAP1}$ (Plasmid) | This paper | N/A | See Materials and methods |
| Recombinant DNA reagent | GFP-Cobl-like$^{1-457}$ (Plasmid) | This paper | N/A | See Materials and methods |
| Recombinant DNA reagent | Cobl-like RNAi/ GFP-Cobl-like*$^{\Delta KRAP1}$ in pRNAT H1.1 (Plasmid) | This paper | N/A | See Materials and methods |
| Recombinant DNA reagent | Cobl-like RNAi/ GFP-Cobl-like*$^{\Delta 1-412}$ in pRNAT H1.1 (Plasmid) | This paper | N/A | See Materials and methods |
| Recombinant DNA reagent | Flag-syndapin I (SdpI) (Plasmid) | *Qualmann and Kelly, 2000* | N/A | |
| Recombinant DNA reagent | Flag-syndapin II-s (SdpII) (Plasmid) | *Dharmalingam et al., 2009* | N/A | |
| Recombinant DNA reagent | Flag-syndapin III (SdpIII) (Plasmid) | *Braun et al., 2005* | N/A | |
| Recombinant DNA reagent | Xpress-syndapin I (SdpI) (Plasmid) | *Qualmann et al., 1999* | N/A | |
| Recombinant DNA reagent | Syndapin I–mRubyRFP (Plasmid) | This paper | N/A | See Materials and methods |
| Recombinant DNA reagent | GFP-syndapin I (Plasmid) | *Kessels and Qualmann, 2006* | N/A | |
| Recombinant DNA reagent | Mito-mCherry-SdpI (Plasmid) | *Kessels and Qualmann, 2002* | N/A | |
| Recombinant DNA reagent | Mito-mCherry-SdpI$^{\Delta SH3}$ (Plasmid) | *Braun et al., 2005* | N/A | |
| Recombinant DNA reagent | Mito-mCherry (Plasmid) | *Dharmalingam et al., 2009* | N/A | |
| Recombinant DNA reagent | SdpI-RNAi in pRNAT-mCherryF (Plasmid) | *Dharmalingam et al., 2009* *Schneider et al., 2014* | N/A | |
| Recombinant DNA reagent | GFP-CaM (Plasmid) | This paper | N/A | See Materials and methods |

*Continued on next page*

*Appendix 1—key resources table continued*

| Reagent type (*species*) or resource | Designation | Source or reference | Identifiers | Additional information |
|---|---|---|---|---|
| Sequence-based reagent | Cobl-like aa1 fw | This paper | PCR primer | 5′-AATTAGATCTATGGACC GCAGCGTCCCCGATCC-3′ (see Materials and methods) |
| Sequence-based reagent | Cobl-like aa261 fw | This paper | PCR primer | 5′-AAAGATCTGATATCAGCAG AGAG-3′ (see Materials and methods) |
| Sequence-based reagent | Cobl-like aa537 fw | This paper | PCR primer | 5′-AAAGATCTAAGGATCCT GATTCAGC-3′ (see Materials and methods) |
| Sequence-based reagent | Cobl-like aa740 fw | This paper | PCR primer | 5′-GCCTCAAGAGAATTCAGG-3′ (see Materials and methods) |
| Sequence-based reagent | Cobl-like aa376 fw | This paper | PCR primer | fw: 5′-TTGAATTCTTAAACCATGA TCGCTTC-3′ (see Materials and methods) |
| Sequence-based reagent | Cobl-like aa182 fw | This paper | PCR primer | 5′- TTAGATCTCCTACA CCTATAATC-3′ (see Materials and methods) |
| Sequence-based reagent | Cobl-like aa457 rv | This paper | PCR primer | 5′- AACTCGAGCCCGGG ACCAAGGGAGC-3′ (see Materials and methods) |
| Sequence-based reagent | Cobl-like aa741 rv | This paper | PCR primer | 5′-TCCTGAATTCTCTTGAGG-3′ (see Materials and methods) |
| Sequence-based reagent | Cobl-like aa540 rv | This paper | PCR primer | 5′-TTCTCGAGTTAATC AGGATCCTTCTC-3′ (see Materials and methods) |
| Sequence-based reagent | Cobl-like aa411 rv | This paper | PCR primer | 5′-GCAAGCTTGGTTTT CGAAGGTGG-3′ (see Materials and methods) |
| Sequence-based reagent | Cobl-like aa272 rv | This paper | PCR primer | 5′-AAGAATTCTCAGTT GTGTGATATTTG-3′ (see Materials and methods) |
| Sequence-based reagent | Cobl-like aa380 rv | This paper | PCR primer | 5′-TTGAATTCGAAGCGAT CATGGTG-3′ (see Materials and methods) |
| Sequence-based reagent | Cobl-like$^{\Delta CaM\ NT}$ aa1-10+46–51 fw | This paper | PCR primer | 5′-**AAAGATCTATGGACCGCAGCGT CCCGGATCCCGTACCCAAGAATCAC AAATTCCTG**-3′ (see Materials and methods) |
| Sequence-based reagent | Cobl-like aa413 fw | This paper | PCR primer | 5′-**TTAAGCTTCTGGCTCAGAC TGATG**-3′ (see Materials and methods) |
| Sequence-based reagent | Cobl-like aa58 rv | This paper | PCR primer | 5′-TTAAGCTTGCTCTGAC AAATATG-3′ (see Materials and methods) |
| Sequence-based reagent | Cobl-like aa70 fw | This paper | PCR primer | 5′-TTAAGCTTGCCGAGA CGAAGGGC-3′ (see Materials and methods) |
| Sequence-based reagent | Cobl-like aa333 rv | This paper | PCR primer | 5′-TTAAGCTTTGCATCCGAGGGC-3′ (see Materials and methods) |

*Appendix 1—key resources table continued*

| Reagent type (*species*) or resource | Designation | Source or reference | Identifiers | Additional information |
|---|---|---|---|---|
| Sequence-based reagent | Cobl-like aa411 rv Sal I | This paper | PCR primer | 5'-GGGTCGACGGTTTTC GAAGGTGG-3' (see Materials and methods) |
| Recombinant protein | TrxHis | *Hou et al., 2015* | N/A | |
| Recombinant protein | TrxHis-Cobl-like[1-411] | This paper | N/A | |
| Recombinant protein | GST-Cobl-like[1-411] | This paper | N/A | |
| Recombinant protein | GST-SdpI[SH3] | *Qualmann et al., 1999* | N/A | |
| Recombinant protein | GST-SdpII[SH3] | *Qualmann and Kelly, 2000* | N/A | |
| Recombinant protein | GST-SdpIII[SH3] | *Seemann et al., 2017* | N/A | |
| Recombinant protein | GST-SdpI | *Qualmann et al., 1999* | N/A | |
| Recombinant protein | GST-SdpI[SH3mut] | *Qualmann and Kelly, 2000* | N/A | |
| Commercial assay or kit | NucleoSpin Plasmid | Macherey-Nagel | Cat#740588.50 | |
| Commercial assay or kit | NucleoBond Xtra Midi | Macherey-Nagel | Cat#740410.50 | |
| Commercial assay or kit | Lipofectamine 2000 transfection reagent | Invitrogen | Cat#11668019 | |
| Commercial assay or kit | Turbofect transfection reagents | Thermo Fisher Scientific | Cat#R0532 | |
| Commercial assay or kit | Calmodulin Sepharose 4B | GE Healthcare | Cat#GE17-0529-01 | |
| Commercial assay or kit | PreScission protease | GE Healthcare | Cat#27-0843-01 | |
| Chemical compound, drug | MitoTracker Deep Red Alexa Fluor633 | Molecular Probes | Cat#M22426 | |
| Software, algorithm | ZEN2012 | Zeiss | RRID:SCR_013672 | |
| Software, algorithm | Prism5, Prism6 | GraphPad Prism | RRID:SCR_002798 | |
| Software, algorithm | ImageJ | Other | RRID:SCR_003070 | Open source software |
| Software, algorithm | IMARIS 7.6 | Bitplane | RRID:SCR_007370 | |
| Software, algorithm | Adobe Photoshop | Adobe | RRID:SCR_014199 | |

