## [Decision Letter]

**Acceptance summary:**

Your work provides additional insight into how the different actin regulators are coordinated to power dendritogenesis. Your live-cell imaging data represent an important first effort to determine the temporal sequence of events occurring at the membrane.

**Decision letter after peer review:**

Thank you for submitting your article "Functional interdependence of the actin nucleator Cobl and Cobl-like in dendritic arbor development" for consideration by *eLife*. Your article has now been reviewed by 3 peer reviewers, including Alphee Michelot as the Reviewing Editor and Reviewer #1, and the evaluation has been overseen by and Catherine Dulac as the Senior Editor.

The reviewers have discussed their reviews with one another, and the Reviewing Editor has drafted this to help you prepare a revised submission. This decision letter is long and detailed, but main comments 1 to 4 should be easy to address and only main comments 5 and 6 would require additional experiments.

Essential Revisions:

1. Some results appear inconsistent between different Figures. For example, in Figure 1D, Cobl RNAi shifts numbers of dendritic branch points from 10 to 6, while in Figure 2E, Cobl RNAi leaves numbers of dendritic branch points pretty much unchanged (around 7 or 8). Could you make sure that all data are consistent between Figures or explain apparent inconsistencies?

2. We find experiments of Figure 2 insufficient to conclude that Cobl and Cobl-like factors depend strictly on each other. One could imagine scenarios where effects of Cobl or Cobl-like are highly concentration dependent, and lead to detectable effects in cells only below or above certain thresholds (especially for multi-domain binding proteins such as Cobl and Cobl-like, which are likely to undergo complex phase transition behaviors when clustering at the membrane).

Therefore, we would recommend here simply to be very careful with wording in the conclusions of these experiments.

Other mentions such as (line 328) "their functions were cooperative", should also be avoided unless you provide further explanations; Mentions such as (line 101) "Functional redundancy seemed unlikely, because both individual loss-of-function phenotypes were severe." should be explained so that readers can assess whether functional redundancy is indeed unlikely or not (for example by referencing a paper describing mild versus severe phenotypes).

3. Some findings presented in the current manuscript were already published. While it is perfectly logical to base a study on previous findings and results, in the current manuscript the fraction of findings already published in previous manuscripts is non-negligible, which in some ways hides the originality of the data presented in the study.

For instance, the findings that Cobl-like is involved in the formation of dendritic branches and its localization at branch points (Figure 1, current manuscript) were already showed in a previous article from your group (Izadi et al., JCB 2018; Figure 3, Figure 4). The coordination of Cobl-like and ca^2+^/CaM in this process was already demonstrated in the same article (Figure 9, CaM inhibitor CGS9343B), even though in the previous article you focused more on the Cter ca^2+^/CaM binding site. Likewise, the coordinated role of Cobl and syndapin in the formation of dendritic branches and their localization at branch points was already demonstrated in previous studies (Schwintzer et al., EMBO J 2011; Hou et al., PlosBiol 2015). In these two articles you also demonstrated the crucial role of ca^2+^/CaM in that process.

As the current manuscript is very long (11 Figures), could you please present only new data and refer to previous papers when needed? We believe that this would give you an opportunity to limit the number of Figures and that the manuscript would overall gain in clarity.

4. The reviewers agree that a discussion on the role of various actin nucleation factors in neuronal development could benefit uninitiated readers. Could you also please discuss in more details other potential mechanisms of coordination between actin regulators based on your extensive previous studies and the existing literature?

5. In the present manuscript, you show, using fluorescence time-lapses, the co-localization of Cobl-like/Cobl, Cobl-like/Syndapin at branch points. In previous articles, your group demonstrated the localization at branching points of ca^2+^ spikes (using GCaMP), actin (using LifeAct), CaM, Cobl (Hou et al., Blos Biol 2015); co-localization of Cobl/Syndapin (Schwintzer et al., EMBO J 2011); co-localization of Cobl-like/Actin (LifeAct), Cobl-like/CaM (Izadi et al., JCB 2018).

However, in these previous studies and in the present manuscript, no quantifications were made concerning the spatio-temporal recruitments of these molecules. In this manuscript, it seems on top of that Cobl-like appears before syndapin, which would not be quite coherent with a recruitement of Cobl-like through syndapin.

Therefore, it would be important to quantify here more precisely the spatiotemporal relationship between Cobl, Cobl-like, syndapin and CaM (and ideally ca^2+^) during the formation of dendritic branches. This would require to record data at shorter time intervals, but you should have all the biological material necessary to do these experiments. You could use if possible the initiation of the protrusions as a time reference to then quantify the assembly and disassembly of the different molecular actors. This type of analysis has been performed previously for clathrin-mediated endocytosis (e.g. Taylor et al., Plos Biol 2011). For instance, you could measure evolutions of the fluorescence signals (e.g. fluorescence enrichment fluo foci/fluo outside) as a function of time before and after branch formation. It would also be very interesting to quantify the fraction of aborted or effective formation of branches according to the spatiotemporal evolution of the different molecular actors.

6. Another missing experiment in this story is whether this strong effect of ca^2+^/CaM in promoting Cobl-like's interaction with syndapin I through the first of the three "KRAP" motifs is indeed critical for Cobl-like recruitement at the membrane in cells. Could you provide experimental evidence that KRAP1 is directly involved in the ca^2+^/CaM-mediated recruitment of Cobl-like at the plasma membrane?

*Reviewer #1:*

This work investigates at the molecular and cellular levels the functional dependence of two actin filament nucleation factors, Cobl and Cobl-like proteins, in the formation of protrusive dendritic structures. Depletion of Cobl or Cobl-like lead to roughly similar phenotypes; overexpression of Cobl or Cobl-like induces excessive dendrite formation when the other protein is expressed at normal levels, but not when this other protein is depleted. Altogether, these observations lead the authors to conclude that these proteins work strictly interdependently. The authors then investigate how Cobl and Cobl-like are recruited, and identify syndapin as an essential component to bring Cobl and Cobl-like together at the membrane. This interaction is beautifully documented through a large number of pulldown experiments in vitro, and critical domains for these interactions are identified. These interactions are also confirmed in physiological conditions through ectopic localization experiments of those components to mitochondria. Syndapin I is identified as clusters at dendritic initiation sites by electron microscopy and all three components colocalize at the same nascent dendritic branch sites. In the last part of the manuscript, the authors further document the interaction between Cobl-like and syndapin, and find that calcium-dependent calmodulin binding to Cobl-like increases syndapin I's association through the first of the three KRAP's domains.

Comments to be addressed in a revised manuscript:

1. Some results appear inconsistent between different Figures. For example, in Figure 1D, Cobl RNAi shifts numbers of dendritic branch points from 10 to 6, while in Figure 2E, Cobl RNAi leaves numbers of dendritic branch points pretty much unchanged (around 7 or 8). Could the authors make sure that all data are consistent between Figures or explain apparent inconsistencies?

2. I find experiments of Figure 1 and 2 insufficient to conclude that Cobl and Cobl-like factors depend strictly on each other. One could imagine many scenarios where effects of Cobl or Cobl-like are highly concentration dependent, and lead to detectable effects in cells below or under certain thresholds (especially for multi-domain binding proteins such as Cobl and Cobl-like, which are likely to undergo complex phase transition behaviors when clustering at the membrane). Therefore, I would recommend the authors to be very careful with wording and conclusions of their experiments, and stick to what can strictly be concluded.

Other mentions such as (line 328) "their functions were cooperative", should also be avoided without any further explanations; Mentions such as (line 101) "Functional redundancy seemed unlikely, because both individual loss-of-function phenotypes were severe." should be explained so that readers can assess whether functional redundancy is indeed unlikely or not (for example by referencing a paper describing mild versus severe phenotypes).

3. One missing experiment in this story is whether this important effect of ca^2+^/CaM signaling promoting syndapin I's association with the first of the three "KRAP" motifs is key to account for Cobl-like's clustering at the plasma membrane. Could the authors measure the effect of calcium for Cobl-like (KRAP1 deleted) clustering at the plasma membrane (as compared to wild-type Cobl-like)?

4. I regret sometimes the lack of quantification for some experiments. For example, protein colocalization in cells should be quantified (for example by calculating Pearson's correlation coefficients of red and green signals at mitochondrial sites) because colocalization (or absence of) is not always obvious for non-expert eyes.

5. Figure 6 is beautiful, but I am wondering if these data could be exploited better. Is it possible to record data at shorter time intervals? It seems that Cobl-like appears before syndapin. Is that correct and if so, how is this coherent with a recruitement of Cobl-like through syndapin?

Recommendations for the authors:

1. Graph bar representation is not recommended nowadays. Please show individual data points with box and whisker plots to represent the variability of the data (and asymmetry of distributions). Also it would be useful to provide significance levels (α factors) in addition to p-values, and indicate exact α/p values in legends in addition of the stars in the Figures. On the contrary, statements such as "highly statistically significantly" (line 106) should be avoided. How p-values were calculated should also be mentioned.

2. Please make sure that all abbreviations are explained when mentioned first.

3. Mentions to unpublished efforts is not recommended anymore. Please show results from these efforts in a Supplementary Figure or do not mention them.

4. Please correct the following inaccuracy:

"Two powerful molecular machines for actin filament formation": These proteins are not machines, because they do not consume energy to form new filaments.

5. Figure 5A: Is it possible to present both gels with the same molecular weight scale so that corresponding bands are aligned?

*Reviewer #2:*

The manuscript by Izadi et al., "Functional interdependence of the actin nucleator Cobl and Cobl-like in dendritic arbor development" deals with the fundamental question of how actin regulators are orchestrated to control the formation of membranes protrusions during cells morphogenesis. In particular, the authors explored how actin nucleators are coordinated to trigger the formation of branches in neuronal dendritic arbor.

In that context, Cobl have a crucial role in dendritic arbor formation in neuronal cells. Cobl contains a repeat of three WH2 domains interacting with actin and enabling nucleation of new actin filaments (F-actin). The initial idea was that tandem repeat of WH2 domains could be sufficient to trigger F-actin nucleation. However, other studies have shown that the WH2 repeat of Cobl has no nucleation activity of its own. Importantly, Cobl activity was shown to work in coordination with other actin regulators including the F-actin-binding protein Abp1 (Haag, J Neuro 2012) and the BAR domain protein syndapin (Schwintzer, EMBO J 2011).

The manuscript of Izadi et al. builds on previous articles from the same group, in particular a study demonstrating that Cobl-like, an evolutionary ancestor of Cobl, is also crucial for dendritic branching (Izadi et al., 2018 JCB). This previous article showed that like Cobl (Haag, J Neuro 2012), Cobl-like protein works in coordination with the F-actin-binding protein Abp1 and ca^2+^/CaM to promote dendritic branching through regulation of F-actin nucleation or/and assembly. In the current manuscript the authors showed that the two actin nucleators Cobl and Cobl-like proteins are interdependent to trigger dendritic branching.

The authors used functional assays by quantifying the formation of dendritic branches in primary hippocampal neurons. Using fluorescence microscopy and siRNA-based knockdowns, the authors showed that Cobl and Cobl-like are functionally interdependent during dendritic branch formation in dissociated hippocampal neurons. They showed that siRNA decreasing Cobl or Cobl-like expression reduced the number of dendritic branch points to the same extent. Fluorescence time-lapses indicated that Cobl and Cobl-like proteins co-localized at abortive and effective branching points. Furthermore, they showed that the increase in branching induced by Cobl-like overexpression is reversed by using a siRNA that decreases Cobl expression, they also performed the reciprocal experiments. Using a variety of biochemistry assays (co-immunoprecipitation, in vitro reconstitutions with purified components…) the authors demonstrated that Cobl and Cobl-like do not interact directly, but that Cobl-like associates with syndapins, as previously shown for Cobl (Schwintzer et al., 2011; Hou et al., 2015). Thus, syndapin is the molecular and functional link between Cobl and Cobl-like proteins. The authors performed a very thorough characterisation of the biochemical interactions between the Cobl-like protein and syndapins. Syndapins and Cobl-like interactions were direct and based on SH3 domain/Prolin rich motif interactions respectively on syndapins and Cobl-like. The Prolin rich motifs were located in 3 KRAP domains at the Nter of Cobl-like proteins. The authors also showed that the interaction of the Nter proximal KRAP domain with syndapin is ca^2+^/CaM dependent, and that this ca^2+^/CaM dependent interaction is crucial for the function of the Cobl-like protein in the regulation of dendritic arbor formation. The authors confirmed most of their biochemical results by visualizing the formation of protein complexes on the surface of mitochondria in intact COS-7 cells. They also used time-lapse fluorescent microscopy to demonstrate that Syndapin and Cobl-like are co-localized at sites of dendritic branch induction. Importantly, the authors used Immunogold labeling of freeze-fractured plasma membranes combined with electron microscopy. Using this strategy, they showed that membrane-bound syndapin nanoclusters are preferentially located at the base of protrusive membrane topologies in developing neurons. Throughout the manuscript, the authors confronted their biochemistry experiments with functional assays quantifying the formation of dendritic branches.

The overall conclusion of the manuscript is that a molecular complex involving Cobl, Cobl-like and syndapin and regulated by ca^2+^/CaM, promotes the formation of actin networks leading to dendritic protrusions to initiate dendritic branches. Importantly, this manuscript demonstrated that multiple actin nucleators can be coordinated in neurons to trigger the formation of subcellular structures.

The conclusions of the manuscript are, in most cases, convincingly supported by the results. In particular, the authors have performed a very comprehensive characterization of the biochemical interactions between Cobl, Cobl-like and syndapin, which are well supported by the functional results. However, the results found concerning the spatiotemporal relationship between Cobl, Cobl-like and syndapin during dendritic branch formation are more preliminary and do not take into account the roles of ca^2+^/CaM. In addition, some of the findings were already published by the same group in previous articles. Thus, there are a number of issues that need to be addressed by the authors. These critical points are the following: (1) Need for quantifications concerning the spatiotemporal relationship between Cobl, Cobl-like and syndapin during the formation of dendritic branches. (2) Some of the findings presented in this manuscript have already been published by the same group, which diminishes the inherent originality of this manuscript. Apart from the main points raised above, the manuscript is experimentally solid and contains interesting results that are likely to stimulate further experiments in the fields of actin cytoskeleton but also in the fields of cellular neurobiology and neurodevelopment.

1) Need for quantifications concerning the spatiotemporal relationship between Cobl, Cobl-like, syndapin, ca^2+^, CaM during the formation of dendritic branches.

In the present manuscript, the authors have shown, using fluorescence time-lapses, the co-localization of Cobl-like/Cobl, Cobl-like/Syndapin at branch points. In previous articles, the same group demonstrated the localization at branching points of ca^2+^ spikes (using GCaMP), actin (using LifeAct), CaM, Cobl (Hou et al., Blos Biol 2015); co-localization of Cobl/Syndapin (Schwintzer et al., EMBO J 2011); co-localization of Cobl-like/Actin (LifeAct), Cobl-like/CaM (Izadi et al., JCB 2018). However, in these previous studies and in the present manuscript, no quantifications were made concerning the spatio-temporal recruitments of these biomolecules.

The authors have emphasized the complex and subtle regulations of these biochemical interactions leading to the functional coordination of these actin regulators. This is actually one of the key points of the manuscript, the demonstration that Cobl, Cobl-like, Syndapin and ca^2+^/CaM are orchestrated at the molecular level to control dendritic branching.

It would be very interesting to quantify the spatiotemporal relationship between the formation of the branch point and the specific recruitment of all these molecular actors. The authors should use the initiation of the protrusion as a time reference to then quantify the assembly and disassembly of the different molecular actors. This type of analysis has been performed previously for clatherin-mediated endocytosis (e.g. Taylor et al., Plos Biol 2011). For instance, the authors could measure evolutions of the fluorescence signals (e.g. fluorescence enrichment fluo foci/fluo outside) as a function of time before and after branch formation. For this specific manuscript, the authors should quantify this for at least Cobl, Cobl-like, Syndapin, and CaM. It would also be very interesting to quantify the fraction of aborted or effective formation of branches according to the spatiotemporal evolution of the different molecular actors.

2) Some findings presented in the current manuscript were already published by the same group:

While it is perfectly logical to base a study on previous findings and results, in the current manuscript the fraction of findings already published in previous manuscripts is non-negligible, which in some ways limits the originality of the data presented in the study.

For instance, the findings that Cobl-like is involved in the formation of dendritic branches and its localization at branch points (Figure 1, current manuscript) were already showed in a previous article from the same group (Izadi et al., JCB 2018; Figure 3, Figure 4). The coordination of Cobl-like and ca^2+^/CaM in this process was already demonstrated in the same article (Figure 9, CaM inhibitor CGS9343B), even though in the previous article the focused more on the Cter ca^2+^/CaM binding site. Likewise, the coordinated role of Cobl and syndapin in the formation of dendritic branches and their localization at branch points were already demonstrated in previous studies form the same group (Schwintzer et al., EMBO J 2011; Hou et al., PlosBiol 2015). In these two articles they also demonstrated the crucial role of ca^2+^/CaM in that process.

The manuscript would gain in strength and originality if the authors could deepen their molecular understanding of the branching process. One way could be to quantify the spatiotemporal coordination of these different molecular players (Cobl, Cobl-like, Syndapin, CaM, ca^2+^, actin, N-WASP, Arp2/3…) in that process, as suggested in the point #1.

*Reviewer #3*:This manuscript by Izadi et al., explores the contribution of two actin nucleating proteins, Cobl and Cobl-like, to dendritic arborization. This work links CaCaM signaling with different post-translation modes of Cobl at the plasma membrane via a physical linkage between Cobl and Cobl-like proteins mediated by the F-BAR protein Syndapin I and coordination with the actin disassembly factor Cyclin-dependent kinase 1 (Srv2/CAP) to ultimately dictate actin-based neuromorphogenesis. The strength of this study includes a robust set of imaging and molecular biology analyses to show the localization and interaction of Cobl, Cobl-like, and Syndapin I. A potential weak point in this work is a lacking comparison between this actin nucleation mode and other neuronal actin nucleation proteins (i.e., Spire, Arp2/3 complex, or formin). This could allow readers to assess and/or compare the effectiveness of the Cobl and Cobl-like to previously discovered single actin-nucleation protein activities on neurogenesis.

– The authors claim this work is the first demonstration of actin nucleation factors working in concert to promote neuronal morphology. Synergistic promotion of actin assembly by netrin/WASP/Arp2/3 and combinations of Spire / formins and other ligands are essential for several cell processes including vesicle trafficking, DNA repair and neuronal morphology in purkingee neurons (Wagner et al., 2011, Pfender et al., 2011; Schuh, 2011; Montaville et al., 2014; Belin et al., 2015; Sundararajan 2019), although perhaps not shown as clearly as these authors in this work. They show many careful details of the interaction of cobl and cobl-like in neuronal morphology but do not compare how these nucleation effects compare to more other nucleation factors.

– All figures and images throughout this manuscript should be recolored to avoid red/green for comparison to allow to allow for interpretation by color blind individuals.

– In several instances throughout the manuscript the authors refer to "highly significant" results based on statistical analyses. For accuracy and clarity, the authors should refer to results as "significant" or "not significant" as the statical tests used do not indicate more than this.

– For the ease of the reader the N values for each analysis should be listed in each figure legend rather than the methods section.

– More discussion on the role of various actin nucleation factors in neuronal development could benefit uninitiated readers. What are the contributions of Arp2/3, formins, spire? For example, a preprint (Bradley et al. 2019) suggests the formin Capu and spire cooperate to stimulate actin nucleation from both (barbed and pointed) ends of actin filaments.

– The images of hippocampal neurons shown in each of the figures are gorgeous!

– Additional quantitative analysis of co localization could strengthen the localization argument presented in Figure 1H.

– The interdependence of Cobl and Cobl-like could be more convincing at more timepoints than the single- 34 h one. Is the interdependence consistent on different days of neuronal development?

– A direct binding event doesn't mediate the interdependent morphological phenotype. Thus, the authors explore whether Syndapin proteins are responsible for linking Cobl and Cobl-like from previous observations demonstrating a direct relationship between Syndapin I and Cobl at the plasma membrane. Could other proteins particularly spire or formin proteins also mediate connections between Cobl and Cobl-like?

– Lines 106-107: I think the authors mean to say "all were statistically significant compared to neurons cotransfected…"

– Line 150 "artifacts" is misspelled.

– Line 226 there is no "highly significant" result in statistics, it is either significant or not based on the p-value cut off used. Please reword accordingly.

– Define KO on line 381.

---

## [Author Response]

Essential Revisions:1. Some results appear inconsistent between different Figures. For example, in Figure 1D, Cobl RNAi shifts numbers of dendritic branch points from 10 to 6, while in Figure 2E, Cobl RNAi leaves numbers of dendritic branch points pretty much unchanged (around 7 or 8). Could you make sure that all data are consistent between Figures or explain apparent inconsistencies?

It is correct that the both the absolute numbers of dendritic branches, terminal points and dendritic length and also the relative effects of Cobl and Cobl-like RNAi differ in particular between former Figure 1 and 2. Roughly, one can say that the RNAi effects in former Figure 1,9 and 10 are about twice as strong as in the former Figure 2. There are two simple reasons for this, which we unfortunately failed to communicate properly in our original manuscript:

i) time frame of phenotype development and suppression, respectively and

ii) expression of only one versus two plasmids in the different types of experiments

Concerning i): The former Figure 2 and actually also the former Figure 4 (suppression by syndapin I RNAi) both are suppressions of gain-of-function phenotypes, whereas the former Figures 1, 9 and 10 are loss-of-function experiments. Because the gain-of-function effects represent strong and fast inductions of dendritic arborization it suffices to do a short transfection (max. 34 h) and then evaluate. Loss-of-function effects in conditions using the RNAi tools alone are not very strong at such short times when compared to control (all three analyzed parameters are -15-25% for the stronger Cobl-like RNAi and 0 to -10% for the weaker Cobl RNAi at this short time; former Figure 1, please see Figure 1—figure supplement 2 of the revised manuscript).

The loss-of-function experiments (former Figure 1, 9, 10) are different. The times need to be longer, as the phenotype is the normal growth of the dendritic arbor in controls vs. the putative suppression of this developmental process upon RNAi. Thus, transfections in these experiments usually need to be substantially longer (37-46 h) to show loss-of-function phenotypes compared to control – which then also may be more obvious (Cobl-like RNAi, -30 to -40 %; Cobl RNAi, -33% (***), -20% (**) and -10% (n.s.); former Figure 1D-F – now Figure 1—figure supplement 2).

Concerning ii): Suppression of gain-of-function experiments require the coexpression of two plasmids (one for the induction of the gain-of-function phenotype and the second for the RNAi (including reporter expression)), whereas we are able to drive loss-of-function/rescue experiments from only one plasmid driving both RNAi and the expression of a reporter or rescue mutant. Such transfections with two plasmids usually leads to gain-of-function but also suppression effects that are weaker than the effects of either overexpressing or knocking down proteins alone.

The revised manuscript now provides information on the different time frames of transfection (see improved and expanded Figure legends and Material and Method section) and also briefly touches on the coexpression issue leading to different numbers in the different types of experiments.

2. We find experiments of Figure 2 insufficient to conclude that Cobl and Cobl-like factors depend strictly on each other. One could imagine scenarios where effects of Cobl or Cobl-like are highly concentration dependent, and lead to detectable effects in cells only below or above certain thresholds (especially for multi-domain binding proteins such as Cobl and Cobl-like, which are likely to undergo complex phase transition behaviors when clustering at the membrane).Therefore we would recommend here simply to be very careful with wording in the conclusions of these experiments.

We share the reviewer’s concerns that suppression experiments are sometimes difficult to interpret, if the experiments are not designed in a careful manner and/or show a complex outcome. This is not the case in our experiments, however (see details below).

Of strong concern would be the following outcome: A presence of significant RNAi effect(s) alone compared to control and the results of the suppression attempt and the RNAi run for comparison are not equal but the effects of conducting RNAi alone are stronger. In this case of experimental outcome, one should rather abstain from any interpretation and try to adapt the experimental design to reach a clear conclusion. The reason is that, in this particular case, two processes (one positive, the other one negative) could simply operate in parallel, may not necessarily have anything to do with each other directly and may potentially be affected by unspecifiable dose effects as well – thus the experiment is not informative.

In our experiments, the situation is different and the revised manuscript now contains an elucidation of the considerations required for a correct interpretation for the two vice versa suppression experiments we conducted and reported in the former Figure 2 (Figure 1C-P in the revised manuscript).

In general, the reviewers will acknowledge that when component A is able to elicit a certain cell biological effect and this does not happen when component B is not present, then component A’s functions depend on B. This is a very classical experimental design and conclusion. The same can also be done with inhibitors – then A’s functions depend on B’s activity. However, it is absolutely crucial that the individual effects of the manipulations as well as the baseline control values are considered in the interpretation, too. If the suppression of the overexpression effect is larger than any putative RNAi effects compared to control or there is no such RNAi effect, the experiment and interpretation actually is very straight forward.

In our study, this is the case for Cobl RNAi in the suppression of Cobl-like functions (Figure 1C-I in the revised manuscript): We observed complete suppression of Cobl-like’s effects with Cobl-like RNAi. Yet, the effects of GFP+Cobl RNAi expression are not distinguishable from control and the result thus is straight forward to interpret. We actually designed the experiment in a way that the individual RNAi conditions remained neglectable to reach this straight forward interpretation scenario.

The same applies to the suppression of Cobl-like effects by syndapin I RNAi (Figure 3 in the revised manuscript). Under the conditions shown, syndapin I RNAi would not cause any phenotypes, yet, it completely suppressed the strong Cobl-like-mediated effects on all four parameters of dendritic arborization determined (former Figure 4; now Figure 3 in the revised manuscript).

For the suppression of the Cobl gain-of-function phenotypes by Cobl-like RNAi (Figure 1J-P in the revised manuscript) the situation is a bit less obvious and we understand the concern of the reviewer that this may need a more detailed look. Here, in all three parameters shown, GFP+Cobl-like RNAi causes a relatively mild but significant phenotype when compared to GFP+Scrambled control. However, the reviewer will acknowledge that the RNAi effects deviating negatively from the GFP+Scrambled control are much smaller than the suppression of the Cobl-mediated effects on dendritic arborization, which are twice as high (branch points; total dendritic length) and three times as high (terminal branches), respectively. Thus, also here, we clearly observe a suppression of specifically Cobl functions and can exclude additive actions in opposite directions. Importantly, this conclusion is formally further underscored by the fact that in all three phenotypical analyses GFP-Cobl+Cobl-like RNAi and GFP+Cobl-like RNAi are not statistically different from one another but equal (Figure 1J-P in the revised manuscript). This makes the interpretation of the results of also this suppression experiment straight forward again.

Other mentions such as (line 328) "their functions were cooperative", should also be avoided unless you provide further explanations.

Please see our statement above. If A needs B to work and B relies on A, that simply means A and B work together and not in parallel – and working in together is nothing else than cooperation. The revised manuscript now contains an elucidation of the considerations required for a correct interpretation for the two vice versa suppression experiments.

Mentions such as (line 101) "Functional redundancy seemed unlikely, because both individual loss-of-function phenotypes were severe." should be explained so that readers can assess whether functional redundancy is indeed unlikely or not (for example by referencing a paper describing mild versus severe phenotypes).

We apologize for the too much shortened argumentation in the original manuscript. A parallel action of Cobl and Cobl-like appeared unlikely because the DIV4-to-DIV6 developmental phenotypes of both components were so severe that a third of the entire arborization normally reached at DIV6 was lacking when only one of the two components was knocked down.

This paragraph has been changed in the revised manuscript and now explains better why parallel action of Cobl and Cobl-like appeared unlikely and why we thus addressed the alternative hypothesis.

3. Some findings presented in the current manuscript were already published. While it is perfectly logical to base a study on previous findings and results, in the current manuscript the fraction of findings already published in previous manuscripts is non-negligible, which in some ways hides the originality of the data presented in the study.For instance, the findings that Cobl-like is involved in the formation of dendritic branches and its localization at branch points (Figure 1, current manuscript) were already showed in a previous article from your group (Izadi et al., JCB 2018; Figure 3, Figure 4). The coordination of Cobl-like and ca^2+^/CaM in this process was already demonstrated in the same article (Figure 9, CaM inhibitor CGS9343B), even though in the previous article you focused more on the Cter ca^2+^/CaM binding site. Likewise, the coordinated role of Cobl and syndapin in the formation of dendritic branches and their localization at branch points was already demonstrated in previous studies (Schwintzer et al., EMBO J 2011; Hou et al., PlosBiol 2015). In these two articles you also demonstrated the crucial role of ca^2+^/CaM in that process.As the current manuscript is very long (11 Figures), could you please present only new data and refer to previous papers when needed? We believe that this would give you an opportunity to limit the number of Figures and that the manuscript would overall gain in clarity.

We have to admit that we are a bit irritated to be confronted with the claim that our manuscript to a “non-negligible” part contains data already published before. We usually carefully avoid publishing redundant data.

i) Concerning Figure 1: We acknowledge that the initial side-by-side comparison of Cobl and Cobl-like loss-of-function phenotypes can be considered as partially redundant with the literature, as the Cobl-like phenotype has been reported as an IMARIS-based evaluation before (Izadi et al., 2018 *J Cell Biol.*). However, the Cobl loss-of-function phenotype has not been evaluated in detail before, as all previous publications unfortunately only included a very limited manual analyses of branch points and of the number of primary dendrites (Ahuja et al., 2007 *Cell;* Schwintzer et al., 2011 *EMBO J;* Haag et al., *2012 J. Neurosci.;* Hou et al., 2015 *PLoS; Hou* et al., 2018 *Dev Cell*) – it thus was impossible to compare the phenotypes of Cobl to Cobl-like loss-of-function because neither terminal points, the total dendritic length nor Sholl analyses of the entire dendritic arbor have ever been analyzed before. Besides this obvious novelty, we have described which hypothesis led to the necessity of a full evaluation of Cobl loss-of-function effects in early neuronal development and of a side-by-side comparison with full consistency on the cell, method and experimenter sides. Since this work is the logical starting point of our study we cannot omit this data entirely but moved this block of data into the Supplemental Material (please see new Figure 1-Supplement2) and fused the remaining panels of the former Figure 1 with Figure 2 (now Figure 1 in revised manuscript) to additionally reduce the numbers of main figures from formerly 11 to now 10, as demanded.

ii) Concerning Figure 9: It is correct that we have demonstrated before by the use of the CaM inhibitor CGS9343B that Cobl-like functions require ca^2+^/CaM signaling (Izadi et al., 2018 J. Cell Biol.). However, these data are NOT recapitulated here. In the current study we instead identify a binding site of CaM in the N terminal part of Cobl-like and address the functions of specifically this site by in vitro, in vivo and functional analyses using a mutant that specifically lacks this site. None of this has been published before. The allegation of redundant publishing of data thus is completely unfounded.

In case the reviewer meant to question the novelty that Cobl-like functions in principle have something to do with ca^2+^/CaM signaling, we herewith stress that nowhere in the manuscript such general novelty was claimed but that we exclusive focused at the newly identified ca^2+^/CaM binding site and its molecular and cell biological functions.

Furthermore, we would like to remind the reviewer that it is valuable for a cell biological understanding of a given protein to evaluate a second site of binding to some interaction partner or a second or third site of modification mechanistically and functionally – please e.g. just have a look at all the individual phosphorylation sites of the many tyrosine kinase receptors and their (in part) different functions, which have been published in a plethora of independent publications over decades.

iii) Likewise, it is correct that it is known that syndapin interacts with Cobl (Schwintzer et al., 2011 EMBO J.). Here, however, we for the first time show an interaction of Cobl-like with syndapin I. Furthermore, we unveil that syndapin I can physically link and thereby coordinate Cobl with syndapin I’s new binding partner Cobl-like. And finally we demonstrate how this novel Cobl-like interaction with syndapin I is regulated.

All three points are novel and we thus fail to understand the background of the reviewer’s allegation of redundant publishing of data.

iv) Concerning reducing the figure number and citing more literature: The manuscript does refer to data previously reported wherever possible (the originally submitted manuscript already had more than 60 literature citations).

In order to comply with the suggestion to reduce the figure number, the revised manuscript now presents the former Figure 1 in the Supplemental Material (new Figure 1-Supplement). Despite the fact that the reviewers also asked for additional examinations to be included (see below), the revised manuscript therefore now contains 10 instead of 11 figures.

4. The reviewers agree that a discussion on the role of various actin nucleation factors in neuronal development could benefit uninitiated readers. Could you also please discuss in more details other potential mechanisms of coordination between actin regulators based on your extensive previous studies and the existing literature?

The revised manuscript now covers ten papers suggesting some crosstalk between actin nucleators in the discussion.

Most of these studies are not studying any processes in hippocampal neurons, still, though, they serve as great examples for valuable hints towards the fact that there may be much more complexity and coordination of different players leading to the formation of actin filaments than we currently think.

5. In the present manuscript, you show, using fluorescence time-lapses, the co-localization of Cobl-like/Cobl, Cobl-like/Syndapin at branch points. In previous articles, your group demonstrated the localization at branching points of ca^2+^ spikes (using GCaMP), actin (using LifeAct), CaM, Cobl (Hou et al., Blos Biol 2015); co-localization of Cobl/Syndapin (Schwintzer et al., EMBO J 2011); co-localization of Cobl-like/Actin (LifeAct), Cobl-like/CaM (Izadi et al., JCB 2018).However, in these previous studies and in the present manuscript, no quantifications were made concerning the spatio-temporal recruitments of these molecules. In this manuscript, it seems on top of that Cobl-like appears before syndapin, which would not be quite coherent with a recruitement of Cobl-like through syndapin.Therefore, it would be important to quantify here more precisely the spatiotemporal relationship between Cobl, Cobl-like, syndapin and CaM (and ideally ca^2+^) during the formation of dendritic branches. This would require to record data at shorter time intervals, but you should have all the biological material necessary to do these experiments. You could use if possible the initiation of the protrusions as a time reference to then quantify the assembly and disassembly of the different molecular actors. This type of analysis has been performed previously for clathrin-mediated endocytosis (e.g. Taylor et al., Plos Biol 2011). For instance, you could measure evolutions of the fluorescence signals (e.g. fluorescence enrichment fluo foci/fluo outside) as a function of time before and after branch formation. It would also be very interesting to quantify the fraction of aborted or effective formation of branches according to the spatiotemporal evolution of the different molecular actors.

We thank the reviewers for these further suggestion. It is indeed obvious that spatiotemporal analyses of different factors in dendritic branch induction is an attractive area of research offering a chance of a deeper understanding of the process. It seems to be a complex process involving quite a variety of molecular players. At the moment, one has to admit that we are mostly still at the stage of identifying critical and involved players and of investigating their individual properties and interactions, respectively. With the current study, we for the first time put three of these players and their CaM-coordinated action into the spot light. The revised manuscript now also contains some quantitative spatiotemporal data showing and comparing the degree of accumulation of these players and their average peak time prior to branch induction (please see newly added data panels Figure 5C and D).

We used the first morphological protrusion as reference point for the analysis of the critical time period prior to this, as suggested by the reviewers. As the reviewers will see from the new data added our data nicely confirms that not only Cobl (Hou et al., 2015) and Cobl-like (Izadi et al., 2018) show elevated values of accumulation prior to branch induction but that also syndapin I and even also CaM do so (see newly added Figure 5C).

Although the technical challenges were significant (primary neurons are hard to transfect, the developmental processes of dendritic branching occurs with rather moderate frequency somewhere in the extended dendritic arbor of a given neurons, and the analysis nevertheless requires a time resolution in the range of seconds and quite some 3D spatial resolution), we nevertheless were able to obtain n numbers of protrusions high enough to evaluate for each of these four proteins. The newly added data clearly makes the point that syndapin I, Cobl-like, Cobl and CaM in average show spatiotemporal overlap in a time window ranging from about -30 to -10 seconds prior to branch initiation (see newly added Figure 5D).

Interestingly, it furthermore seemed that Cobl-like together with syndapin I appeared earlier than Cobl, while CaM seemed to peak inbetween. The latter may be related to complex formations of CaM with both Cobl and Cobl-like and with the coordination of all three components by CaM signaling we revealed in our study.

Beyond these additional efforts made in our revision work for all four components studied here (Cobl-like, syndapin I, Cobl and CaM), we agree with the reviewer that comprehensive and detailed spatiotemporal analyses of all – in part yet to be identified – molecular players in dendritic branch induction will in the future certainly represent a powerful research avenue towards a better understanding of the temporal order of the actions of these players, of how they may be coordinated to bring about a new dendritic branch during development but may be also during repair processes of neurons.

6. Another missing experiment in this story is whether this strong effect of ca^2+^/CaM in promoting Cobl-like's interaction with syndapin I through the first of the three "KRAP" motifs is indeed critical for Cobl-like recruitement at the membrane in cells. Could you provide experimental evidence that KRAP1 is directly involved in the ca^2+^/CaM-mediated recruitment of Cobl-like at the plasma membrane?

In order to address a putative impact of the first, ca^2+^/CaM-regulated KRAP motif on the membrane recruitment of Cobl-like, we knocked-down endogenous Cobl-like and then quantified the membrane-association of reexpressed, RNAi-insensitive Cobl-like lacking KRAP1 at the plasma membrane of neurons in comparison to wild-type Cobl-like. Although KRAP1 is only one out of three identified syndapin I binding sites, we observed that deletion of merely this one site had a profound, statistically significant (p<0.0001; ****) impact on Cobl-like’s membrane localization in developing hippocampal neurons. This data obtained in our revision work is reported as Figure 9G-I in the revised manuscript.

Reviewer #1:This work investigates at the molecular and cellular levels the functional dependence of two actin filament nucleation factors, Cobl and Cobl-like proteins, in the formation of protrusive dendritic structures. Depletion of Cobl or Cobl-like lead to roughly similar phenotypes; overexpression of Cobl or Cobl-like induces excessive dendrite formation when the other protein is expressed at normal levels, but not when this other protein is depleted. Altogether, these observations lead the authors to conclude that these proteins work strictly interdependently. The authors then investigate how Cobl and Cobl-like are recruited, and identify syndapin as an essential component to bring Cobl and Cobl-like together at the membrane. This interaction is beautifully documented through a large number of pulldown experiments in vitro, and critical domains for these interactions are identified. These interactions are also confirmed in physiological conditions through ectopic localization experiments of those components to mitochondria. Syndapin I is identified as clusters at dendritic initiation sites by electron microscopy and all three components colocalize at the same nascent dendritic branch sites. In the last part of the manuscript, the authors further document the interaction between Cobl-like and syndapin, and find that calcium-dependent calmodulin binding to Cobl-like increases syndapin I's association through the first of the three KRAP's domains.Comments to be addressed in a revised manuscript:1. Some results appear inconsistent between different Figures. For example, in Figure 1D, Cobl RNAi shifts numbers of dendritic branch points from 10 to 6, while in Figure 2E, Cobl RNAi leaves numbers of dendritic branch points pretty much unchanged (around 7 or 8). Could the authors make sure that all data are consistent between Figures or explain apparent inconsistencies?

We thank the reviewer for his/her careful evaluation of our data. The discrepancy noticed, however, is only an apparent inconsistency, as the experimental set-ups and purposes were different. Please see our above detailed answer to Essential revision list point 1.

2. I find experiments of Figure 1 and 2 insufficient to conclude that Cobl and Cobl-like factors depend strictly on each other. One could imagine many scenarios where effects of Cobl or Cobl-like are highly concentration dependent, and lead to detectable effects in cells below or under certain thresholds (especially for multi-domain binding proteins such as Cobl and Cobl-like, which are likely to undergo complex phase transition behaviors when clustering at the membrane). Therefore, I would recommend the authors to be very careful with wording and conclusions of their experiments, and stick to what can strictly be concluded.

We share the reviewer’s concerns that suppression experiments are sometimes difficult to interpret. Please see our detailed response to this topic above (Essential Revision list point 2).

We hope the reviewer will be content with the revised version of our manuscript.

Other mentions such as (line 328) "their functions were cooperative", should also be avoided without any further explanations; Mentions such as (line 101) "Functional redundancy seemed unlikely, because both individual loss-of-function phenotypes were severe." should be explained so that readers can assess whether functional redundancy is indeed unlikely or not (for example by referencing a paper describing mild versus severe phenotypes).

As already written in the Essential Revision list above, we apologize for the too much shortened argumentation in the original manuscript. This paragraph has been changed in the revised manuscript and now explains better why parallel action of Cobl and Cobl-like appeared unlikely and why we thus addressed the alternative hypothesis.

3. One missing experiment in this story is whether this important effect of ca^2+^/CaM signaling promoting syndapin I's association with the first of the three "KRAP" motifs is key to account for Cobl-like's clustering at the plasma membrane. Could the authors measure the effect of calcium for Cobl-like (KRAP1 deleted) clustering at the plasma membrane (as compared to wild-type Cobl-like)?

We thank the reviewer for his/her suggestion of experiments suitable to significantly strengthen the manuscript. Please see our above response to the Essential revision list point 6.

In brief, this type of experimentation was done as part of our revision efforts during the last weeks. It demonstrated a remarkable strong impact of deletion of KRAP1 on Cobl-like’s membrane localization in developing hippocampal neurons and is now reported in the newly added revised Figure 9G-I.

4. I regret sometimes the lack of quantification for some experiments. For example, protein colocalization in cells should be quantified (for example by calculating Pearson's correlation coefficients of red and green signals at mitochondrial sites) because colocalization (or absence of) is not always obvious for non-expert eyes.

It may have been overlooked that calculating Pearson's correlation coefficients is not useful in our case, as we are not addressing a correlation of the occurrence of individual signals of one type with another type but are addressing coaccumulations of components under a given condition versus a more diffuse localization under other condition.

The original manuscript highlighted such coaccumulations by false-color heat map representations and marking sites of interest in two of our main figures.

In order to also comply with the reviewer’s request concerning the other figures (the in vivo protein complex reconstitutions at mitochondrial membrane surfaces), we added high-magnification insets to all of these figures in the main manuscript and in the Supplementary information visualizing in a more easily accessible manner than in the small full-size images whether the respective mitochondrial patterns are occurring or only a diffuse localization pattern prevails. We furthermore conducted line scans to quantitative visualize coincidences of elevated or diminished signal intensities. We hope that the reviewer is content with these additional figure panels added to many of our revised figures.

5. Figure 6 is beautiful, but I am wondering if these data could be exploited better. Is it possible to record data at shorter time intervals? It seems that Cobl-like appears before syndapin. Is that correct and if so, how is this coherent with a recruitement of Cobl-like through syndapin?

Please see our response to the Essential Revision list point 5 above. We acknowledge that analysis of the spatiotemporal relationship of molecular players involved in dendritic branch induction only is in its infancy, as at the current stage of research not even all important players of this process are known and this type of analysis is technically challenging to do in a quantitative manner in neurons.

The revised manuscript does now clearly demonstrate by quantitative evaluations of peak signal intensities that all four components studied (Cobl, Cobl-like, syndapin I and CaM) indeed show accumulation at branch induction sites prior to branch initiation. These data are quite well in line with the relative accumulation data collected for two of the components at the 30 s time point prior to protrusion initiation for Cobl (Hou et al., 2015 PLoS Biol.) and for Cobl-like (Izadi et al., 2018 J. Cell Biol.).

Furthermore the revised manuscript now contains a preliminary assessment of the average peak times of all for components highlighting that they indeed do not only show spatial but also temporal overlap at branch initiation sites, as it can be expected from our finding that Cobl-like and Cobl can be interconnected by Cobl-like’s novel interaction partner syndapin I in a CaM-regulated mechanism converging on one particular of the three syndapin I binding motifs we identified in Cobl-like. The Cobl-like and the syndapin I data hereby showed significant variances and a surprisingly early appearance of both components together. The data obtained thus far do not suggest that Cobl-like is recruited before syndapin but in average showed the same peak time (please see revised Figure 5C,D; former Figure 6). Thus, while we honestly do not claim that we have detailed enough data on the different aspects of the spatiotemporal behaviors of all players in dendritic branch initiation and this will definitively require further studies focusing on these aspects specifically, there at least is no discrepancy with any of the molecular mechanisms involving Cobl-like and syndapin I, which we demonstrate in this manuscript.

Recommendations for the authors:1. Graph bar representation is not recommended nowadays. Please show individual data points with box and whisker plots to represent the variability of the data (and asymmetry of distributions). Also it would be useful to provide significance levels (α factors) in addition to p-values, and indicate exact α/p values in legends in addition of the stars in the Figures. On the contrary, statements such as "highly statistically significantly" (line 106) should be avoided. How p-values were calculated should also be mentioned.

We hope that the reviewer is content with the accessibility of individual data points in our revised manuscript.

2. Please make sure that all abbreviations are explained when mentioned first.

This should be taken care of in the revised manuscript.

3. Mentions to unpublished efforts is not recommended anymore. Please show results from these efforts in a Supplementary Figure or do not mention them.

Done.

4. Please correct the following inaccuracy:"Two powerful molecular machines for actin filament formation": These proteins are not machines, because they do not consume energy to form new filaments.

Done.

5. Figure 5A: Is it possible to present both gels with the same molecular weight scale so that corresponding bands are aligned?

Done.

Reviewer #2:The manuscript by Izadi et al., "Functional interdependence of the actin nucleator Cobl and Cobl-like in dendritic arbor development" deals with the fundamental question of how actin regulators are orchestrated to control the formation of membranes protrusions during cells morphogenesis. In particular, the authors explored how actin nucleators are coordinated to trigger the formation of branches in neuronal dendritic arbor.In that context, Cobl have a crucial role in dendritic arbor formation in neuronal cells. Cobl contains a repeat of three WH2 domains interacting with actin and enabling nucleation of new actin filaments (F-actin). The initial idea was that tandem repeat of WH2 domains could be sufficient to trigger F-actin nucleation. However, other studies have shown that the WH2 repeat of Cobl has no nucleation activity of its own. Importantly, Cobl activity was shown to work in coordination with other actin regulators including the F-actin-binding protein Abp1 (Haag, J Neuro 2012) and the BAR domain protein syndapin (Schwintzer, EMBO J 2011).The manuscript of Izadi et al. builds on previous articles from the same group, in particular a study demonstrating that Cobl-like, an evolutionary ancestor of Cobl, is also crucial for dendritic branching (Izadi et al., 2018 JCB). This previous article showed that like Cobl (Haag, J Neuro 2012), Cobl-like protein works in coordination with the F-actin-binding protein Abp1 and ca^2+^/CaM to promote dendritic branching through regulation of F-actin nucleation or/and assembly. In the current manuscript the authors showed that the two actin nucleators Cobl and Cobl-like proteins are interdependent to trigger dendritic branching.The authors used functional assays by quantifying the formation of dendritic branches in primary hippocampal neurons. Using fluorescence microscopy and siRNA-based knockdowns, the authors showed that Cobl and Cobl-like are functionally interdependent during dendritic branch formation in dissociated hippocampal neurons. They showed that siRNA decreasing Cobl or Cobl-like expression reduced the number of dendritic branch points to the same extent. Fluorescence time-lapses indicated that Cobl and Cobl-like proteins co-localized at abortive and effective branching points. Furthermore, they showed that the increase in branching induced by Cobl-like overexpression is reversed by using a siRNA that decreases Cobl expression, they also performed the reciprocal experiments. Using a variety of biochemistry assays (co-immunoprecipitation, in vitro reconstitutions with purified components…) the authors demonstrated that Cobl and Cobl-like do not interact directly, but that Cobl-like associates with syndapins, as previously shown for Cobl (Schwintzer et al., 2011; Hou et al., 2015). Thus, syndapin is the molecular and functional link between Cobl and Cobl-like proteins. The authors performed a very thorough characterisation of the biochemical interactions between the Cobl-like protein and syndapins. Syndapins and Cobl-like interactions were direct and based on SH3 domain/Prolin rich motif interactions respectively on syndapins and Cobl-like. The Prolin rich motifs were located in 3 KRAP domains at the Nter of Cobl-like proteins. The authors also showed that the interaction of the Nter proximal KRAP domain with syndapin is ca^2+^/CaM dependent, and that this ca^2+^/CaM dependent interaction is crucial for the function of the Cobl-like protein in the regulation of dendritic arbor formation. The authors confirmed most of their biochemical results by visualizing the formation of protein complexes on the surface of mitochondria in intact COS-7 cells. They also used time-lapse fluorescent microscopy to demonstrate that Syndapin and Cobl-like are co-localized at sites of dendritic branch induction. Importantly, the authors used Immunogold labeling of freeze-fractured plasma membranes combined with electron microscopy. Using this strategy, they showed that membrane-bound syndapin nanoclusters are preferentially located at the base of protrusive membrane topologies in developing neurons. Throughout the manuscript, the authors confronted their biochemistry experiments with functional assays quantifying the formation of dendritic branches.The overall conclusion of the manuscript is that a molecular complex involving Cobl, Cobl-like and syndapin and regulated by ca^2+^/CaM, promotes the formation of actin networks leading to dendritic protrusions to initiate dendritic branches. Importantly, this manuscript demonstrated that multiple actin nucleators can be coordinated in neurons to trigger the formation of subcellular structures.The conclusions of the manuscript are, in most cases, convincingly supported by the results. In particular, the authors have performed a very comprehensive characterization of the biochemical interactions between Cobl, Cobl-like and syndapin, which are well supported by the functional results. However, the results found concerning the spatiotemporal relationship between Cobl, Cobl-like and syndapin during dendritic branch formation are more preliminary and do not take into account the roles of ca^2+^/CaM. In addition, some of the findings were already published by the same group in previous articles. Thus, there are a number of issues that need to be addressed by the authors. These critical points are the following: (1) Need for quantifications concerning the spatiotemporal relationship between Cobl, Cobl-like and syndapin during the formation of dendritic branches. (2) Some of the findings presented in this manuscript have already been published by the same group, which diminishes the inherent originality of this manuscript. Apart from the main points raised above, the manuscript is experimentally solid and contains interesting results that are likely to stimulate further experiments in the fields of actin cytoskeleton but also in the fields of cellular neurobiology and neurodevelopment.

We thank the reviewer for the positive assessment of the quality and impact of our work.

As far as the first point of the reviewer is concerned, the spatiotemporal relationship between Cobl, Cobl-like and syndapin I, please see our response to the Essential Revision list point 5 above.

We acknowledge that analysis of the spatiotemporal relationship of molecular players involved in dendritic branch induction only is in its infancy, as at the current stage of research not even all important players of this process are known and this type of analysis is technically challenging to do in a quantitative manner in neurons.

The revised manuscript does now clearly demonstrate by quantitative evaluations of peak signal intensities that all four components studied (Cobl, Cobl-like, syndapin I and CaM) indeed show accumulation at branch induction sites prior to branch initiation. These data are quite well in line with the relative accumulation data collected for two of the components at the 30 s time point prior to protrusion initiation for Cobl (Hou et al., 2015 PLoS Biol.) and for Cobl-like (Izadi et al., 2018 J. Cell Biol.).

Furthermore the revised manuscript now contains a preliminary assessment of the average peak times of all for components highlighting that they indeed do not only show spatial but also temporal overlap at branch initiation sites, as it can be expected from our finding that Cobl-like and Cobl can be interconnected by Cobl-like’s novel interaction partner syndapin I in a CaM-regulated mechanism converging on one particular of the three syndapin I binding motifs we identified in Cobl-like. The Cobl-like and the syndapin I data hereby showed significant variances and a surprisingly early appearance of both components together. The data obtained thus far suggest that Cobl-like and syndapin I are in average recruited at the same peak time, whereas Cobl may perhaps peak a bit later (n.s.) and CaM overlaps with both (please see revised Figure 5C,D).

However, even with these additional efforts we made during our revision work, one has to honestly admit that it is too early to claim that we have detailed enough data on the different aspects of the spatiotemporal behaviors of all players in dendritic branch initiation (which currently we may not all even have identified, yet). Although technically challenging to do at high enough resolution, with large enough time frames to capture the only relatively rare events of dendritic branch induction, at sufficient frame rates to not miss key events and with high enough numbers of transfected primary neurons of suitable developmental stages to reach sound quantitative data, this will require further comprehensive studies focusing on these aspects specifically.

As far as the second point of the reviewer is concerned, the criticism that some of the findings presented in this manuscript have already been published, please see our response to the Essential Revision list point 3 above.

As all other points presented are novel, this probably refers to the side-by-side, software-based, detailed evaluation of Cobl and Cobl-like loss-of-function phenotypes during early dendritic arborization originally presented in Figure 1. This data has been moved to the Supplemental Material (Figure 1—figure supplement 1) in the revised manuscript, as one half of the data set of course indeed merely is a reproduction of the Cobl-like phenotype identified by the same method before (Izadi et al., 2018).

However, the reviewers will acknowledge and readers will immediately understand that, without this comparison revealing the high degree of phenotypical copy, we would not have followed up and discovered the coordinated action of the two components powering actin filament formation during dendritic branch initiation we report here.

1) Need for quantifications concerning the spatiotemporal relationship between Cobl, Cobl-like, syndapin, ca^2+^, CaM during the formation of dendritic branches.In the present manuscript, the authors have shown, using fluorescence time-lapses, the co-localization of Cobl-like/Cobl, Cobl-like/Syndapin at branch points. In previous articles, the same group demonstrated the localization at branching points of ca^2+^ spikes (using GCaMP), actin (using LifeAct), CaM, Cobl (Hou et al., Blos Biol 2015); co-localization of Cobl/Syndapin (Schwintzer et al., EMBO J 2011); co-localization of Cobl-like/Actin (LifeAct), Cobl-like/CaM (Izadi et al., JCB 2018). However, in these previous studies and in the present manuscript, no quantifications were made concerning the spatio-temporal recruitments of these biomolecules.The authors have emphasized the complex and subtle regulations of these biochemical interactions leading to the functional coordination of these actin regulators. This is actually one of the key points of the manuscript, the demonstration that Cobl, Cobl-like, Syndapin and ca^2+^/CaM are orchestrated at the molecular level to control dendritic branching.It would be very interesting to quantify the spatiotemporal relationship between the formation of the branch point and the specific recruitment of all these molecular actors. The authors should use the initiation of the protrusion as a time reference to then quantify the assembly and disassembly of the different molecular actors. This type of analysis has been performed previously for clatherin-mediated endocytosis (e.g. Taylor et al., Plos Biol 2011). For instance, the authors could measure evolutions of the fluorescence signals (e.g. fluorescence enrichment fluo foci/fluo outside) as a function of time before and after branch formation. For this specific manuscript, the authors should quantify this for at least Cobl, Cobl-like, Syndapin, and CaM. It would also be very interesting to quantify the fraction of aborted or effective formation of branches according to the spatiotemporal evolution of the different molecular actors.

Please see our response to the Essential Revision list point 5 above. Please also see our response to this point in the review provided by the reviewer.

2) Some findings presented in the current manuscript were already published by the same group:While it is perfectly logical to base a study on previous findings and results, in the current manuscript the fraction of findings already published in previous manuscripts is non-negligible, which in some ways limits the originality of the data presented in the study.For instance, the findings that Cobl-like is involved in the formation of dendritic branches and its localization at branch points (Figure 1, current manuscript) were already showed in a previous article from the same group (Izadi et al., JCB 2018; Figure 3, Figure 4). The coordination of Cobl-like and ca^2+^/CaM in this process was already demonstrated in the same article (Figure 9, CaM inhibitor CGS9343B), even though in the previous article the focused more on the Cter ca^2+^/CaM binding site. Likewise, the coordinated role of Cobl and syndapin in the formation of dendritic branches and their localization at branch points were already demonstrated in previous studies form the same group (Schwintzer et al., EMBO J 2011; Hou et al., PlosBiol 2015). In these two articles they also demonstrated the crucial role of ca^2+^/CaM in that process.The manuscript would gain in strength and originality if the authors could deepen their molecular understanding of the branching process. One way could be to quantify the spatiotemporal coordination of these different molecular players (Cobl, Cobl-like, Syndapin, CaM, ca^2+^, actin, N-WASP, Arp2/3…) in that process, as suggested in the point #1.Reviewer #3:This manuscript by Izadi et al., explores the contribution of two actin nucleating proteins, Cobl and Cobl-like, to dendritic arborization. This work links CaCaM signaling with different post-translation modes of Cobl at the plasma membrane via a physical linkage between Cobl and Cobl-like proteins mediated by the F-BAR protein Syndapin I and coordination with the actin disassembly factor Cyclin-dependent kinase 1 (Srv2/CAP) to ultimately dictate actin-based neuromorphogenesis. The strength of this study includes a robust set of imaging and molecular biology analyses to show the localization and interaction of Cobl, Cobl-like, and Syndapin I. A potential weak point in this work is a lacking comparison between this actin nucleation mode and other neuronal actin nucleation proteins (i.e., Spire, Arp2/3 complex, or formin). This could allow readers to assess and/or compare the effectiveness of the Cobl and Cobl-like to previously discovered single actin-nucleation protein activities on neurogenesis.– The authors claim this work is the first demonstration of actin nucleation factors working in concert to promote neuronal morphology. Synergistic promotion of actin assembly by netrin/WASP/Arp2/3 and combinations of Spire / formins and other ligands are essential for several cell processes including vesicle trafficking, DNA repair and neuronal morphology in purkingee neurons (Wagner et al., 2011, Pfender et al., 2011; Schuh, 2011; Montaville et al., 2014; Belin et al., 2015; Sundararajan 2019), although perhaps not shown as clearly as these authors in this work. They show many careful details of the interaction of cobl and cobl-like in neuronal morphology but do not compare how these nucleation effects compare to more other nucleation factors.

Please see our response to Essential revision list point 4: The revised manuscript covers nine papers suggesting some crosstalk between actin nucleators in the discussion. These papers e.g. include the seminal work highlighting the cooperation of the actin nucleator Spire with formin 2 and for this also cites some of the literature the reviewer mentioned (Quinlan et al., 2007; Pfender et al., 2011; Montaville et al., 2014). The Belin et al. 2015 work on Spire and formin 2 in DNA damage repair is now also cited in the revised manuscript and expands the biological range of cooperations of these two actin nucleators.

Wagner et al., 2011 – in case the reviewer is referring to Wolfgang Wagner, Stephan D. Brenowitz, and John A. Hammer, III 2011 *Nat. Cell Biol.* – is a great paper but is dealing with mammalian Myosin-Va motor protein-driven transport of endoplasmic reticulum into spines in Purkinje cells and we failed to see the connection to actin nucleation. Sundararajan 2019, in case the reviewer referred to the PLoS Genetic paper, studies a set of mutants affecting actin filament dynamics in PVD neurons of *C. elegans* but to which extent and by which mechanisms these may or may not directly cooperate with each other remained somewhat unclear to us.

In general, we hope that the quite extensive and broad literature we report and cite highlights that there may be much more complexity and coordination of different players leading to the formation of actin filaments than we currently think and that this topic is an exciting field of contemporary research.

– All figures and images throughout this manuscript should be recolored to avoid red/green for comparison to allow to allow for interpretation by color blind individuals.

Taken care of.

– In several instances throughout the manuscript the authors refer to "highly significant" results based on statistical analyses. For accuracy and clarity, the authors should refer to results as "significant" or "not significant" as the statical tests used do not indicate more than this.

The word “highly” has been deleted from the revised manuscript, as the reviewer requested. However, the reviewer may acknowledge that a probability of error of almost 5% (P<0.05; *) clearly is less trustworthy than a probability of error of 0.1% (P<0.001; ***) or even 0.01% (P<0.0001; ****), as in our data. Thus, a finding never is just statistically significant or not (convention P<0.05) but the levels of statistical significance also need to be mentioned.

– For the ease of the reader the N values for each analysis should be listed in each figure legend rather than the methods section.

We would like to refer the reviewers to the fact that the n numbers for all of our quantitative evaluations were not only described in full detail in our Material and Method section but also reported directly in the figures.

If for editorial/journal style reasons indeed considered necessary, we shall be happy to repeat this information in all the figure legends yet again.

– More discussion on the role of various actin nucleation factors in neuronal development could benefit uninitiated readers. What are the contributions of Arp2/3, formins, spire? For example, a preprint (Bradley et al. 2019) suggests the formin Capu and spire cooperate to stimulate actin nucleation from both (barbed and pointed) ends of actin filaments.

This certainly is a research question that is not completely answered, as there are several cell biological processes in developing neurons that seem to be powered by actin filament formation and the list of actin filament formation-promoting factors discovered is still growing. As the topic is rather broad and in part also still in its infancy, we have not extensively covered this aspect but just wrote some introductory words about this and besides this rather focused on further hints on crosstalk between different actin nucleators in general to relate our work to observations made for other actin nucleators in different cell systems. We hope the reviewer is content with this solution.

– The images of hippocampal neurons shown in each of the figures are gorgeous!

We thank the reviewer for his/her appreciation of the beauty of nature.

– Additional quantitative analysis of co localization could strengthen the localization argument presented in Figure 1H.

As the distributions of both components previously shown in Figure 1H (now revised Figure 1A) are actually relatively continuously, quantitative analyses are not so easy, as e.g. calculating Pearson's correlation coefficients for example tests the coincidence of the occurrence or absence of two signals at one spot but not levels of coaccumulation. In the other figures we have added additional line scans of fluorescence signal intensity to provide quantitative access to correlating or not correlating peaks of signal intensities. Here, the best one can do to highlight the accumulation of both Cobl and Cobl-like at one spot is false-color heat map representations of signal intensities and marking of sites to show the perfect spatial fit.

We hope the reviewers will be content with the efforts made to visualize coaccumulations of the different proteins studied at particular sites within both COS-7 cells and primary hippocampal neurons.

– The interdependence of Cobl and Cobl-like could be more convincing at more timepoints than the single- 34 h one. Is the interdependence consistent on different days of neuronal development?

We have not addressed whether the interdependence of the two components would also be detectable at some other day of neuronal development, as we have little knowledge of the functions of Cobl or Cobl-like at other stages of development – let alone about any putative functional crosstalk to each other, to syndapin I, to Abp1, to PRMT2, to CaM or to some other components still to be identified. Thus, although this is an interesting question, due to this lack of basic knowledge, addressing any putative crosstalk would not represent a research question addressable in a sound manner at this stage.

– A direct binding event doesn't mediate the interdependent morphological phenotype. Thus, the authors explore whether Syndapin proteins are responsible for linking Cobl and Cobl-like from previous observations demonstrating a direct relationship between Syndapin I and Cobl at the plasma membrane. Could other proteins particularly spire or formin proteins also mediate connections between Cobl and Cobl-like?

In theory of course, yes. However, thus far we have no evidence that any spire or formin protein associates with Cobl or Cobl-like, although we are of course actively searching for further components interacting with Cobl and Cobl-like to better understand by which molecular mechanisms and accessory components they execute their cell biological functions.

– Lines 106-107: I think the authors mean to say "all were statistically significant compared to neurons cotransfected…"

The sentence has been changed in the revised manuscript. It should now be correct and easier to understand.

– Line 150 "artifacts" is misspelled.

We thank the reviewer for noticing this. The British spelling of the word has been changed to the American spelling in the revised manuscript.

– Line 226 there is no "highly significant" result in statistics, it is either significant or not based on the p-value cut off used. Please reword accordingly.

The word “highly” has been deleted in the revised manuscript, as the reviewer requested. However, the reviewer may acknowledge that a probability of error of almost 5% (P<0.05; *) clearly is less trustworthy than a probability of error of merely 0.1% (P<0.001; ***) or even 0.01% (P<0.0001; ****), as in our data. Thus, a finding never is just statistically significant or not (convention P<0.05) but the levels of statistical significance also need to be described and considered.

– Define KO on line 381.

Done.